# Ice-nucleating particle concentration measurements from Ny-Ålesund during the Arctic Spring-Summer in 2018

Matteo Rinaldi[1], Naruki Hiranuma[2], Gianni Santachiara[1], Mauro Mazzola[3], Karam Mansour[1,4,5], Marco Paglione[1], Cheyanne A. Rodriguez[2], Rita Traversi[6], Silvia Becagli[6], David Cappelletti[7,3], Franco Belosi[1]

[1]Institute of Atmospheric Sciences and Climate (ISAC), National Research Council (CNR), 40129 Bologna, Italy
[2]Department of Life, Earth and Environmental Sciences, West Texas A&M University, Canyon, TX, USA
[3]Institute of Polar Sciences (ISP), National Research Council (CNR), 40129 Bologna, Italy
[4]Department of Physics and Astronomy, University of Bologna, 40127 Bologna, Italy
[5]Department of Oceanography, Faculty of Science, University of Alexandria, 21511 Alexandria, Egypt
[6]Department of Chemistry "Ugo Schiff", University of Florence, 50019 Florence, Italy
[7]Dipartimento di Chimica, Biologia e Biotecnologie, Università degli Studi di Perugia, 06123 Perugia, Italy

*Correspondence to*: Matteo Rinaldi (m.rinaldi@isac.cnr.it)

**Abstract.**

In this study, we present atmospheric ice-nucleating particle (INP) concentrations from the Gruvebadet (GVB) observatory in Ny-Ålesund (Svalbard). All aerosol particle sampling activities were conducted in April – August 2018. Ambient INP concentrations ($n$INP) were measured for aerosol particles collected on filter samples by means of two offline instruments: the Dynamic Filter Processing Chamber (DFPC) and the West Texas Cryogenic Refrigerator Applied to Freezing Test system (WT-CRAFT) to assess condensation and immersion freezing, respectively. DFPC measured $n$INPs for a set of filters collected through two size-segregated inlets: one for transmitting particulate matter of less than 1 µm ($PM_1$), the other for particles with an aerodynamic diameter of less than 10 µm aerodynamic diameter ($PM_{10}$). Overall, $n$INP$_{PM10}$ measured by DFPC at a water saturation ratio of 1.02 ranged from 3 to 185 m$^{-3}$ at temperatures ($T$s) of -15 to -22°C. On average, the super-micrometer INP ($n$INP$_{PM10}$ - $n$INP$_{PM1}$) accounted for approximately 20-30% of $n$INP$_{PM10}$ in spring, increasing in summer to 45% at -22°C and 65% at -15°C. This increase of super-micrometre INP fraction towards summer suggests that super-micrometer aerosol particles play an important role as the source of INPs in the Arctic. For the same $T$ range, WT-CRAFT measured 1 to 199 m$^{-3}$. Although the two $n$INP datasets were in general agreement, a notable $n$INP offset was observed, particularly at -15°C. Interestingly, the results of both DFPC and WT-CRAFT measurements did not show a sharp increase in $n$INP from spring to summer. While an increase was observed in a subset of our data (WT-CRAFT, between -18 and -21°C), the spring-to-summer $n$INP enhancement ratios never exceeded a factor of three. More evident seasonal variability was found, howerver, in our activated fraction (AF) data, calculated by scaling the measured $n$INP to the total aerosol particle concentration. In 2018, AF increased from spring to summer. This seasonal AF trend corresponds to the overall decrease in aerosol concentration towards summer and a concomitant increase in the contribution of super-micrometre particles. Indeed, the AF of coarse particles resulted markedly higher than that of sub-micrometre ones (2 orders of magnitude). Analysis of low-travelling back-trajectories and meteorological conditions at GVB matched to our INP data suggests that the summertime INP population is influenced by

both terrestrial (snow-free land) and marine sources. Our spatiotemporal analyses of satellite retrieved Chlorophyll-a, as well as spatial source attribution, indicate that the maritime INPs at GVB may come from the seawaters surrounding the Svalbard archipelago and/or in proximity to Greenland and Iceland during the observation period. Nevertheless, further analyses, performed on larger datasets, would be necessary to reach firmer and more general conclusions.

## 1 Introduction

A climate-change sensitive region, the Arctic is experiencing a higher temperature ($T$) increase as compared to mid-latitudes due to a phenomenon called Arctic amplification (Serreze and Barry, 2011). While sea ice-albedo feedback appears to be a clear factor, the roles of other processes are difficult to quantify (Schmale et al., 2021). The contributions of forcing and feedback mechanisms associated with aerosol particles able to trigger heterogeneous ice nucleation, as well as Arctic mixed-phase cloud formation remains especially uncertain and unpredictable (Murray et al., 2021).

Arctic mixed-phase clouds are composed of both ice and supercooled liquid water and structured in persistent stratiform layers (Shupe et al., 2006; Shupe et al., 2011). In mixed-phase clouds, the hydrometeor phase formed through aerosol-cloud interactions plays an important role in determining cloud albedo and lifetime (de Boer et al., 2014). The strong sensitivity of stratiform mixed-phase cloud lifetime to the number of ice crystals was reported by Harrington and Olsson (2001) and Jiang et al. (2000). This effect was attributed in part to the Wegener-Bergeron-Findeisen mechanism (Bergeron, 1935; Findeisen, 1938; Wegener, 1911), whereby ice grows at the expense of liquid water due to its lower saturation vapour pressure. The resultant microphysical instability can glaciate clouds within a few hours or less (Jiang et al., 2000; Pinto, 1998; Rangno and Hobbs, 2001; Harrington et al., 1999). It follows that the presence of aerosol particles triggering heterogeneous ice nucleation (ice-nucleating particles, INPs) in the Arctic atmosphere can potentially have a substantial impact on precipitation formation, cloud radiative properties and climate (Solomon et al., 2018; Murray et al., 2021).

Atmospheric heterogeneous ice nucleation generally occurs by means of four major pathways: deposition, condensation, immersion and contact freezing (Vali et al., 2015). Ice formation by deposition occurs when the ambient is supersaturated with respect to ice in water-subsaturated conditions so that ice forms on an INP without prior formation of liquid. In condensation freezing, ice forms as water vapours condense on an INP at subzero $T$s, while nuclei immersed in a water droplet freeze via immersion freezing at sub-zero $T$s. In contact freezing, an INP promotes freezing when it comes into contact with a supercooled droplet from the outside. The distinction between condensation freezing and immersion freezing is still a matter of debate (Dymarska et al., 2006). The recent results of Wex et al. (2014) and Hiranuma et al. (2015) suggest that they might be the same process. However, the recent inter-comparison study with two different organic fiber samples shows a difference between condensation freezing and immersion freezing measurements (i.e., ice nucleation efficiency of the former is higher than the latter (Hiranuma et al., 2019). Further laboratory and field assessments are therefore necessary to understand the similarity of ice nucleation modes and processes.

In general, INPs can be of abiotic (e.g., mineral dust, volcanic ashes and soil dust) or biotic (e.g., bacteria, fungi, microalgae and pollen) origin (Hoose and Mohler, 2012; Murray et al., 2012). Seawater has been identified to be a source of ice nucleation active organic molecules (Knopf et al., 2011; Wang et al., 2015; Wilson et al., 2015) transferable to the atmosphere as a component of sea spray particles (e.g., McCluskey et al. (2017)). According to Hoose and Mohler (2012), mineral particles are typically dominant immersion- and condensation-mode INPs below -20°C with the exceptions of K-feldspar and quartz,

which can facilitate ice nucleation at much higher $T$s compared to other mineral compositions (Atkinson et al., 2013; Harrison et al., 2019). In addition, biogenic INPs tend to favour the formation of ice at relatively higher $T$s than abiotic INPs (Murray et al., 2012), even though even though the ice nucleation efficiency of biotic INPs varies widely (Kanji et al., 2017).

Only a few multi-season measurements of the Arctic INP concentration ($n$INP) are currently available. A summary of previously reported ground-based $n$INP measurements and $T$ ranges can be found in Table 1. The first ground-level $n$INP data

from the Arctic region were reported by Borys (1983). Measurements were performed with an offline dynamic condensation chamber at $T$s between -28 and -16°C in water saturation conditions. It was observed that pollution from lower latitudes contributed insignificantly to the Arctic INP burden as low $n$INP values coincided with the Arctic haze period. Bigg (1996) measured INPs active at -15°C in a static chamber and at relative humidity (RH) of just above 100% during an icebreaker cruise to the North Pole. The ocean was identified as a major source of INPs since $n$INP fell as a function of the length of time

that had elapsed since the air masses had left the open sea. Similar measurements were performed by Bigg and Leck (2001) in the central Arctic Ocean (20 July – 18 September 1996). The authors identified bacteria and fragments of marine organisms in the samples, suggesting biotic material to be the source of INPs in the Arctic.

More recent measurements of Arctic INPs were mostly performed in the immersion freezing mode. Conen et al. (2016) measured $n$INP at a coastal mountain observatory in Northern Norway. During the summer, the authors observed that $n$INP ($T$

of -15°C) in air masses from the ocean increased three fold after about one day of passage over land. Both marine and terrestrial INP sources were identified by Creamean et al. (2018) in the Northern Alaskan Arctic during spring. Irish et al. (2019) measured $n$INP in the Canadian Arctic marine boundary layer during summer 2014 on board the research ship Amundsen. $n$INP values correlated positively with total residence time over land and negatively with total residence time over sea ice and open water, suggesting a higher contribution of mineral dust particles than sea-spray-related sources. Similar conclusions were

found by Si et al. (2018) from measurements performed in the Canadian Arctic. Mason et al. (2016) found a large fraction of the INPs observed at the Alert station in spring and summer to belonged to the coarse size range. Likewise, a size-dependent ice nucleation efficiency, with larger particles being more ice active, was reported by Creamean et al. (2018) and Si et al. (2018). Recently, some studies have reported marked $n$INP seasonal variability in the Atlantic sector of the Arctic (Wex et al., 2019; Tobo et al., 2019; Santl-Temkiv et al., 2019). In particular, Wex et al. (2019) observed an $n$INP increase of more than

one order of magnitude from spring to summer (e.g., ~14 times at T=-15°C) at the Gruvebadet (GVB) observatory in 2012. Following two field campaigns at the Mt. Zeppelin station in July 2016 and March 2017, Tobo et al. (2019) reported $n$INPs at -20°C of about 0.01 L$^{-1}$ in spring and about 0.1 L$^{-1}$ in summer. This substantial increase was interpreted as the effect of local INP sources active when land and sea are free from snow and ice (Santl-Temkiv et al., 2019; Wex et al., 2019; Tobo et al.,

2019). Conversely, the results of multi-year (May 2015 - January 2017) INP observations at Mt. Zeppelin (Svalbard) by Schrod

et al. (2020) evidenced no significant seasonal trend in $n$INP.

Our study aims to add to the still scant INP observations in the Arctic environment, investigating $n$INP and potential INP sources during spring and summertime at the ground-level site of GVB. In particular, we extend the INP observations at GVB, previously only 13 samples (Wex et al., 2019), presenting the results of 61 samples investigated with two offline INP measurement techniques. We also analyze the ice nucleation efficiency of Arctic aerosol particles by calculating their activated

fraction (AF). To date, only a limited number of studies provide information on INP trends scaled to the total aerosol concentration over the Arctic (Si et al., 2018). AF estimation can be understood as a simple metric indicating the ice-nucleating efficiency of particles within a specific aerosol sample (Schrod et al., 2020). In our specific case, it provides further insight, over and above the $n$INP data, into INP characteristics over the Atlantic sector of the Arctic.

**2 Methods**

**2.1 Sampling**

Aerosol particle sampling was performed at the GVB observatory located in proximity to the village of Ny-Ålesund (78°55' N, 11°56' E) on Spitsbergen Island, Svalbard (Fig. 1). The observatory is located about 40 m above sea level and about 1 km southwest of the village, a location that guarantees minimal influence by local pollution sources given the prevailing south winds (Udisti et al., 2016). Aerosol particle sampling activities for offline $n$INP analyses were arranged independently for the

two methods (see Sect. 2.2) with different sampling time intervals through different inlets. All inlet heights were set about 5 m above ground level.

Aerosol particle samples for the Dynamic Filter Processing Chamber (DFPC) application, were collected on nitrocellulose membrane filters (Millipore HABG04700, nominal porosity 0.45 μm). We deployed two parallel sampling inlet systems, one with a PM$_1$ cut size, the other for PM$_{10}$ (cut-point-Standard EN 12341, TCR Tecora). Operative flow was 38.3 (±2.0) LPM in

each sampling line was generated by two independent pumps (Bravo H Plus, TCR Tecora). Sampling for DFPC was carried out over two meteorological seasons: from 17 April to 2 May 2018 in spring and from 11 to 27 July 2018 in summer. A pair of samples (duration of 3 to 4 hours) was collected each day from the two inlet systems. Our short sampling span was employed to avoid aerosol particle overloading on the filters. To coordinate with other scheduled activities at GVB, sampling generally started in the morning during the spring campaign and typically in the afternoon during the summer campaign. A total of 33

PM$_1$-PM$_{10}$ pairs of samples were collected, 16 in spring and 17 in summer. Samples were stored at room $T$ until analysis.

For the West Texas Cryogenic Refrigerator Applied to Freezing Test system (WT-CRAFT) analysis, a total of 28 samples were collected from April 16 to August 15, 2018. Aerosol particles were collected using 47 mm membrane filters (Whatman, Track-Etched Membranes, 0.2 μm pore). Aerosol particle-laden air was drawn from a central total suspended particulate (TSP) inlet with a constant average inlet flow of 5.4 LPM (± 0.2 LPM standard deviation). The TSP inlet was custom made and

designed to operate with isokinetic and laminar flow at 150 LPM. From the central inlet, an 8 mm outside diameter stainless

steel tube was directly connected to the filter sampler to intake a subset of airflow. An excess flow in the TSP inlet was drawn through other instruments connected to the inlet or a central inlet pump. More detailed conditions of our filter sampling, including sampling time stamps, air volume sampled through filter cross-section, and the resulting HPLC water volume used to suspend aerosol particles for WT-CRAFT analysis, are summarized in Table S1. Below the filter sampler, the filtered air

was constantly pumped through a diaphragm pump (KnF, IP20-T). A critical orifice was installed upstream of the pump to ensure a constant volume flow rate and control the mass flow rate through the sampling line. A typical sampling interval was approximately 4 days, with only one exception (i.e., 8 days for the sample collected starting 26 May 2018).

## 2.2 Ice Nucleation Measurements

### 2.2.1 DFPC

All DFPC measurements were carried out in the lab on completion of the campaigns using the membrane filter technique presented in Santachiara et al. (2010) and Rinaldi et al. (2017). All measurements were completed within ca. 6 months from sampling. A replica of the Langer dynamic filter processing chamber (Langer and Rodgers, 1975) housed in a refrigerator was used to determine $n$INP at different $T$s. Before the analysis, each filter was placed on a metal plate (5.5 cm diameter, 0.5 mm thick) previously coated with a smooth paraffin layer to ensure good thermal contact between filter and supporting substrate.

The paraffin was then flash heated at 70°C for less than 5 seconds and rapidly cooled in order to fill the filter pores. Particle-free air entered the DFPC chamber through a perforated plate, spreading to an ice bed to become saturated with respect to ice but undersaturated with respect to water. The air then proceded to the filter, cooled by a Peltier device in contact with the supporting metal plate. Only at this point, did the air become supersaturated with respect to water. By controlling the $T$s of the filter and surrounding air, the samples could be exposed to different $T$s while keeping the water saturation ratio ($S_w$) above 1.

The supersaturation ratio was calculated theoretically from vapour pressures of ice and water at the $T$s considered (Buck, 1981). More details of the DFPC working principle can be found in the supplement of deMott et al. (2018).

Ice nucleation was visually evaluated by counting the number of ice crystals growing on individual aerosol particles on the sampled filter illuminated by a visible light source. Measurements were performed at activation $T$s of -15°C, −18°C and −22°C and at $S_w$ = 1.02. Uncertainties for air $T$, filter $T$ and $S_w$ are about 0.2°C, 0.1°C and 0.02, respectively. The overall uncertainty

in the DFPC-based INP assessment was estimated by considering the effect of $S_w$ variations on $n$INP extrapolated from the laboratory results by Belosi et al. (2018) and found to be around ±30%. The filter background INP contribution was evaluated by analyzing blank filters at the same evaluation conditions as the samples. Measurements were corrected for the filter background.

### 2.2.2 WT-CRAFT

To complement the DFPC results, we also used the WT-CRAFT offline droplet-freezing assay instrument to measure $T$-resolved $n$INP at $T$ > -25°C, with a detection capability of >1 INP per m$^3$ of air. All measurements were completed within one

year from the collection of the samples, and the samples were stored in a fridge (4°C) before analysis. Although WT-CRAFT was originally a replica of NIPR-CRAFT (Tobo, 2016), the two CRAFT systems exhibit different sensitivities to artifact and detectable $T$ ranges as described by Hiranuma et al. (2019) and Vepuri et al. (2021). The reason for the different detectable $T$
ranges is not known. Hiranuma et al. (2019, i.e., Table S2) report the uncertainties of $T$ and ice nucleation efficiency in WT-CRAFT as ± 0.5°C and ±23.5%, respectively. The $T$ uncertainty reported was derived based on an observed systematic difference between the thermal sensor (TGK, SN-170N) measurements and setpoint $T$s of a cryo-cooling system (Scinics Corporation, Model CS-80CP) at > -25 °C. The uncertainty of ice nucleation efficiency estimation in WT-CRAFT was previously estimated based on the average standard deviation at $T$s > -25°C for a known composition (microcrystalline
cellulose). Alternatively, as demonstrated in Vepuri et al. (2021) and Schiebel (2017), our experimental uncertainty in estimated $n$INP can be evaluated using the 95 % confidence interval method.

While there are no major systematic differences between the two CRAFT instruments, WT-CRAFT employs different image recording systems, stage and clean housing compared to NIPR-CRAFT. For imaging, WT-CRAFT uses a combination of an Opti-Tekscope OT-M HDMI microscope camera and a Logitech c270 camera. This combination is used to correctly capture
the transition of droplet brightness/contrast to opaque ice with 30 frame-per-second time resolution, with reasonable pixel resolution and magnification (if needed). As for a cold stage, we used a thin (<5 mm) polished aluminium plate to ensure efficient thermal cooling and that the cryo-cooling system $T$ was equivalent, within known uncertainties, to the $T$ measured at the plate surface. Finally, WT-CRAFT was operated in a vertical clean bench (LABCONCO, Purifier®). All droplet preparations were conducted in the clean bench to minimize possible contamination with the lab air.
For each experiment, 70 solution droplets (3 µL each) placed on a hydrophobic Vaseline layer were analyzed. At a cooling rate of 1°C min$^{-1}$, we manually counted a cumulative number of frozen droplets based on the color contrast shift in the off-the-shelf video recording camera. $n$INP of 3µL-sized droplets containing aerosol particles from the samples were then estimated as a function of $T$ for every 0.5°C (Sect. 2.2.3).

Prior to each WT-CRAFT experiment, we suspended particles on an individual filter sample in a known volume of ultrapure
High Performance Liquid Chromatography (HPLC) grade water, in which the first frozen droplet corresponded to ≈ 1 INP m$^{-3}$ (in the range of 0.93 – 1.02 m$^{-3}$; Table S1). The HPLC water volume was determined according to Eqns. 1-2 in Sect. 2.2.3. Half of each filter was used for each WT-CRAFT experiment, the other half was saved for other and future uses. Our suspension-generating protocol entailed: (1) cutting the filter in two and soaking one filter half in ultrapure water in a sterilized falcon tube; (2) vortex-mixing the suspension tube to scrub particles on the filter in suspension; (3) applying an idle time of 5
min to have the quasi-steady state suspension and (4) preparation of droplets out of the suspension through micro-pipetting in the clean bench. If necessary, the suspension sample was diluted until we observe their freezing spectrum collapsing onto the water background curve. Our diluted spectra and original freezing spectrum reasonably agreed in their overlapped $T$ region (within a factor of three at most) without any notable artifacts at $T$ above -25°C. Having no failures, we simply stitched all spectra so that the data point with the best (smallest) 95% confidence interval represented $n_{INP}(T)$ for the overlapping $T$ region

if observed. Due to negligible background contribution of water freezing at -25 °C (i.e., < 3%), we did not apply any background corrections on our $n$INP data.

### 2.2.3 Derivation of INP atmospheric concentrations

For DFPC samples, $n$INP, expressed hereafter in units of $m^{-3}$, was calculated by dividing the number of ice crystals quantified on each filter by the sampled volume of air passed through the filter. For the WT-CRAFT analysis, we first computed the
$C_{INP}(T)$ value, which is the freezing nucleus concentration in HPLC suspension ($L^{-1}$ water) at a given $T$ as described by Vali (1971). This $C_{INP}(T)$ value was calculated as a function of the unfrozen fraction, $f_{unfrozen}(T)$ (i.e., the ratio of the number of droplets unfrozen to the total number of droplets) as:

$$C_{INP}(T) = -\frac{\ln(f_{\text{unfrozen}}(T))}{V_d} \qquad (1)$$

where, $V_d$ is the volume of individual droplets ($3\,\mu$L). Next, we converted $C_{INP}(T)$ to $n$INP$(T)$. The cumulative $n$INP per unit
volume of sample air, described in deMott et al. (2017), was estimated as:

$$nINP(T) = C_{INP}(T) \times DF \times \frac{V_l}{V_{air}} \qquad (2)$$

where DF is a serial dilution factor (e.g., DF = 1 or 10 or 100 and so on). The sampled air volume ($V_{air}$) and the suspension volume ($V_l$) are provided in Table S1.

To estimate the efficiency of the sampled aerosol particles to nucleate ice, we calculated AF by scaling $n$INP to the aerosol
particle number concentration measured onsite in parallel (see 2.3.1). For the AF estimation we considered aerosol particles in the 0.1-10 µm size range were considered since this size range is reasonably accountable for heterogeneous ice nucleation in the atmosphere (Kanji et al., 2017).

### 2.3 Complementary Analyses

### 2.3.1 Particle size distribution measurements

The aerosol particle number size distribution was continuously monitored at the GVB station using a Scanning Mobility Particle Sizer (SMPS), model TSI 3034, for the diameter range between 10 and 500 nm (54 channels). An Aerodynamic Particle Sizer (APS), model TSI 3321, was used for measuring the aerodynamic aerosol particle diameters between 0.5 and 20 micrometers in parallel with the SMPS. Both instruments were connected to a common stack inlet, where the WT-CRAFT filter sampler was deployed, and recorded data averaged over 10 minutes (Giardi et al., 2016; Lupi et al., 2016). The
aerodynamic diameters measured by the APS were corrected to the volume equivalent diameters using an average particle mass density of 1.95 g $cm^{-3}$, assuming a mixture of different substances based on the findings of Lisok et al. (2016), and a dynamic shape factor of 1. The number concentration in the resulting overlapping range was taken from the SMPS data. Finally, commutative aerosol particle counts of SMPS and APS were considered as a total aerosol particle number concertation. In order to compare with $n$INP and to calculate the AF, the particle number concentrations at 10 minutes time resolution were

averaged over each filter sampling period. AF was calculated using the size range 0.1 – 10 µm for $DFPC_{PM10}$ and WT-CRAFT data, 0.1 – 1 µm for $DFPC_{PM1}$ data and 1 – 10µm for DFPC data in the super-micrometre regime. The SMPS-APS unerwent maintenance during August 2018 and was therefore not operational.

### 2.3.2 Meteorology

Meteorological parameters ($T$; pressure, P; RH; wind speed, WS) were provided by the Amundsen-Nobile Climate Change Tower positioned less than 1 km N-E of GVB (Mazzola et al., 2016), while precipitation data (type and amount) were taken from the eKlima database, provided by the Norwegian Meteorological Institute (https://seklima.met.no/observations/).

### 2.3.3 Offline Chemical Analysis

The chemical analysis of major and trace ion species, used in this work as aerosol particle source tracers, was accomplished on Teflon filters (PALL Gelman) collected at GVB with a TECORA Skypost sequential sampler equipped with a $PM_{10}$ sampling head at an operating flow rate of 2.3 m$^3$ h$^{-1}$ (EN 12341) for 24 h. Throughout the sampling and offline analysis at the University of Florence, the filters were handled with care by personnel working under a class 100 laminar flow hood and wearing powder-free latex gloves to minimize potential contamination. After sampling, the filters were shipped at -20°C and stored at the same $T$. Measurements were carried out by a triple Dionex ThermoFisher Ion Chromatography system equipped with electrochemical-suppressed conductivity detectors. In particular, a Dionex AS4A-4 mm analytical column with a 1.8 mM $Na_2CO_3$/1.7 mM $NaHCO_3$ eluent, was used to determine most of the inorganic anions ($Cl^-$, $NO_3^-$, $SO_4^{-2}$, $C_2O_4^{-2}$) while a Dionex AS11 separation column with gradient elution (0.075–2.5 mM $Na_2B_4O_7$ eluent) was used to measure $F^-$ and some organic anions (acetate, glycolate, formate and methanesulfonate). Cationic species ($Na^+$, $NH_4^+$, $K^+$, $Mg^{2+}$, $Ca^{2+}$) were determined by a Dionex CS12A-4 mm analytical column with 20 mM $H_2SO_4$ as eluent. Further analytical details can be found in Udisti et al. (2016) and Becagli et al. (2011).

### 2.3.4 Back trajectories and satellite ground type maps

To investigate the sources that contributed to INPs (i.e., maritime vs. terrestrial), we performed 5-day back trajectory analysis according to Wex et al. (2019). 5-day back-trajectory air masses (HYSPLIT4 with GDAS data: https://ready.arl.noaa.gov/) from the National Oceanic and Atmospheric Administration (NOAA) HYSPLIT model (Rolph et al., 2017; Stein et al., 2015) were simulated for an altitude of 100 m above mean sea level (amsl) over the GVB station. For DFPC samples, back-trajectory arrival time was considered simultaneous to INP samples, while for WT-CRAFT, the trajectories were calculated twice a day (at 06 and 18 UTC) for the INP sampling period from April to August. Only back-trajectories traveling up to an altitude of 500 m amsl were considered for this analysis, which is a reasonable assumption for air masses passing within the marine boundary layer (Dai et al., 2011).

Ground condition maps were obtained from the National Ice Center's Interactive Multisensor Snow and Ice Mapping System (IMS) (Helfrich et al., 2007; National Ice-Center, 2008), National Snow & Ice Data center (NISDC; https://nsidc.org/). We

used the daily Northern Hemisphere maps with a resolution of 4 km. The ground types considered were "seawater", "sea-ice"," "land", and "snow". Seawater indicates passage of the air mass over open seawaters, while sea-ice indicates passage over ice-covered seawaters. "Land" and "snow" categories indicate the passage of air mass over land without and with snow cover, respectively. For each back-trajectory end-point, we applied nearest-neighbour interpolation in space and time to find the closest pixels on the satellite map and associated the end-point with the corresponding ground type. Combining the information obtained along the whole back-trajectory (or group of back-trajectories for WT-CRAFT samples) allowed estimation of the contribution of each ground type to each INP sample.

### 2.3.5 Satellite chlorophyll-a data and correlation analysis

Satellite retrieved chlorophyll-a fields were used to track the evolution of oceanic biological activity in the Arctic ocean during the study period. The best estimate "Cloud Free" (Level-4) daily sea surface chlorophyll-a concentration (CHL; mg m$^{-3}$) data were downloaded from the EU Copernicus Marine Environment Monitoring Service (CMEMS; http://marine.copernicus.eu/) based on a multi-sensor approach (i.e., SeaWiFS, MODIS-Aqua, MERIS, VIIRS and OLCI-S3A). The Level-4 product is available globally at ~4 km spatial and daily time resolution. From this global dataset, CHL fields were extracted in the Arctic Ocean during summer 2018 to be merged with INP data.

Recent literature (Wilson et al., 2015; Knopf et al., 2011; Wang et al., 2015) has shown that sea-spray organic aerosols can act as INPs in the clean marine atmosphere. Mansour et al. (2020b) evidenced that $n$INP over the North Atlantic Ocean follows the patterns of marine biological activity as traced by surface CHL concentration. The relationship between INPs and phytoplankton biomass, traced by CHL concentration, was investigated. Samples clearly influenced by land inputs were excluded so as to focus only on INPs potentially originating from the sea. The DFPC dataset was chosen for this analysis since it provides a higher time-resolution than WT-CRAFT one and allows differention between fine and coarse INPs. Assuming that each set of daily DFPC samples represents a day, enabled us to compare to the daily CHL time series. The Pearson correlation coefficients between INPs and satellite-derived ocean colour data, obtained by standard least squares regression, were computed at each grid point of the Arctic domain for different time-lags to obtain the correlation maps presented in the Results section.

### 2.3.6 Concentration weighted trajectory

The allocation of regional source areas potentially affecting $n$INPs sampled at GVB was achieved by applying the concentration weighted trajectory (CWT) model (Hsu et al., 2003; Jeong et al., 2011). In this procedure, each grid cell within the studied domain is associated with a weighted concentration, which is a measure of the source strength of a grid cell with respect to concentrations observed at the sampling site. The average weighted concentration in the grid cell $(i, j)$ is determined by Eq. (3).

$$CWT_{ij} = \frac{\sum_{t=1}^{L} n\text{INP}_t \, D_{ijt}}{\sum_{t=1}^{L} D_{ijt}} \times W_{ij} \tag{3}$$

where $t$ is the index of the trajectory, $L$ is the total number of trajectories (5 days – hourly time step), $n\text{INP}_t$ is the concentration observed at sampling location (receptor site) on arrival of trajectory $t$, and $D_{ijt}$ is the residence time (time spent) of trajectory $t$ in the grid cell $(i, j)$. Given $n\text{INP}_t$, $D_{ijt}$ can be determined by counting the number of hourly trajectory segment endpoints in each grid cell for each trajectory. This was repeated for all the back trajectories $L$. A high value for $CWT_{ij}$ means that air parcels travelling over the grid cell $(i, j)$ would on average be associated with elevated concentrations at the receptor site.

In this study, five-day low (< 500 m) air mass back-trajectories were used to produce the CWT spatial distribution corresponding to the DFPC summer campaign. The DFPC summer dataset was selected for consistency with the correlation analysis presented in the previous section. Two trajectories per day were associated with the corresponding $n$INP of the day. Similar to the correlation analysis, the INP samples clearly influenced by passage over land were excluded to focus on marine sources. The selected domain extends up to the limits of the area covered by the above-mentioned low back-trajectories (48° – 85° N & 75° W – 42° E), and was divided into 1°×3° latitude/longitude grid cells (1443 cells, 308 cells with at least one trajectory endpoint). To reduce the impact of grid cells containing a low number of endpoints, making CWT calculation statistically less robut, the CWT values were multiplied by a weighting factor ($W_{ij}$) according to Eq. (4).

$$W_{ij} = 1 \ (\text{if } D_{ij} \geq median), \ W_{ij} = 0.8 \ (\text{if } 3 < D_{ij} < median), \text{ and } W_{ij} = 0 \ (\text{if } D_{ij} \leq 3) \tag{4}$$

The introduction of the weighing factor reduces the number of cells considered to 203.

## 2.4. Statistical data treatment

In this study, linear relationships between measured ambient variables were tested by the Pearson correlation method. Regressions with correlation coefficients ® overpassing the critical threshold for a confidence interval of 95% (p<0.05) were considered statistically significant. The Tables detailing the correlation analyses also report non-statistically significant R values but exclude the R values with a confidence interval lower than 80%. The results of the correlation analyses are presented along with the confidence level (p<) and the number of data points (n). The statistical significance of differences between datasets was tested using the standard t-test. Since a subset of our measured data was not normally distributed, we complemented each t-test by also performing the non-parametric Wilkoxon-Mann-Whitney test, which does not require normally distributed data. Outcomes were considered statistically significant only if the two tests provided consistent results.

## 3 Results

### 3.1 INP atmospheric concentration

Figure 2 shows the overall $n$INP range measured for aerosol particle samples from the GBV station in spring-summer 2018. While both datasets show reasonable agreement for $n$INP at -22°C, a notable offset was observed between the two techniques in other activation temperatures. Specifically, $n$INP measured in condensation mode (DFPC) was generally higher than that

measured in immersion mode (WT-CRAFT), the deviation becoming even more apparent at higher $T$. On average, $n\mathrm{INP_{DFPC}}$ was 3 times higher than $n\mathrm{INP_{WT\text{-}CRAFT}}$ at -22°C and 8 times higher at -15°C. Thus, the WT-CRAFT ice nucleation spectra presented a steeper $\Delta n\mathrm{INP}/\Delta T$ slope than DFPC.

DFPC-measured $n\mathrm{INP}$ values ($\mathrm{PM_{10}}$ size range) from GVB during the spring campaign ranged 55-185 (median 115), 5-90 (53)
and 3-37 (20) m$^{-3}$, for $T$ of -22, -18 and -15°C, respectively. During the summer campaign, $n\mathrm{INP}$ ranges were 33-135 (median 77), 18-107 (45) and 6-66 (20) m$^{-3}$, for the same $T$s (Fig. 2). The WT-CRAFT analysis found no ice nucleation activity above -9°C in our GVB samples. In the $T$ range between -9 and -14°C, a subset of the samples (1 sample, at T of -9°C, and 13, at $T$ of -13.5°C, over 28 total samples) presented $n\mathrm{INP}$ above the detection limit (> 3 m$^{-3}$). In the rest of the $T$ spectrum, $n\mathrm{INP}$ ranged 1-3 (median 2) m$^{-3}$ at -14°C and 24-9082 (166) m$^{-3}$ at -25°C.

A compilation of $n\mathrm{INP}$ values from previous ground-level observations at various Arctic stations can be found in Table 1. The range of $n\mathrm{INP}$ from Table 1 is roughly between $10^{-2}$ and $10^{3}$ m$^{-3}$ in the $T$ range between -9 and -25°C, in which we detected ice nucleation activity in our samples. This $n\mathrm{INP}$ range covers the majority of our measurements. It should be noted that comparison with these past studies is only qualitative given the great variability of parameters that may influence $n\mathrm{INP}$ (e.g., different instruments, locations, season, weather conditions, aerosol particle size distribution, ice nucleation mode, etc.).
Nonetheless, both the DFPC and WT-CRAFT datasets overlap fairly consistently with the $n\mathrm{INP}$ results reported by Wex et al. (2019), for samples taken at the same station in 2012 (Fig. 2). While the comparison between our datasets and those of Wex et al. (2019) is also qualitative, since the two studies examined different aerosol particles collected in different years, we found several interesting agreements and discrepancies. First, while Wex et al. (2019) report a very narrow concentration range (27-33 m$^{-3}$) at -22°C, having only three samples, our 61 DFPC plus WT-CRAFT data points span a much wider range (ca. 3-200
m$^{-3}$). The upper limit of observable $n\mathrm{INP}$ in Wex et al. (2019) was roughly 40 m$^{-3}$, depending on the volume of air sampled on the filters analysed. Conversely, the data ranges are in good agreement for $T$s over -18 to -15°C. Finally, the data from Wex et al. (2019) span a wider range (ca. $10^{-1}$ – 10 m$^{-3}$) than those of WT-CRAFT (1-3 m$^{-3}$) for $T$> -15°C. The difference in $n\mathrm{INP}$ towards a lower bound is due to different sensitivities and detection limits of the two methods: WT-CRAFT (ca. 1 m$^{-3}$) and the immersion freezing measurement technique used by Wex et al. (2019, ca. $10^{-1}$ m$^{-3}$).

**3.2 Contribution of fine and coarse INPs**

The sampling strategy applied for DFPC measurements allowed us to investigate fine (< 1 µm) and coarse (>1 µm) INPs. Table 2 reports the number concentrations of INPs measured in the two different size ranges, together with the average contribution of super-micrometer (coarse) INPs, derived by difference. The scatterplots of $n\mathrm{INP_{PM10}}$ vs $n\mathrm{INP_{PM1}}$ are given in Fig. S1. Our spring campaign data are characterized by scant coarse INPs (~20%), suggesting that most INPs may be fine
mode aerosol particles transported for long distances, with consequent depletion of the largest particles during transport due to their higher gravitational deposition velocities (Shaw, 1995; Heidam et al., 1999; Stohl, 2006). During the summer campaign, a statistically significant increase in the contribution of coarse INPs was observed (i.e., 65% at $T$ = -15°C; p<0.05), potentially resulting from the contribution of locally emitted aerosol particles (see Sect. 3.6), in part from the surface exposed

to the air once snow and ice had melted. The same trend is inferred by the particle size distribution measurements, which show a significant enhancement of coarse particles contribution in summer compared to springtime (Fig. S2; $p<0.05$, $n_1>10^3$; $n_2>10^3$). The increase of the coarse INP contribution, from spring to summertime, is progressively more pronounced with increasing activation $T$. An INP population similar to summer values at GVB, with a significant coarse fraction contribution, was reported by Mason et al. (2016), from the Alert Arctic station. The authors conducted INP measurements from 29 March to 23 July 2014 and observed an increasing contribution of coarse INPs as a function of the activation $T$. It is important to note that our results are unique compared to past studies as our measurements and data support the increase of coarse INP contribution during the meteorological season transition from spring to summer with increasing activation $T$.

### 3.3. Aerosol particle Activated Fraction (AF)

Figure 3 presents aerosol particles AF as a function of $T$ derived from DFPC and WT-CRAFT data. For DFPC ($PM_{10}$ size range), AF ranged from ~$5\times10^{-8}$ to $1\times10^{-5}$ in the $T$ range examined. Likewise, the AF range of WT-CRAFT was from ~$1\times10^{-8}$ (minimum value observed at $T$ of -10°C) to ~$1\times10^{-5}$ (maximum value observed at $T$ of -25°C). The seasonal evolution of the aerosol particles AF is discussed in detail in the following Section (Sect. 3.4).

Examining the size-segregated DFPC data (Fig. 3a and b), substantially higher ice nucleation efficiencies were found in coarse compared to sub-micrometer particles. The enhanced AF of coarse particles is due to the significantly lower number of particles in the 1-10 µm compared to the 0.1-1 µm size range (about two orders of magnitude), coupled with the comparable $n$INP observed in both size ranges (see Sect. 3.2). As a result, the AF of coarse particles was more than 2 orders of magnitude greater than that of fine particles. In other words, the AF for coarse particles was estimated to be in the order of $10^{-6}$ to $10^{-3}$ at $Ts$ between -18 and -22°C, while the AF of sub-micrometer particles was in the order of $10^{-8}$ to $10^{-5}$ at the same $Ts$.

Si et al. (2018) and Creamean et al. (2018) also reported a higher ice nucleation efficiency for super-micrometer particles sampled at Arctic stations compared to the sub-micrometer range. Their papers cover the INP data collected in both summertime (Si et al., 2018) and springtime (Creamean et al., 2018). In particular, Si et al. (2018) reported average AF at -25°C of ~$10^{-4}$, $2\times10^{-3}$ and $6\times10^{-2}$ for the 0.56-1.0, 3.2-5.6 and 5.6-10 µm size ranges, respectively.

### 3.4 Seasonal variation in $n$INP and AF

The time series data in Fig. 4 do not indicate a clear seasonal increase in ambient $n$INP from spring to summer. For the DFPC data ($PM_{10}$), a statistically significant $n$INP reduction (by a factor of 1.5) was found at $T$ of -22°C, in the transition from the spring to the summer campaign ($p<0.05$; $n_1=16$, $n_2=17$), while no significant difference was observed for $Ts$ of -15 and -18°C. The $n$INP time series measured by WT-CRAFT agrees with that of DFPC if we consider only the periods in which the two sampling activities were run in parallel: a statistically significant reduction by a factor of 1.6 is observed at -22°C ($p<0.05$; $n_1=4$, $n_2=5$). On the other hand, considering the whole WT-CRAFT data, a statistically significant $n$INP increase in summer compared to spring was observed only for $Ts$ between -18 and -21°C ($p<0.05$; $n_1=11$, $n_2=17$). However, even in these cases,

the $n$INP seasonal enhancement ratios (i.e., $n$INP$_{summer}$/$n$INP$_{spring}$) were limited to a factor of three at the most. Moreover, the seasonal variations observed were smaller than the range of $n$INP data deviations for each season. Thus, the seasonal variation in $n$INP from the WT-CRAFT analysis is not conclusive and should be cautiously interpreted.

A peak in $n$INP was observed by WT-CRAFT during June, at $T$s lower than $T$ = -18°C (Figs. 4a and 4b). Of the 7 samples collected in June, more than 50% (57-71%) were higher than the whole campaign median at this $T$ range. In addition, the average $n$INP during June was up to ~3 ($T$ = -20°C) times higher than the average for the rest of the observation period. As can be seen in Figs. 4a and 4b, the second peak of $n$INP was observed at the end of the WT-CRAFT measurement period, the last sample representing the highest concentrations across the $T$s measured. Further discussion of the $n$INP-AF relationship during these specific periods is provided below.

Recent studies have reported a marked seasonal trend for $n$INP in the Arctic environment, with atmospheric loadings increasing from spring to summertime (Santl-Temkiv et al., 2019; Wex et al., 2019; Tobo et al., 2019), as detailed in the Introduction. A comparison between the seasonal trends observed in this study and those reported by Wex et al. (2019) at the same sampling location can be found in Fig. S3. The comparison supports the lower magnitude of the spring to summer $n$INP increase observed in 2018 (this study) compared to the 2012 observations (Wex et al., 2019).

Both the DFPC and WT-CRAFT datasets showed a general increase of the aerosol particle AF from spring to summer as shown in Fig. 5. This increase in the AF is mainly due to a significant reduction of the particle number concentration in the 0.1-10 µm range (p<0.05; $n_1$>10$^3$; $n_2$>10$^3$; Fig. S4), combined with similar or slightly higher $n$INP (depending on the $T$). DFPC showed a statistically significant AF increase (p<0.05; $n_1$=16, $n_2$=17) going from the spring campaign to the summer period for all the probed activation $T$s. The seasonal increase in the AF was more evident at higher $T$s: the summer to spring mean ratio was 6.2 at $T$ of -15°C and 2.5 at $T$ of -22°C. Fairly consistent results can be observed in the WT-CRAFT dataset. Comparing the samples collected before June 3 with those collected after that date, an AF enhancement (from 1.1 to 3.7 fold) can be estimated for all the activation $T$s. This difference was statistically significant (p<0.05, $n_1$=11, $n_2$=15) for all the activation $T$s between -17 and -22.5°C. Unlike the DFPC data, the spring to summer AF increase from WT-CRAFT data peaked at $T$ = -20°C (3.7; Fig. S5).

The AF time series by WT-CRAFT reported in Fig. 5 reflects the increase in $n$INP for the month of June, as described above. This demonstrates that the $n$INP enhancement observed in June is due, at least in part, to enhanced ice nucleation activity of the particle population (more INP per particle number), rather than only to an increase of aerosol particle concentration. We note that AF data with WT-CRAFT are not available for August as the SMPS was down for maintenance. Thus, whether the increase of $n$INP detected by WT-CRAFT in August (i.e., the last two data points in Figs. 4a and 4b) corresponds to the enhancement of ice nucleation efficiency or an increase in the overall aerosol particle concentration remains uncertain.

**3.5 Relation of $n$INP to meteorological parameters and particle number concentration**

No clear relationship was found between $n$INP and the major meteorological parameters ($T$, pressure, RH and WS). The only exception was precipitation, which was often associated with a reduction of $n$INP (Fig. S6). Although $n$INP tends to covariate

during the spring campaign with particle number concentration in the range 0.5-10 µm, considered for consistency with deMott et al. (2010) (Pearson's R of 0.18, for $INP_{PM10}$ at T= -22°C, and 0.22, for $INP_{PM10}$ at $T$ = -18°C), a significant correlation (R = 0.56; p< 0.05, n=16) was observed only for $INP_{PM10}$ at $T$ = -15°C, in the DFPC dataset from this season. During summer, no correlation at all was observed between $n$INP and particle number (R between -0.13 and -0.25). For WT-CRAFT, significant correlations were observed only for T<-23°C (p<0.05, n=28; R between 0.42 and 0.52). It is, however, important to note that previous studies from different regions report discrepant correlations between INP and particle number concentration. For example, a correlation is often reported with the number concentration of aerosol particles larger than 0.5 µm (deMott et al., 2010; deMott et al., 2015; Mason et al., 2015; Schwikowski et al., 1995). In other cases, no correlation whatsoever was found (Richardson et al., 2007; Rogers et al., 1998), which is not surprising considering that INPs are only a small fraction of total particles. Bigg (1996) reported a good correlation between $n$INP and accumulation mode particles, for one day of measurements over the high Arctic, while a modest but significant correlation (R = 0.25 – 0.30) between $n$INP and particle number concentration in the 50-120 nm range was reported by Bigg and Leck (2001), close to the North Pole. To the best of our knowledge, no other paper has addressed this relation in the Arctic environment.

### 3.6. INP sources in the Arctic

### 3.6.1 Correlation with chemical tracers

To investigate the potential sources of the INPs at GVB, a correlation analysis was performed between both $n$INP datasets and the atmospheric concentration of chemical tracers routinely measured at the station. During the spring campaign, $n$INP correlated with tracers of long-range transported anthropogenic aerosol particles such as nitrates, non-sea-salt-sulfate and non-sea-salt-potassium (Table 3; Fig. S7-S10). Indeed, Udisti et al. (2016) associated springtime non-sea-salt-sulfate at GVB with long-range transported anthropogenic sources. The authors also showed that the production of biogenic non-sea-salt-sulfate from the sea is relevant only in summertime. The springtime peak of anthropogenic aerosol transport from lower latitudes is often referred to as the Arctic haze (Shaw, 1995). A general tendency to anticorrelation with sodium and chlorine was also observed in both the size classes, though only $PM_1$ was statistically significant (p<0.05). The only significant relations observed for the summer DFPC data were at $T$ = -15°C: an anticorrelation was observed between $n$INP$_{PM10}$ and particulate mass, sea spray tracers (sodium and chlorine) plus calcium, magnesium and lithium.

No clear source indications were derived from the correlation analysis of the WT-CRAFT data to the chemical tracers. However, analysis of seasonally categorized $n$INP$_{WT-CRAFT}$ (June 3 being the demarcation date between spring and summer) provided a similar result to the DFPC data (Table 4). In short, in the spring season, $n$INP$_{WT-CRAFT}$ correlated with tracers of anthropogenic long-range transported aerosols (non-sea-salt-sulfate, nitrate, non-sea-salt-potassium), particularly at low activation $T$s. Additionally, calcium concentration exhibited some tendency to correlate with $n$INP (both datasets). Our analysis was unable to determine whether this was from natural dust or other anthropogenic sources and it is not conclusive if it has any impact on $n$INP. In the summer season, no significant correlation was observed. It should be noted, however, that our

tracer analysis only infers the aerosol properties, with the result that further analysis of INP identities and properties (e.g., ice crystal residual analysis) would be necessary to reveal the source of INPs.

### 3.6.2 Influence of ground conditions

The influence of ground conditions (sea-ice, snow, seawater and land) on the low-travelling back-trajectories examined (<500m) was evaluated by merging back trajectories and satellite ground type data (Wex et al., 2019). Figure 6 shows that the contribution of the four ground types considered varies with the seasons. In spring, the majority of contacts occurred with sea-ice or snow-covered land, while in summer low air masses were more influenced by ice-free seawaters. The (snow-free) land contribution was the lowest in every season. Nevertheless, the influence of land sources on $n$INP emerges clearly from Table 5 and Fig. S11: air masses with a higher terrestrial influence have always been associated with $n$INP peaks. This is probably due to the higher ice nucleation efficiency of mineral dust and soil particles compared to marine biological particles (Wilson et al., 2015; McCluskey et al., 2018a; McCluskey et al., 2018b). In summer, contacts with snow-free land occurred mainly over the Svalbard archipelago (local sources) or over Greenland and Iceland (regional sources), as shown in Fig. 6.

### 3.6.3 Contribution of marine biological INP sources

Considering that the sampled air masses had ground contacts mainly over seawater during summer, one may summize that marine biological sources dominate $n$INP at GVB in this season, outside the occasional periods of elevated terrestrial influence. To check this hypothesis, we investigated the spatio-temporal correlation of the INP datasets with satellite retrieved surface CHL, used as a tracer for marine biological activity, following the time-lag approach first introduced by Rinaldi et al. (2013). The DFPC dataset was selected for this analysis because it provides a higher time resolution than the WT-CRAFT data and, most of all, because it allows a distinction between fine and coarse INPs. In fact, both McCluskey et al. (2018b) and Mansour et al. (2020b) have shown that fine INPs tend to correlate better with CHL in clean marine air masses. To exclude interferences from land sources, we screened the samples corresponding to back trajectories that had been in contact with land for more than 10% of the time (3 samples) from the entire dataset. Furthermore, we focused on INP PM$_1$ data obtained at $T$ of -15°C, which were expected to be the most representative of ice nucleation by biological particles and less influenced by mineral particles (Kanji et al., 2017).

The results of the correlation analysis are reported in Fig. 7, in the form of correlation maps. Here, the colour of each pixel represents the correlation coefficient (R) resulting from the linear regression between the CHL concentration in that pixel and $n$INP$_{PM1}$ measured at GVB. Different maps were obtained by considering different time-lags between the two correlated time series, i.e., by considering CHL concentration values shifted back 1 to 27 days with respect to the INP filter sampling times (the maps are shown in Fig. S12). The time-lag approach has been demonstrated to maximize the correlation between in situ coastal measurements of aerosol properties and CHL concentration fields (Rinaldi et al., 2013; Mansour et al., 2020b; Mansour et al., 2020a); it reflects the time scale of the biochemical processes responsible for the production of transferable organic matter in the seawater after the phytoplankton growing phase tracked by CHL patterns. Sea regions characterized by high

correlation (red dots in the maps) are potentially related to the emission of biological particles acting as INPs in our samples. Figure 7 reports three examples of correlation maps, with time-lags of 6, 14 and 16 days. The maps in Fig. 7 were selected as they clearly show high correlation regions in the seawaters surrounding the Svalbard archipelago (time-lag 6 days), close to the Greenland coast (time-lag 14 and 16 days) and to the northeast of Iceland (time-lag 16 days). These regions were all consistently located upwind of GVB during the sampling period (Fig.6). All the maps obtained are available in the Supplementary Material, including those obtained with $PM_{10}$ INP data, which as expected, do not evidence any significant correlation with CHL (Fig. S13). Considerations on the robustness of the correlation maps can be found in the Supplementary Material (Sect. S1 and Figs. S14-S16). In our interpretation, the lack of a correlation between surface CHL concentration and coarse INPs does exclude the potential of the ocean surface to be a source of super-micrometer INPs. Rather, it simply evidences that CHL is not the appropriate proxy to track the emission of large biological INPs from the oceans. Indeed, while CHL has previously been observed to correlate with the enrichment of organic matter in sub-micrometre sea spray (Rinaldi et al., 2013; O'Dowd et al., 2015), no investigation has ever been attempted with super-micrometer particles. In a laboratory-controlled setting, McCluskey et al. (2017) evidenced the production of both sub- and super-micrometer INPs (active at -22°C) from controlled algal blooms, pointing out that different particle types and production mechanisms are involved. However, the possible relationship and time lag between chlorophyll production and sea spray aerosols generation in the atmosphere, and subsequent ambient INP identification from the chlorophyll source are still under debate since the question involves complex processes over an ocean-atmospheric interface on a wide spatiotemporal scale (Crocker et al., 2020; Wolf et al., 2020; Mansour et al., 2020b).

Since our correlations alone could not ascertain a cause-effect relation, we also ran the CWT spatial source attribution model on the same INP dataset (DFPC; $PM_1$; $T = -15°$ C; no land influenced samples). The resulting map (Fig. 8a), composed of 203 cells over the selected domain, evidences that potential sub- micrometre INP sources at GVB during the summer period were broadly located in the same sea regions previously highlighted by the spatio-temporal correlation with CHL. To facilitate the comparison between spatio-temporal correlation maps and the CWT results, we evidenced every pixel with both a CWT value above the median and a significant positive correlation between $nINP_{PM1}$ and surface CHL, considering every delay time between 5 and 20 days (Fig. 8b). The analysis evidenced the sea regions close to Svalbard and immediately east of Greenland. Our results suggest that they they may have been involved in the emission of biogenic INPs sampled at GVB, outside the major episodes of terrestrial influence. The combined analysis also suggests that the region in the northeast of Iceland may also be a potential INP source area even though the spatial distribution of the pixels evidenced is more scattered and, therefore, less convincing.

## 4. Discussion

### 4.1. Interpretation of the discrepancy between $n\text{INP}_{\text{DFPC}}$ and $n\text{INP}_{\text{WT-CRAFT}}$

Several factors may be responsible for the discrepancy observed between the DFPC and WT-CRAFT data. These factors include (1) measurement uncertainties, (2) sampling apparatus, (3) sample storage protocols, (4) substrate types, (5) sampling durations and (6) ice nucleation paths (condensation vs immersion freezing). The difference ($n\text{INP}_{\text{DFPC}} > n\text{INP}_{\text{WT-CRAFT}}$) reported in Sect. 3.1 may therefore derive from a combination of these factors.

The uncertainties of individual ice nucleation measurements cannot entirely explain the discrepancy observed. Even considering the largest error contribution, uncertainties can on average explain up to 50, 66 and 76% of the observed $n\text{INP}$ offset at -18, -22 and -15°C, respectively. These percentages were calculated by assuming that the measurement uncertainties combined with each other to determine the maximum possible reduction of $n\text{INP}$ difference (i.e., assuming the maximum possible underestimation of $n\text{INP}_{\text{DFPC}}$ and the maximum overestimation of $n\text{INP}_{\text{WT-CRAFT}}$) and considering only periods of parallel sampling (to minimize sources of discrepancy unrelated to measurement uncertainty).

A difference in the size dependant collection efficiency of the two aerosol particle samplers may have contributed to some extent to the $n\text{INP}$ offset. The $PM_1$ and $PM_{10}$ sampling inlet systems used for DFPC are certified with 100% collection efficiency at the flow rates employed. Similarly, the collection efficiency of sub- micrometre aerosol particles (tested using 200-300 nm mode test mineral dust particles) through the 47 mm filter sampler for WT-CRAFT is virtually 100%. For super-micrometre population, sampling efficiency falls to ~70% (tested with 2-3 µm test fibrous particles) presumably because the test particles stack inside the sampler inlet and/or a filter holder wall. Based on these considerations, we cannot rule out the impact of particle losses on $n\text{INP}_{\text{WT-CRAFT}}$ and resultant deviation from $n\text{INP}_{\text{DFPC}}$, especially under the conditions where super-micrometre INPs prevail (i.e., up to 32% and 65% at -15 °C in spring and summer, respectively, as shown in Table 2). If this collection size difference were a dominant factor, the expected gap would be different in the summer season when super-micrometre aerosol particles are more abundant than in spring. However, Figs. 2-4 show no systematic spring-to-summer increase in the deviation between $n\text{INP}_{\text{WT-CRAFT}}$ and $n\text{INP}_{\text{DFPC}}$.

Another factor could be the difference in the substrates used and their pore sizes (nitrocellulose and Track-Etched membranes with 0.45 and 0.2 µm pore size, respectively; see Sects. 2.1). While we cannot rule out the possibility that DFPC misses ice nucleation active aerosol particles in the size range between 0.2 and 0.45 µm, this difference might not substantially contribute to the gap as $n\text{INP}_{\text{DFPC}}$ is generally higher than $n\text{INP}_{\text{WT-CRAFT}}$.

The difference in sampling durations (~4 hours for the DFPC and ~4 days for WT-CRAFT) is another concern. Although we cannot exclude the possibility that short episodes of high INPs-containing air masses increased $n\text{INP}_{\text{DFPC}}$, it is unlikely that this factor can explain the systematic difference we are discussing here. This would presume a strong $n\text{INP}$ daily trend with the peak values coinciding with the DFPC sampling time. We excluded this by analysing the daily evolution of the particle number concentration, which presents no evident diurnal trend either in spring or summer (not shown). Similarly, the difference in sample storing methods cannot be the sole factor to explain the discrepancy. Beall et al. (2020) recently reported a decrease

in $n$INP of precipitation samples up to approximately 42% at -17°C < $T$ < -7°C due to different sample storing methods. As we kept the samples for DFPC at room air $T$ (and the WT-CRAFT samples at 4 °C except during transportation), INP suppression should have been more pronounced for DFPC than for WT-CRAFT. This is not supported by our observations, which showed $n$INP$_{DFPC}$ > $n$INP$_{WT-CRAFT}$.

Different sensitivity of Arctic INPs to different ice nucleation modes may be a plausible reason to explain the different results. Indeed, in condensation mode measurements, water vapours condense on the surface of sampled aerosol particles, possibly triggering pore condensation freezing (David et al., 2019; Wagner et al., 2016). In the case of immersion freezing measurements, pore condensation freezing is not assessable because all particles are scrubbed in the bulk suspension water and physicochemical properties of particles suspended in water may not be the same as the particles assessed by the condensation freezing method (e.g., soluble components are washed in the droplet water). The literature offers diverse results and data interpretations, evidencing both increase and suppression of the ice nucleation ability by soluble aerosol components (Reischel and Vali, 1975; Boose et al., 2016; Paramonov et al., 2018; Kumar et al., 2018; Whale et al., 2018). In fact, our past attempts to intercompare DFPC and WT-CRAFT measurements with different aerosol types yielded differing results. For instance, the analyses of microcrystalline and fibrous cellulose samples showed that DFPC tended to form more ice crystals than WT-CRAFT (Hiranuma et al., 2019), while the analyses of ambient continental aerosol particles collected in the Po Valley with identical sampling systems resulted in equivalent or higher ice crystal numbers in WT-CRAFT (not shown as the data are unpublished). The variations observed in measured $n$INP at least suggest some sensitivity of the aerosol particle type to the different ice activation modes (and vice versa). Nevertheless, more robust evidence is necessary to be conclusive regarding the influence of ice nucleation modes, which should be addressed in future studies, given its scientific relevance.

Finally, we consider that our INP detection techniques are within a reasonable agreement, according to the overall uncertainty of a subset of existing INP measurement techniques in a recent intercomparison study (deMott et al., 2017) and that, given the consistency of the time trends of $n$INP$_{DFPC}$ and $n$INP$_{WT-CRAFT}$, the discrepancy observed does not affect the conclusions presented in this study.

### 4.2. Interpretation of seasonal variability of $n$INP

In 2018, we observed limited INP seasonal variation depending on the activation $T$. In short, only within a limited range of $T$s (-18 to -21°C), our $n$INP$_{WT-CRAFT}$ exhibited statistically significant yet small variations (Sect.3.4). A similar observation of insignificant seasonal $n$INP trends from Ny-Ålesund has been reported by Schrod et al. (2020). The discrepancy between this current and previous studies demonstrating seasonal variations in $n$INP, may be due to the inter-annual variability of meteorological conditions and aerosol particle sources influencing the ambient abundance of INPs. However, studies of $n$INP in the Arctic and their temporal coverage are too limited to derive any conclusive interpretation of seasonal $n$INP trends at this stage. Future application of long-term online INP measurements (e.g., Möhler et al. (2021)) may shed light on the seasonal evolution of $n$INP at GVB and over the Arctic in general.

The analysis of the AF evidences more notable seasonal trends than $n$INP. Both techniques showed a statistically significant increase in the ice nucleation efficiency of atmospheric aerosol particles going from spring to summer (Sect. 3.4). The chemical tracer correlation analysis, the ground contribution analysis and the above-mentioned considerations on the different contributions of sub- and super-micrometer INPs in spring and summertime all suggest that the main sources of springtime INPs at GVB may be located outside the Arctic. They are deemed to derive from the lower latitude regions together with anthropogenic aerosols during the Arctic haze (Heidam et al., 1999; Stohl, 2006). Conversely, the summertime aerosol particle population is more related to local (Arctic) sources. Our AF estimates support the hypothesis that long-range transported aerosol particles from lower latitudes nucleate ice less efficiently than local-origin aerosol particles. This is in agreement with the findings of Hartmann et al. (2019), which showed a low impact of anthropogenic emissions on Arctic $n$INP, by comparing present-day and pre-industrial $n$INP values through the analysis of ice core records. We note, however, that although the correlation with chemical tracers suggests a common spatial origin for springtime INPs and anthropogenic aerosol particles, we were not able to assess to what extent anthropogenic aerosol particles contributed to the springtime INP loads observed.

The higher AF of summertime (local) aerosol particles may be related to the enhanced contribution of super-micrometer aerosol particles, which we have shown to be markedly more ice active than sub-micrometer particles. Nevertheless, we cannot exclude or quantify, the contribution of other physico-chemical properties of aerosol particles, which may vary between spring and summer (e.g., chemical composition).

It is worth considering that changes in the estimated AF are influenced not only by variations of $n$INP but also by variations of the concentration of non-ice-active aerosol particles, including secondary aerosols formed through new particle formation (NPF) mechanisms. Secondary aerosol particles may not contribute to INPs (Kanji et al., 2017), but they can lower the estimated AF. Recently, Beck et al. (2021) evidenced that different mechanisms, precursors and formation rates characterize spring and summertime NPF events at GVB. Dall'Osto et al. (2019) evidenced that the production of fresh particles is frequent during the period from May to August at GVB, while April is characterized by the presence of aged, accumulation mode particles. These aspects may influence the seasonal variation of the estimated AF. Dall'Osto et al. (2017; 2018; 2019) linked NPF frequency in the Arctic atmosphere to the fast-decreasing sea ice extent, probably via increased phytoplankton productivity. This leads to the hypothesis of increasing NFP impact in the future. By the same token, the predicted shrinking of snow and sea-ice coverage in the Arctic is likely to increase the ambient $n$INP from sea spray and terrestrial sources, such as mineral and soil dust particles (Tobo et al., 2019). Predicting future $n$INP and aerosol particle AF over the Arctic in such a rapidly changing scenario is challenging. It, however, provides the motivation for further investigation of INP processes in the Arctic region.

### 4.3. Sources of INPs at GVB

Our analysis points out that both marine and terrestrial sources may contribute to the INP population in the study area. Land sources may be potentially important given the higher ice activity of mineral dust and soil particles in comparison to marine particles (McCluskey et al., 2018). On the contrary, marine sources may be significant on account of the extension of ice-free

sea waters during the Arctic summer. This would also have implications for the future balance of terrestrial and marine INP sources in the Arctic, which is becoming increasingly warmer (Murray et al., 2021). The major limitation of our spatio-temporal correlation analysis and of the INP spatial source attribution approach (CWT) is the small number of samples available. This limits the time representativeness of the dataset and increases ouput uncertainty. Nevertheless, the consistency of the two independent approaches (spatio-temporal correlation analysis and CWT source location) provides a certain measure of credibility to the results presented. In particular, the approach adopted highlights the sea waters to the southwest of Svalbard, those immediately to the east of Greenland and to the northeast of Iceland as potential INP hotspots during our summer campaign. For this reason, we believe our results suggest that the marine biota may be a source of INPs in the Arctic. Nevertheless, further studies, based on more robust datasets, are necessary to confirm this result and achieve a more quantitative understanding of the relative importance of marine vs. terrestrial INP sources over the Arctic. In particular, online INP quantification methods have the potential to provide more suitable data for this kind of statistical approach and thereby help to clarify INP sources over the Arctic in the future.

## 5 Conclusions

This work presents the ambient concentration of INPs from the GVB observatory, near Ny-Ålesund, during spring-summer 2018. Aerosol particle samples were assessed for their ice nucleation ability and efficiency using offline immersion and condensation freezing techniques. The $n$INP values measured by DFPC ranged 33-185, 5-107 and 3-66 m$^{-3}$, for $T$s of -22, -18 and -15°C, respectively. At the same activation $T$s, WT-CRAFT measured 3-199, 1-34 and 1-4 m$^{-3}$. Although the two sets of data present a fair agreement in terms of $n$INP trends, a notable offset is evident, with the DFPC generally presenting higher $n$INPs than WT-CRAFT. This offset appears to be a factor of 8 at $T$ of -15°C. We considered many factors that could potentially explain the discrepancy observed (Sect. 4). While differences in the sampling approach and overall measurement uncertainties have certainly contributed to the offset, a different response of aerosol particles to the ice nucleation mode could be also considered as a potential contributing factor. All future investigations into Arctic INP compositions and the ice nucleation process employing both the condensation and immersion freezing approach will provide further understanding of this issue.

This study also examined the seasonality of INPs in the Arctic with respect to $n$INP and AF. Neither the condensation nor the immersion INP datasets indicate any marked $n$INP seasonal trend. We report a statistically significant spring to summer enhancement in $n$INP only for a narrow range of $T$s (-18 to -21°C) in WT-CRAFT observations, with the associated $n$INP enhancement never, however, exceeding a factor of three. On the other hand, the AF of atmospheric aerosol particles from GVB presents a statistically significant spring to summer increase for all the $T$s probed by DFPC, and between -17 and -22.5°C for WT-CRAFT. The AF increased up to ~6 times (-15°C) for DFPC and ~4 times (-20°C) for WT-CRAFT. A clear seasonal evolution of the super-micrometre INP contribution was observed by DFPC. This contribution was around 20% in spring (a maximum of 32% at -15 °C), increasing markedly in summer and at high $T$s (45% at $T$ of -22°C and 65% at $T$ of -15°C). Our calculations also evidence a markedly higher AF of coarse, compared to sub-micrometre, particles, with at least two orders of

magnitude difference between the two size regimes. This implies that super-micrometre aerosol particles play an important role as the source of summertime INPs at GVB. Additionally, our chemical tracer and back-trajectory analyses show the dominance of local aerosol particle sources during summer. In short, summer season ground conditions influenced the sampled air masses, suggesting that the summertime INP population is influenced by terrestrial and marine sources. Our summer-season

analysis also suggests a relationship between the biological activity in specific seawater regions and $n$INP at the sampling point. Nevertheless, we evidence that this result was achieved with a limited number of observations and that further studies, based on larger datasets, would be desirable for a better understanding of marine sources of INPs over the Arctic.

In contrast, the springtime INP population at GVB is mostly influenced by long-range transport of aerosol particles from lower latitudes. Spring-season air masses are characterized by markers of anthropogenic aerosol particles. Determining if such

anthropogenic particles contribute to the INP loads observed is beyond the scope of the current study. However, future investigation study of anthropogenic vs. natural INPs in the Arctic region with detailed assessment of INP composition will be an important step toward a comprehensive understanding of the impact of INPs in the Arctic cloud formation, precipitation and climate.

## 6 Data availability

The data discussed in this work are available at http://dx.doi.org/10.17632/zf4wdcc3bw.1

Satellite Chlorophyll data are available for download at http://marine.copernicus.eu/ (product identifier: OCEANCOLOUR_GLO_CHL_L4_REP_OBSERVATIONS_009_082).

## 7 Author contribution

MR, NH, and FB designed the concept of this collaborative research. MR led the writing off the manuscript with the support

of all authors. Methodology was developed by GS, FB, and NH, and measurements were conducted by GS, FB, CAR, MM, DC, SB, RT, and MP. Formal data analyses were carried out by MR, NH, FB, and KM. The revision effort was led by MR and NH with support of all authors.

## 8 Competing interests.

The authors declare that they have no conflict of interest.

## 9 Acknowledgements

The authors thank DSSTTA-CNR and its staff for the logistical support that allowed the realization of the experimental activity. This activity was funded by the H2020 EU Project FORCeS (Constrained aerosol forcing for improved climate projections;

Grant agreement ID: 821205). Francescopiero Calzolari is greatly acknowledged for his support to the maintenance of the DFPC.

N. Hiranuma and C. A. Rodriguez acknowledge the contributions of H.S. Vepuri, Y. Hou and Z. Salcido for their technical support on WT-CRAFT measurements. N. Hiranuma thanks for the funding support from Killgore Faculty Research Grant and the WTAMU's IoT and Research Computing program. N. Hiranuma also acknowledges partial financial support from Higher Education Assistance Fund (HEAF). This material is based upon work supported by the National Science Foundation under Grant No. 1941317 (CAREER: The Role of Ice-Nucleating Particles and Their Feedback on Clouds in Warming Arctic

Climate). The authors acknowledge the NySMAC, Ny-Ålesund Atmosphere Research Flagship Programme, for allowing the organization of a collaborative workshop meeting held in Bologna, Italy, in 2017. The workshop provided a venue for authors to come together that fostered this collaboration.

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

**Table 1. Compilation of previous ground level measurements of $n$INP in the Arctic**

| Reference | Location | Period | INP quantification method | Ice nucleation mode | $T$ range (°C) | $n$INP (m$^{-3}$) |
|---|---|---|---|---|---|---|
| Borys (1983) | Multiple | winter-summer | dynamic chamber | condensation | -16 to -28 | <1-~80 (-16°C) ~50-~300 (-28°C) |
| Bigg (1996) | High Arctic (cruise) | 1 August - 6 September 1991 | static chamber | condensation | 12.5, -15.0, -17.5 | <1-250 (-15°C) |
| Bigg and Leck (2001) | High Arctic (cruise) | 16 July - 23 September 1996 | static chamber | condensation | -15 | <1-~100 |
| Conen et al. (2016) | Haldde observatory, Norway | 02 - 06 July 2015 | droplet freezing | immersion | -7 to -15 | 0-0.3 (-8°C) 1.7-12.2 (-15°C) |
| Mason et al. (2016) | Alert, Canada | 29 March - 23 July 2014 | droplet freezing | immersion | -15 to -25 | 50 (-15°C)* 220 (-20°C)* 990 (-25°C)* |
| Creamean et al. (2018) | Oliktok Point, Alaska | 1 March - 31 May 2017 | droplet freezing | immersion | -5 to -28 | 0.07-2 (10°C) 30-70 (-25°C) |
| Si et al. (2018) | Lancaster Sound, Canada | 20 July 2014 | droplet freezing | immersion | -15 to -25 | 0 (-15°C) 160 (-20°C) 670 (-25°C) |
| Irish et al. (2019) | Multiple (cruise) | 14 July - 12 August 2014 | droplet freezing | immersion | -15 to -25 | 5 (-15°C)* 44 (-20°C)* 154 (-25°C)* |
| Santl-Temkiv et al. (2019) | Villum, Greenland | summer 2016 | droplet freezing | immersion | -6 to -20 | 17.8 (-10°C) 71.5 (-15 °C) |
| Si et al. (2019) | Alert, Canada | March 2016 | droplet freezing | immersion | -10 to -30 | 5±2 (-15°C)* 20±4 (-20°C)* 186±40 (-25°C)* |
| Tobo et al. (2019) | Mt. Zeppelin, Svalbard | July 2016; March 2017 | droplet freezing | immersion | -9 to -25 | ~1-5 (-15°C) 2-300 (-20°C) 30-~1000 (-25°C) |
| Wex et al. (2019) | Alert, Canada | May 2015 - April 2016 | droplet freezing | immersion | -5 to -26 | 0.02-20 (-7°C) ~0.4-20 (-15°C) 10-20 (-23°C) |
| | Utqiagvik, Alaska | June 2012 - May 2013 | droplet freezing | immersion | -5 to -26 | 0.02-20 (-7°C) 0.2-~20 (-15°C) ~3-~20 (-19°C) |
| | Ny-Ålesund, Svalbard | March - July 2012 | droplet freezing | immersion | -5 to -26 | <0.1-~0.7 (-7°C) 0.7-~30 (-15°C) ~30 (-23°C) |
| | Villum, Greenland | 2015 | droplet freezing | immersion | -5 to -26 | <0.1-0.2 (-6°C) ~1-~10 (-15°C) ~20 (-20°C) |
| Welti et al. (2020) | High Arctic (cruise) | Multiple | droplet freezing | immersion | -5 to -40 | 1-20 (-15°C) |
| Schrod et al. (2020) | Mt. Zeppelin, Svalbard | May 2015 - Jan 2017 | FRIDGE | condensation | -20 to -30 | <40-~3000 (-20°C) <100-~2000 (-25°C) |

* Average values


**Table 2: Average (± standard error) and median (in brackets) *n*INP measured at GVB during 2018. The min-max range is also reported besides the median value. "dl" indicates *n*INP below the detection limit. The reported statistics refer to 16, 17, 11 and 17 samples for DFPC in spring, DFPC in summer, WT-CRAFT in spring and WT-CRAFT in summer, respectively.**

| | | -22°C | | | -18°C | | | -15°C | | |
|---|---|---|---|---|---|---|---|---|---|---|
| | | $PM_1$ | $PM_{10}$ | Coarse contrib. | $PM_1$ | $PM_{10}$ | Coarse contrib. | $PM_1$ | $PM_{10}$ | Coarse contrib. |
| | | $m^{-3}$ | $m^{-3}$ | % | $m^{-3}$ | $m^{-3}$ | % | $m^{-3}$ | $m^{-3}$ | % |
| DFPC | Spring | 97±12 (85; 29-183) | 116±11 (115; 55-185) | 21±6 (20; 0-68) | 45±6 (49;3-85) | 55±7 (53;5-90) | 20±5 (17; 0-63) | 13±2 (14; dl-30) | 18±2 (20; 3-37) | 32±9 (22; 0-100) |
| | Summer | 43±7 (38;6-103) | 74±6 (77; 33-135) | 45±6 (48; 0-80) | 23±3 (23; 9-61) | 50±5 (47; 18-107) | 53±4 (58; 8-70) | 9±2 (7; dl-39) | 24±3 (20; 6-66) | 65±6 (72; 12-100) |
| WT-CRAFT | Spring(*) | - | 20±4 (19; 3-43) | - | - | 2±<1 (1; dl-6) | - | - | 1±<1 (dl-4) | - |
| | Summer(**) | | 46±12 (37; 4-199) | - | - | 9±2 (6; 2-33) | - | - | 2±<1 (2; dl-4) | - |

*16 April – 03 June 2018

**03 June - 15 August 2018

Table 3a: Correlations of *n*INP, in PM$_1$ and PM$_{10}$ samples by DFPC, with chemical tracers during the spring campaign. Coefficients reported in italic are statistically significant with p<0.10, while those in bold are statistically significant with p<0.05. Coefficients associated with p>0.2 have not been reported.

| | -22°C (n=16) | | -18°C (n=16) | | -15°C (n=15) | |
|---|---|---|---|---|---|---|
| | PM$_1$ | PM$_{10}$ | PM$_1$ | PM$_{10}$ | PM$_1$ | PM$_{10}$ |
| PM$_{10}$ mass | | | | | | *0.49* |
| Na$^+$ | **-0.61** | *-0.49* | **-0.59** | -0.36 | **-0.60** | |
| Mg$^{+2}$ | **-0.52** | | -0.38 | | -0.43 | |
| Ca$^{+2}$ | | *0.45* | | | 0.34 | **0.64** |
| Cl$^-$ | **-0.64** | *-0.51* | **-0.64** | -0.42 | **-0.65** | |
| NO$_3^-$ | **0.61** | **0.59** | **0.67** | **0.73** | **0.72** | **0.54** |
| MSA | | | -0.42 | **-0.52** | -0.40 | **-0.65** |
| Li$^+$ | -0.36 | | | | | |
| nssSO$_4^{-2}$ | *0.43* | *0.44* | **0.53** | *0.43* | **0.62** | **0.67** |
| nssK$^+$ | **0.60** | **0.56** | **0.68** | **0.56** | **0.77** | **0.72** |

Table 3b: Correlations of *n*INP, in PM$_1$ and PM$_{10}$ samples by DFPC, with chemical tracers during the summer campaign. Coefficients reported in italic are statistically significant with p<0.10, while those in bold are statistically significant with p<0.05. Coefficients associated with p>0.2 have not been reported.

| | -22 (n=17) | | -18 (n=17) | | -15 (n=16) | |
|---|---|---|---|---|---|---|
| | PM$_1$ | PM$_{10}$ | PM$_1$ | PM$_{10}$ | PM$_1$ | PM$_{10}$ |
| PM$_{10}$ mass | | | | -0.35 | -0.32 | ***-0.49*** |
| Na$^+$ | | -0.36 | -0.39 | *-0.43* | | **-0.52** |
| Mg$^{+2}$ | | -0.35 | -0.41 | *-0.48* | -0.35 | **-0.57** |
| Ca$^{+2}$ | | -0.33 | | *-0.42* | -0.44 | **-0.55** |
| Cl$^-$ | | -0.38 | -0.37 | *-0.45* | | **-0.51** |
| NO$_3^-$ | | | | | -0.33 | -0.36 |
| MSA | | | | | | -0.37 |
| Li$^+$ | | -0.32 | -0.37 | *-0.42* | -0.35 | **-0.49** |
| nssSO$_4^{-2}$ | | | | | | -0.32 |
| nssK$^+$ | 0.36 | | | | | |

**Table 4a: Correlations of $n$INP by WT-CRAFT with chemical tracers during spring (April-May) 2018. Coefficients reported in italic are statistically significant with p<0.10, while those in bold are statistically significant with p<0.05. Coefficients associated with p>0.2 have not been reported.**

| | -15.0 (n=5) | -18.0 (n=7) | -20.0 (n=10) | -22.0 (n=11) | -24.0 (n=11) |
|---|---|---|---|---|---|
| $PM_{10}$ mass | 0.77 | -0.63 | | | |
| $Na^+$ | | | | | |
| $Mg^{+2}$ | | | | | |
| $Ca^{+2}$ | *0.83* | | | *0.53* | 0.47 |
| $Cl^-$ | | | | | |
| $NO_3^-$ | | | **0.63** | **0.74** | *0.60* |
| MSA | *-0.81* | | | **-0.81** | **-0.82** |
| $Li^+$ | | -0.59 | | | |
| $nssSO_4^{-2}$ | 0.75 | | 0.43 | **0.95** | **0.87** |
| $nssK^+$ | | | | **0.87** | **0.80** |

**Table 4b: Correlations of $n$INP by WT-CRAFT with chemical tracers during summer (June-August) 2018. Coefficients reported in italic are statistically significant with p<0.10, while those in bold are statistically significant with p<0.05. Coefficients associated with p>0.2 have not been reported.**

| | -15.0 (n=15) | -18.0 (n=16) | -20.0 (n=17) | -22.0 (n=17) | -24.0 (n=17) |
|---|---|---|---|---|---|
| $PM_{10}$ mass | | | | | |
| $Na^+$ | *-0.45* | -0.36 | | | -0.33 |
| $Mg^{+2}$ | -0.42 | -0.37 | | | |
| $Ca^{+2}$ | -0.40 | | | | |
| $Cl^-$ | -0.41 | -0.38 | -0.36 | | -0.37 |
| $NO_3^-$ | | -0.38 | | | |
| MSA | | | | | |
| $Li^+$ | | | | | |
| $nssSO_4^{-2}$ | | *-0.44* | | | |
| $nssK^+$ | | | | | |

**Table 5: Correlation coefficient (R) resulting from the linear regression between $n$INP (at $T$ = -15, -18 and -22°C) and the contribution of the four considered ground types. Values reported in bold are statistically significant (p<0.05).**

| | DFPC_spring | | |
| --- | --- | --- | --- |
| | INP_-15 (n=15) | INP-18 (n=16) | INP-22 (n=16) |
| Sea-Water | **-0.63** | **-0.54** | -0.39 |
| Land | -0.05 | 0.36 | -0.25 |
| Sea-Ice | 0.24 | 0.16 | 0.08 |
| Snow | 0.23 | 0.18 | 0.25 |
| | DFPC_summer | | |
| | INP_-15 (n=16) | INP-18 (n=17) | INP-22 (n=17) |
| Sea-Water | **-0.60** | -0.43 | **-0.48** |
| Land | **0.86** | **0.72** | **0.65** |
| Sea-Ice | -0.15 | -0.24 | -0.11 |
| Snow | 0.39 | 0.32 | 0.33 |
| | WT-CRAFT | | |
| | INP_-15 (n=20) | INP-18 (n=23) | INP-22 (n=28) |
| Sea-Water | -0.04 | 0.17 | 0.02 |
| Land | 0.29 | **0.54** | **0.42** |
| Sea-Ice | -0.21 | -0.16 | 0.01 |
| Snow | 0.40 | -0.19 | -0.18 |

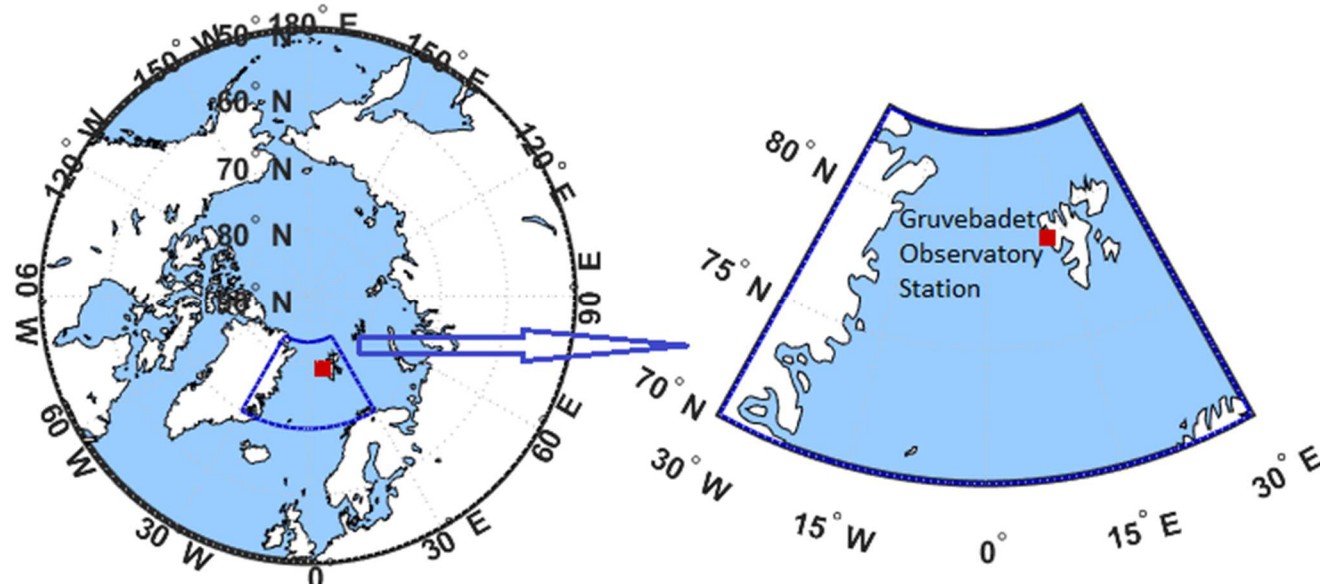

**Figure 1: Geographic location of the sampling station, Gruvebadet observatory at Ny-Ålesund, Svalbard Islands.**

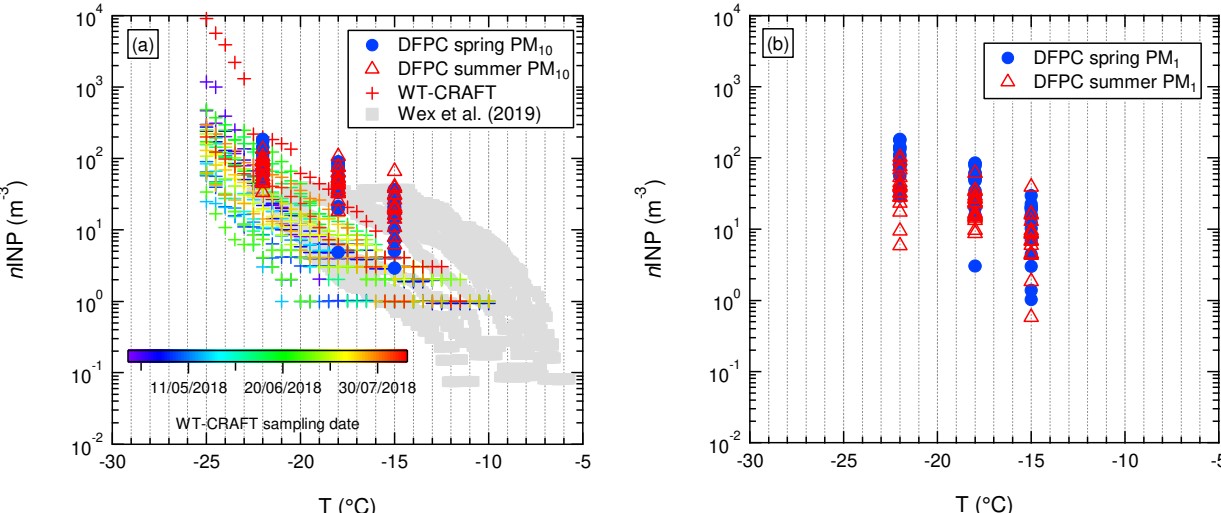

**Figure 2: Ambient $n$INP as a function of the activation $T$ assessed for samples from GVB during 2018 by DFPC and WT-CRAFT. DFPC data are divided in spring (blue) and summer (red) samples, while WT-CRAFT data are colour-coded according to the sampling date. (a) PM$_{10}$ (DFPC) and TSP (WT-CRAFT) data. (b) PM$_1$ data (available only for DFPC). For comparison purposes, the data from Wex et al. (2019), which refer to PM$_{10}$ samples, are also reported in the plot (a).**

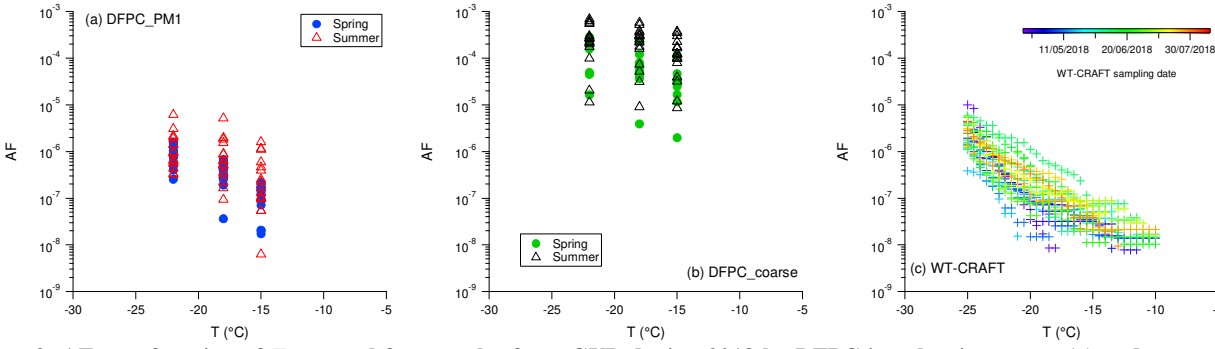


**Figure 3: AF as a function of *T* assessed for samples from GVB during 2018 by DFPC in sub-micrometer (a) and coarse (b) size ranges and by WT-CRAFT (c). Particle size ranges used to calculate AF are 0.1-1 μm, 0.1-10 μm and 0.1-10 μm for DFPC PM₁, DFPC PM₁₀ and WT-CRAFT samples, respectively.**

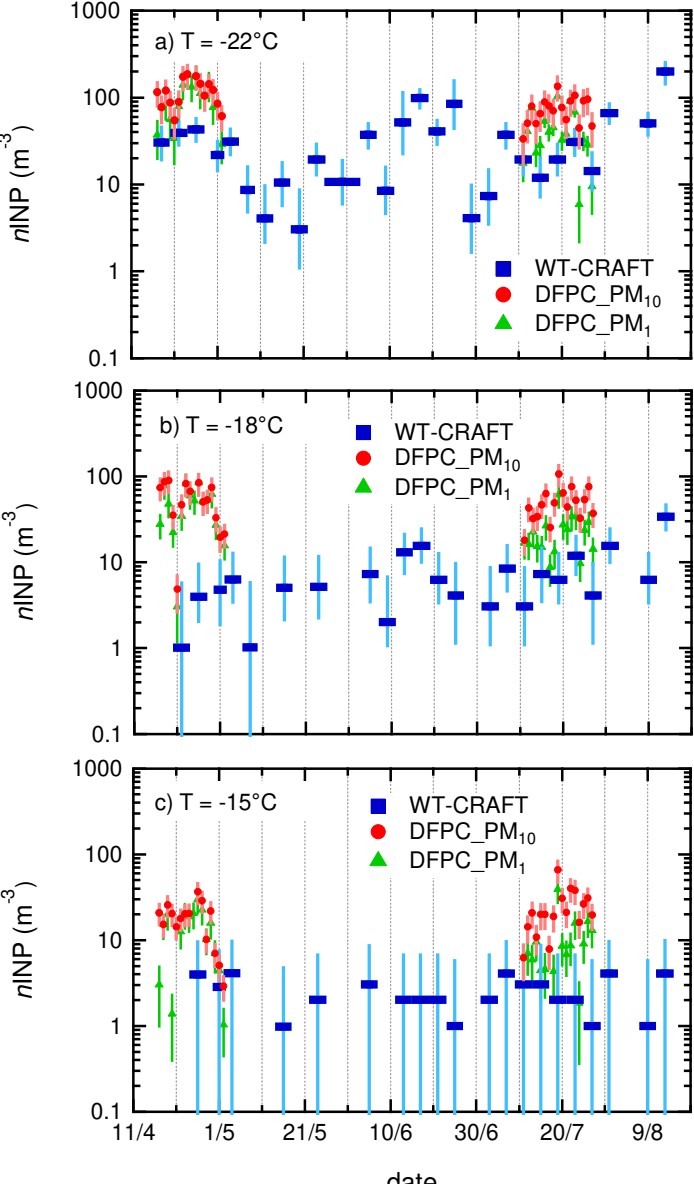

**Figure 4: Time series of *n*INP at GVB during 2018 measured by DFPC (PM₁₀ and PM₁) and WT-CRAFT. Horizontal bars indicate the time span of WT-CRAFT samples (ca. 4 days for the majority of samples). Vertical bars indicate the overall measurement uncertainty as indicated in the Sect. 2.2.1 (DFPC) and 2.2.2 (WT-CRAFT).**


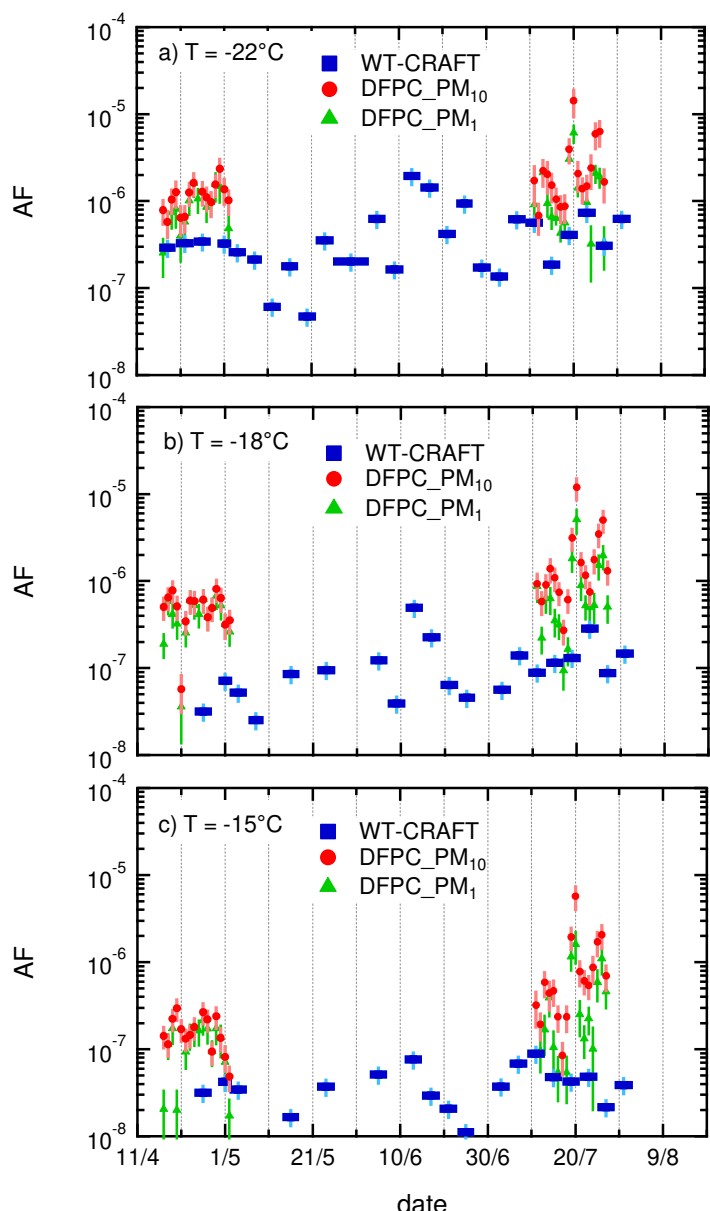

**Figure 5: Time series of the activated fraction at GVB during 2018 measured by DFPC (PM₁₀ and PM1) and WT-CRAFT. Horizontal bars indicate the period of WT-CRAFT samples (ca. 4 days for the majority of samples). Vertical bars indicate the overall AF uncertainty as indicated in the Sect. 2.2.1 (DFPC) and 2.2.2 (WT-CRAFT).**


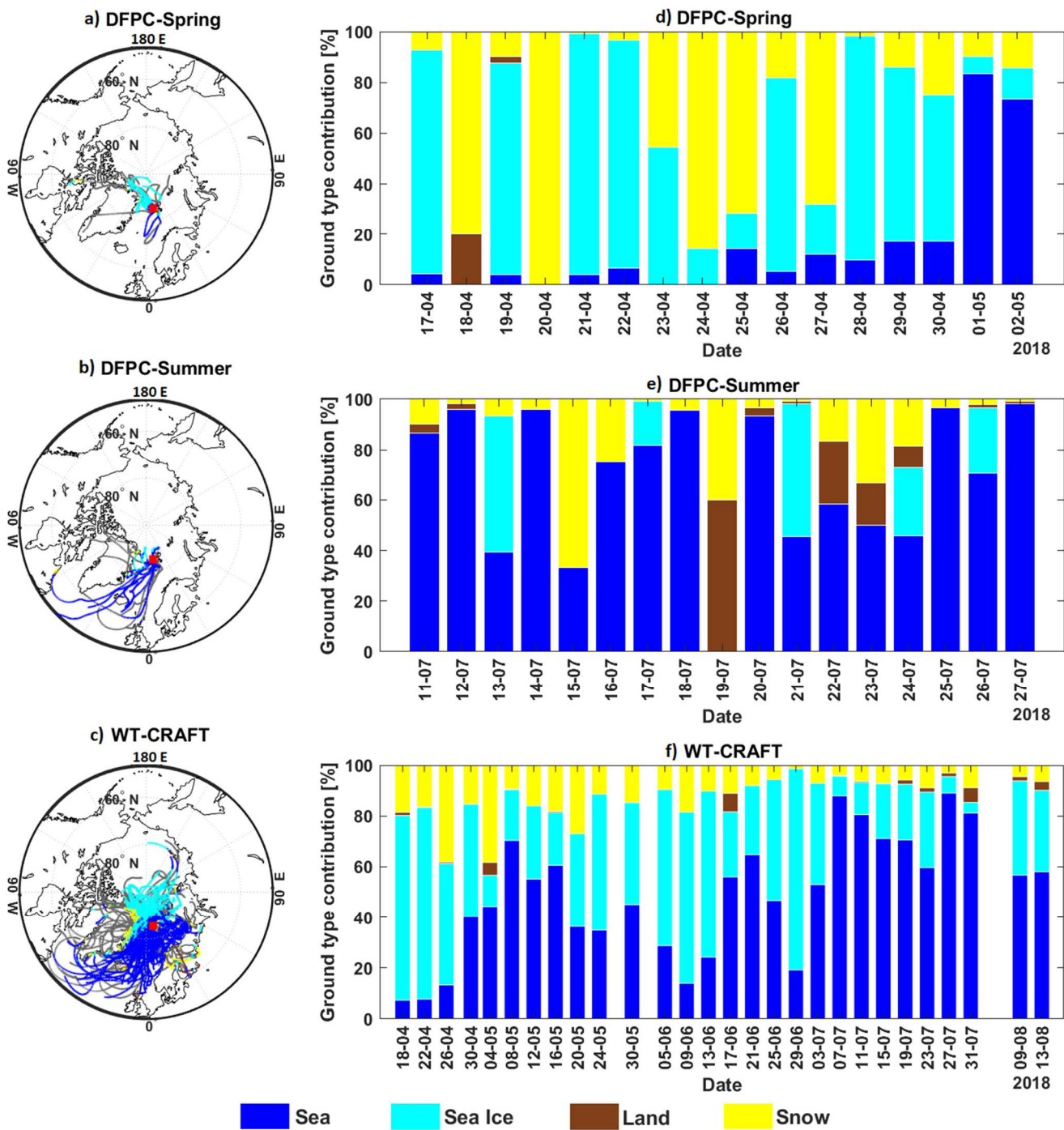

**Figure 6: Air mass back-trajectories and ground type influence on low-travelling (<500 m) air masses for DFPC in spring (a), DFPC in summer (b) and WT-CRAFT (c) measurements. Back-trajectories reported in grey in the maps passed above 500 m amsl and were therefore excluded from the analysis. The ground type categories are described in Sect. 2.3.4.**


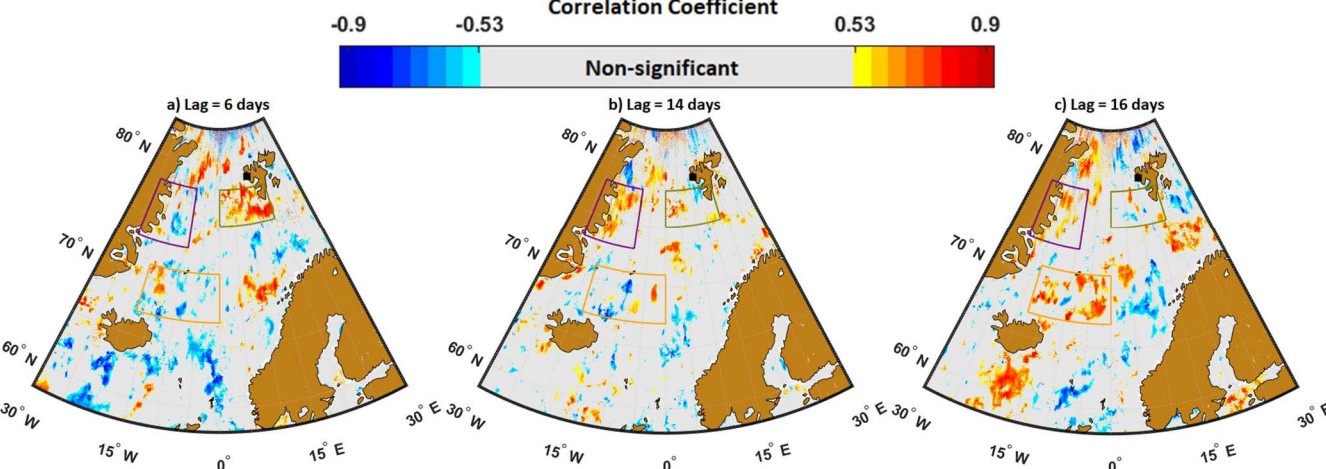

**Figure 7: Correlation maps for** $n$**INP$_{PM1}$ at** $T$ **= -15°C with (left) 6, (center) 14 and (right) 16 days time lag. The color scale indicates the correlation coefficient; not significant (p>0.05) pixels are reported in grey. The green, purple and orange boxes highlight sea regions characterized by high correlation at 6, 14 and 16 days, respectively.**

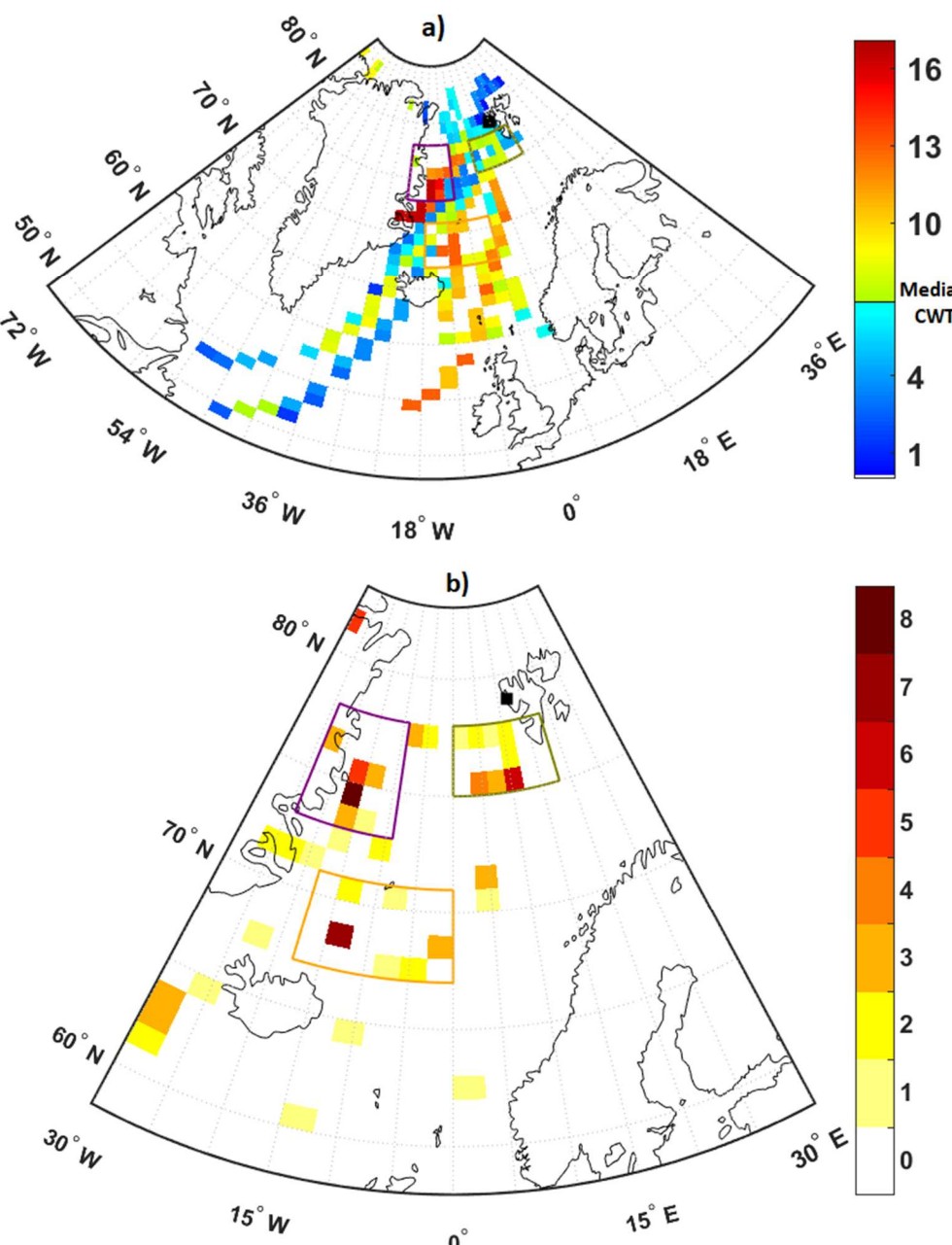

**Figure 8. (a) CWT source map for the $n$INP$_{PM1}$ at $T$ = -15°C dataset. The color scale indicates the CWT value. (b) Spatial distribution of fine INP sources identified by merging the results of the spatio-temporal correlation analysis and of the CWT algorithm. The color scale reflects how many times a given pixel has CWT ≥ median and significant correlation coefficient by running time lag from 5 to 20 days. The same purple, green and orange boxes of Figure 7 are reported to facilitate the comparison.**

