# Peer review of "Ice-nucleating particle concentration measurements from Ny-Ålesund during the Arctic Spring-Summer in 2018"

_Atmospheric Chemistry and Physics, 2020_

## Referee Comment (RC1) · Anonymous Referee #1 · 4 Aug 2020

**General comments**

Ambient measurements from Spitzbergen during spring and summer 2018 are reported and analysed. The dataset is a welcome addition to the growing body of ice nucleation concentration measurements from the Arctic region.

For the analysis the authors correlate the measured INP concentrations to bulk particle properties, season, meteorology and airmass trajectory. Unfortunately, the analysis is not well motivated by hypotheses, and the results not presented clearly. Additionally, I suspect that the set of measurements is too small to perform a robust analysis and the found correlations might be random. By ignoring this, the authors got mislead to overinterpretations and speculative conclusions. The interpretation of data (e.g. concerning INP size and land vs. marine contributions) agrees with previous studies and no new insights are obtained. The one interesting finding is that there was no seasonal variation observed in 2018. The authors need to be more specific in their descriptions, quantitative within reason for the interpretation and visualize their findings clearer to turn this manuscript into a valuable contribution to the field of ambient ice nucleation measurements.

**Specific comments**

Major changes are suggested for each section. Minor corrections or clarification requests at specific line numbers are listed below.

**Structure**
The manuscript could be structured better. Adding a Discussion section instead of including the discussion with the results would help the organization of the paper. Currently the Result section has mixed-in discussion, interpretation, literature comparison and some method description. Isolating the Results, visualizing and explaining them more specifically would be helpful to judge the interpretation.

**Citations**
A high number of citations are given, but it is often unspecific what information can be found in which citation or why several citations are listed. Better integration in the text would be helpful instead listing multiple citations at the end of a sentence.

**Title**
The title does not fit the manuscript. There was never doubt that multiple sources contribute INP at different temperatures. Something along the line of "Concentration of ice nucleating particles measured at Ny-Ålesund during 2018" would be more accurate.

**Abstract**
The abstract uses wordy, euphemistic language. In the interest of clarity, the tone should be revised to be strait forward. At the moment the abstract consists of too many unsupported statements and vague conclusions that are incomprehensive before reading the manuscript in detail.
Line 15 ff: INP sources, INP concentrations and ice nucleation properties should be distinguished clearly. They are not the same.

Line 15: INP sources are unknown not "inadequately understood"
Line 18: specify what properties were characterized
Line 22: why are the temperatures (-15°C, -18°C, -22°C) probed with the DFPC not evenly spaced?
Line 24-25: The dependence on ice nucleation mode is not investigated in the manuscript and a sampling issues is a more reasonable cause for the discrepancy.
Line 26: specify how the increase in coarse INP was observed.
Line 27: Explain why increase in coarse INP fraction suggests local sources.
Line 28: Speculative. INP active at -15°C are not exclusively biological particles. The source of the particles active at -15°C in this study are unknown.
Line 30: specify "distinct behaviours of particles"
Line 31: How was the inter-annual variability of local INP sources previously considered? Specify the evidence for an inter-annual variability based on the current dataset and all available data from Spitzbergen.  Consider that the 2012 data set is only 12 measurements and not a strong dataset to compare to.
Line 33: It seems trivial that the INP population can be contributed by terrestrial and marine sources on an island.
Line 35: specify nucleation ability. Higher particle load can explain higher INP concentrations without increased ice nucleation activity.
Line 37-40: requires reading the paper to understand this description.

**Introduction**
The introduction is not tailored enough to the subject of this study. To make the introduction more effective at explaining and putting into perspective what follows, I suggest to focus on: ice nucleation mechanisms (condensation and immersion mode), INP-cloud interaction in the Arctic without going into detail on radiative effects, previous INP measurements in the Arctic and what has been learned about potential sources, tends, dependencies of nINP in the Arctic. A lot of literature is currently discussed in the Result section. Better to include a concise discussion of relevant literature to the introduction to develop the hypotheses which are then addressed in this study. Currently literature review is used inefficiently in the Result section to point to similar conclusions found elsewhere in literature.
Line 46: provide reference and explain how aerosol affect cloud properties
Line49-57: specify how the numerous local processes and feedbacks interact to affect structure, phase and persistence of clouds in the Arctic. Beyond what is generally true for INP-cloud interactions, explain why Arctic clouds are sensitive to INP concentrations. Provide a reference for the uncertainty associated with nINP, eg. DeBoer et al., 2018.
Line 59: the references are not "recently"
Line 61-62: specify "most" ice nucleation processes. It would be good to introduce ice nucleation mechanism in more detail.
Line 63-64: It is not generally true that biogenic INP nucleate ice at temperatures above -15°C. It is a bit of a stretch from ice nucleation properties to rain. Provide a more coherent explanation and provide the link to INP in the Arctic region.
Line 67-86: Specify that the literature review is separated into condensation and immersion mode measurements as well as separated into short and year around observations.
Line 67: specify "short periods of time"
Line 72: add some information how Hartmann et al. confirmed this.
Line 74: How did Bigg et al. identify the Ocean was the main source? There was a third Arctic cruise in 2001 and a more recent expedition in 2017. They are reported in Welti et al., 2020.
Line 76-86: Mention at what temperatures Mason et al., 2016, Si et al., 2018, Creamean

et al., 2018, Irish et al., 2019 reported data. This section seems to contradict line 61-64 where it is argued that INP active at T>-15°C are biological and not mineral dust.

Line 88: Quantify the increase. Contrast to the fact that often ambient nINP measurements scatter within 1-2 order of magnitude within less than a day, highlighting that caution should be used when interpreting variations smaller than one order of magnitude in such dataset.

Line 90: A time-series showing measured nINP at -15°C as function of DOY from all the listed measurements would be a helpful addition to illustrate the discussion and to show if the Arctic region as a whole experiences seasonal variations or if these are local phenomena.

Line 94: As your literature review exemplifies, there is no general "gap" of INP measurements in the Arctic.

**Methods**
The method of how INP concentrations are determined with the WT-CRAFT starting from the air volume sampled through filter to counting the number of frozen aliquots, should be explained in more detail, focusing on how nINP can be derived step by step from sample volume, water volume, droplet volume. How the preparation is done practically is of secondary interest.

In section 2.3 it should be clarified (by a short explanation at the beginning of each subsection) for what purpose the measurement or analysis is performed or used in the context of this paper. Here it would be helpful to already know from the introduction what the aim of the analysis is or what hypotheses are going to be tested with these data. Give context in the introduction section: why are you investigating ground type, trajectories, chlorophyll,…

Line 99: Refer to the location as Gruvebadet station throughout the paper and introduce the abbreviation (GVB) here.

Line 100: Point to fig. S1 showing the location on a map.

Line 102: Can Longyearbyen in the SE of the GVB station contribute to the aerosol population?

Line 110: Is filter overloading an issue in the clean Arctic air? The WT-CRAFT filters were sampled longer, with higher flow and on filter with smaller pore size. Where these filters potentially overloaded?

Line 115: Pumping 150lpm through filter with 0.2um pore size creates a huge pressure drop. Can you comment on how sampling was possible without fracturing the filter? How was the flow monitored? In this setup the filter probably acted as flow limiter rather than the critical orifice before the pump. An overestimation of the sample flow would explain the offset between WT-CRAFT and DFPC.

Line 116: specify pump model

Line 118: Is filter overloading (line 110) an issue for 4-day samples? The volume sampled is more than 100-times larger for WT-CRAFT than for the DFPC filters.

Line 127: Specify how uncertainties in T and Sw convert into uncertainties in nINP.

Line 128f: Has a systematic difference between condensation and immersion mode ice nucleation been observed in these inter-comparisons?

Line 132: The large sample volume of over 800m$^3$ would allow to detect approximately 100-times lower INP concentrations than 1 m$^{-3}$. Why was the analysis not performed in the full range?

Line 133: State how the two CRAFT systems are different. All I could find in Hiranuma et al., 2019 was that they used different sizes of droplets. This is not an instrumental difference.

Line 135: Explain how the uncertainty in ice nucleation efficiency is derived.

Line 136: repetition, delete

Line 139: provide camera model specifics

Line 140: define how INP concentrations are derived.

Line 141f: specify water volumes, for 1 INP per m$^3$ that would be 90mL for 4 day samples and 180mL for 8 day samples.

Line 143f: specify water volume used for soaking

Line 144: specify how mechanical vibration was applied. By sonication?

Line 145: How were droplets prepared?

Line 146: The method by how much the sample was diluted is not explained clearly. Specify the dilution water volume and how dilution was considered for the derivation of INP concentrations.

Line 149: How was the stitching performed? At what temperature were the spectra stitched? In Fig.1 a a jump in max concentration appears at -23°C. Is this the range of the diluted measurements? To show the "absence of failure" of this technique, it would be helpful to show the individual measurements in Fig.1 and not just the range.

Line 153f: Derivation of nINP must be defined clearer. Dividing the number of INP by the total sample volume is incorrect. The concentration is calculated from the filtered air volume, dilution water volume, droplet volume and number.

Line 154: As you state on line 379, only a small, T-dependent fraction of ambient aerosol are INP. Therefore, dividing the number of INP by the bulk particle surface has no physical meaning for a heterogeneous aerosol population. I recommend changing this approach to deriving only the fraction of particles that are ice active, by dividing the INP concentration by the particle concentration.

Line 161: specify APS measurement range

Line 163: what are the references pointing at? How were the size distributions averaged for the sampling interval of the filters?

Line 164: What substance is assumed for a density of 1.95 gcm$^{-3}$? Mineral dust and sea salt have higher densities.

Line 173: Where was this analysis performed? While handling during the analysis is relevant, handling before and after sampling, storage and transport are equally important and could be described.

Line 181-196 (Section 2.3.4.): Very similar to the text in Wex et al., 2019. Sentences in line 191-195 are copied from Sec. 2.7 in Wex et al., 2019. It is difficult to understand without consulting the original description. Section 2.3.4. should be rewritten entirely, explaining more clearly how ground types were categorized and how trajectories were merged to the filter sampling intervals. I suggest (instead of the analysis in 2.3.5.) to include high chlorophyll concentration as a fifth ground type in this analysis. Additionally, precipitation along the trajectories should be considered.

Line 201: state the temporal resolution of the dataset.

Line 202: Shift the description of how INP concentrations and chlorophyll maps were merged from the Result section to here. The DFPC summer data consists of only 17 measurement days and 3 are excluded because land influence, leaving 14 data points. Demonstrate that correlations are robust by showing some scatterplots of grid cells with a strong correlation as a supplement.

Line 203: Explain why a relationship between INP and chlorophyll concentration is expected.

Line 203: Excluding the samples with land input is mentioned several times. Elaborate why this is important.

Line 212, 213: specify, concentration of INP

Line 217: Specify how many trajectories were used and demonstrate that this is a large enough sample to draw conclusions. Looking at the figures it seems that higher CWT is found where more trajectories passed.

Line 227: Justify that longer residence time in a grid box is related to higher INP concentration. I would expect high windspeed to generate more particles, but also less

endpoints at the location because the trajectory moves faster. Discuss assumptions made for this analysis.

Line 227: What uncertainties are avoided by weighting? Motivate the application of a weighting factor. This methodology makes no sense to me and needs a clearer explanation.

**Results**

Line 237-238, 244ff: Discuss different ice nucleation modes in the introduction section. Remove here.

Line 240: specify "sharper"

Line 241: Provide a more detailed explanation how "time resolution" and "sampling activities" can explain these differences. Calculate how much of the difference can be explained by the uncertainty introduced by the ice nucleation analysis and how much from uncertainties in sample volume.

Line 242: unclear what the references point at

Line 242f: Elaborate based on what it is a valid assumption that the ice nucleation mode generates the observed difference of higher nINP from condensation than immersion mode.

Line 247: Vali 1975 is a better reference for ice nucleation modes

Line 247-255: This section is speculative. Provide an explanation how the different mechanisms can exert an influence on nINP and why in particular on mixed particles. Much more probable would be an uncertainty in the sample volume.

Line 257ff (Sec. 3.2): For a field study as this, aiming to learn something about the abundance and nature or source of INP, I would consider the differences in concentration of minor importance. Focus should be on the big picture, on trends while being cautious not to overinterpret the data.

Line 264: give concentration ranges at -15°C, -18°C, -22°C to compare to DFPC instead. Specify what can be learned from these concentration ranges.

Line 265: repetition from introduction line 67-68.

Line 272: It is implied that Borys, 1983, Bigg 1996, Bigg 2001 did not measure in the immersion mode. This should be clarified. I recommend merging the literature review here into the introduction.

Line 265-290: Consider presenting the comparison to literature in form of a table and to shift it into a Discussion section. Point out and discuss any systematic differences between marine and land influenced data from the Arctic region at specific temperatures.

Line 279-280: Explain how parameters intervene with INP concentrations. Specify what is meant by "particle activation modality". Quantify the conclusion that the data are generally consistent to literature.

Line 282: Quantify "reasonable agreement"

Line 285: Quantify "overlaps well"

Line 286: Quantify "wider range"

Line 289: What other factors can explain the differences? It would be helpful to specify the upper and lower detection limits of the methods used here for a comparison to Wex et al., 2019.

Line 291-299: Wide reached and speculative. Sec. 3.7.2. does not provide quantitative evidence on the contribution of continental particles.

Line 299: Name the locations of the high-altitude and coastal measurements in Rinaldi et al., 2017, 2019.

Line 300: Ice formation is usually observed at -15°C in filter based INP measurements and not unique. It is also present in dust rich environments. Provide references for examples showing otherwise.

Line 303: Specify the "special feature"

Line 304: This is the only reference to Fig.2. The figure is not relevant and can be removed.
Line 309: Two size ranges do not qualify as "size distribution".
Line 309: Instead of Table 1, provide a figure showing a scatterplot of INP concentrations measured on PM1 versus PM10 filter in the same time interval (day). All 3 temperatures can be included. Use different colours for spring and summer data.
Line 311: Specify why long distance to source is suggested. Quantify "long distance".
Line 314: To substantiate this interpretation, compare to by how much the concentration of particles in the coarse fraction change from spring to summer, based on the measured size distribution.
Line 316: Speculative, the coarse particles could be dust particles. It is not clear to what "above considerations" this is liked to.
Line 308: PM1 data is not depicted in any figure but used in the analysis. Include DFPC PM1 data to Fig.1 and Fig.3.
Line 308-318: the difference between PM1 and PM10 samples is not obvious from this section. Please provide a figure showing both time series together at -15°C, -18°C, -22°C as well as a scatterplot comparing PM1 to PM10 INP concentrations.
Line 319-325: INP concentrations from PM1 and PM10 should be compared to particle concentrations <1um and <10um including all smaller sizes instead of only super-micrometre particles. Otherwise the comparison is not objective and only implies that INP concentrations were similar for both cut-offs and the difference is introduced by the choice through what size range was divided.
Line 327-332: shift to Discussion or introduction. Show how the current data compares to trends found in other studies, e.g., Wex et al., 2019 by plotting the data (at -15°C, -18°C, -22°C) as a function of DOY into the same time-series.
Line 333-343: The main findings need to be worked out clearer in this section. Listing a lot of factors at random temperatures in the text is not helpful to understand the situation. Show that the small dataset can be used to determine robust trends. Factors on the order of 2 are small and should not be overinterpreted. Scattering within a season is much higher.
Line 337: What is the reason for limiting the WT-CRAFT dataset to the same period?
Line 340: Why only at -17.5°C and -21.5°C? A plot showing the individual measured T-spectra would be helpful to show how relevant this increase is.
Line 340: quantify "clear nINP peak"
Line 344: Why was the last sample excluded?
Line 345: It is an often-misinterpretation of DeMott et al., 2010. The concentration of particles >0.5um are simply used to parameterize INPs of all size, not an actual size fraction of them.
Line 346: This is incorrect. Aerosol were not more ice active, there were only more INP.
Line 347-350: Speculative. Maybe the nINP is higher because more activity at the station towards the end of sampling.
Line 352: How was the statistical significance of a seasonal trend determined?
Line 353: Quantify "peaked mainly"
Line 355: Speculative. One order of magnitude scattering occurs also on short timescales.
Line 360-371: Explain the relevance of scavenging values for the interpretation of measurements here or delete these lines.
Line 372: Covariance with particle concentration was not shown. This could be an interesting addition to discuss the ice active particle fraction.
Line 374: Quantify "even more accentuated".
Line 375-376: Sentence fragment.
Line 376-383: Unclear what this discussion is aiming for. Clarify main point.
Line 385ff (Sec. 3.6) Converting nINP to ns doesn't yield new insights. As stated in line

379, INP are only a small fraction of total particles and the total surface area from all different particle types is not related to the number of INP. I recommend deleting the section and Fig. 4 and Fig.5. Instead include the ice active particle fraction at different temperatures and the spring-summer contrast.

Line 389: Repetition of line 72

Line 397: Explain why significance is not found for -16°C. Contradicting results in a narrow T-range could indicate that this analysis is not robust.

Line 399f: The difference indicates that there is no general trend.

Line 408: Quantify "substantial good agreement"

Line 409f: The aerosol population at GVB is a mixture of many particle types and only a tiny fraction acts as INP. Interpreting ns compared to ns from well constrained particle types is speculative.

Line 417: Show scatterplots of significant correlations in the supplement.

Line 421: Explain why and how the results are in line with what considerations.

Line 422: If these are general tendencies they should agree with the PM10 data as well. Explain why the analysis is limited to PM1.

Line 423: Quantify "Less clear"

Line 427: Explain why these elements are not good tracers for the soil type. What would be good tracers for the local mineralogy of the soil?

Line 415-429 (Sec. 3.7.1): Add a conclusion, lesson learned from this exercise.

Line 430-441 (Sec. 3.7.2) Suggest some arguments why a larger land fraction (residence time) of a trajectory should linearly correlate to the INP concentration at -15°C. Why not at lower temperatures as well? The distance of land contact to the receptor, time past, precipitation formation along the trajectory and source strength in different land locations should make a large difference.

Line 433: Figure S1 is more informative than Fig.6 to show the overpassed ground types. I suggest changing Fig.6 for Fig.S1. In addition, Fig. S1 is referred to more often than Fig. 6 later in the manuscript.

Line 436: A scatterplot showing fLand versus nINP instead of timeseries would be more helpful than Fig.S3 and Table 3, to show the influence of land sources.

Line 435-438: If a fLand effect is found at -15°C it could be evidence against biological INPs dominating nINP at this temperature.

Line 440: specify what the "outcome" is and provide an overall conclusion from Sec. 3.7.2.

Line 443-482 (Sec. 3.7.3) Specify that this analysis was performed using 14 datapoints from PM1 DFPC. It should be demonstrated that the limited dataset yields robust correlations with CHL. Show some scatterplots. I suggest to include high CHL regions as a ground condition, subdividing the sea category, and include it in the analysis of sec. 3.7.2.

Line 443-459: The hypothesis and description of how INP and CHL maps are correlated fits better to the Method section.

Line 451: Why are trajectories with land contact excluded and why only some? A short land contact can have a large impact on nINP.

Line 460: The time lag doesn't make sense to me. Why would the aerosol generating, biochemical process not change location in 6 days or 16 days? The movement of the surface water should be considered.

Line 464: justify why 6- and 16-day time-lag was selected

Line 477-478: Explain how this can be seen in Fig. 7c?

Line 479-482: Consistency is not obvious. There seems to be even more negative correlations. The pattern looks random. I would expect some high productive areas based on ocean currents and biological factors that do not change rapidly.

**Conclusion**
Avoid euphemistic language.
Line 484-489: It seems that the paper gains little by including the WT-CRAFT dataset. It is only marginally relevant to discuss seasonality in sec. 3.4. In all other sections it is only mentioned that the data agrees with what was seen from analysing DFPC data. It there was a dependency on ice nucleation mechanism (condensation, immersion) it seems not to make a difference on size and source of INP.
Line 490-493: More, larger INP in summer seems contradictory to the absence of a seasonal trend.
Line 497: inter-annual variability is a trivial statement. The question is how large the variation is and why it happens.
Line 499: Explain the importance of this study in detail.
Line 502-505: This is a weak conclusion. There is no reason mentioned to assume that only one source contributes INP at all temperatures in the spectra.
Line 506-508: The relation has not been proven without doubt. It is a speculative interpretation.

**Figures**
The provided figures do not support the content of the manuscript. Fig.2 for example can be deleted is only referred to in a side note and supplementary figures are referred to more often than the actual figures included to the manuscript.
Helpful figures could include:
1. Temperature spectra with all nINP measured with WT-CRAFT and all datapoints (PM1 and PM10) measured with DFPC, with colour code for DOY of measurement and 3 symbols to differentiate the techniques.
2. Timeseries of nINP data at -15C, -18C, -22C as function of day of year including the 12 data points from Ny-Alesund in Wex et al, 2019,
3. Scatterplot of PM1 vs. PM10 nINP measured with DFPC
4. Timeseries showing activated fraction of particles (nINP divided by number of particles in PM1 for DFPC, in PM10 for DFPC and CRAFT) at -15C, -18C, -22C.
5. Fig. S1

Fig.1: Showing the individual measurements would be more informative than only median, min, max. Please change the figure accordingly and indicate summer, spring, PM1, PM10 samples in different colours and symbols. Indicate the detection limits of the DFPC and WT-CRAFT.
Fig.2: Comparing it to Fig. 4a in Irish et al., 2017 did not make it obvious how it was adapted. What assumptions are made to overlap the two y-axes (INP in water and INP in air)? This figure can be deleted.
Fig.3: It needs to be specified how the measurement uncertainty is determined from the sample volume and the analysis. It appears several times throughout the paper and is important.
Fig. 4: It would be more informative to scale the nINP with the total aerosol number. This would show that INP are not from the bulk aerosol population but rare exceptions. Summer, spring and PM1, PM10 can be contrasted.
Fig.5: redundant to Fig.4 no new information in this figure. Remove.
Fig.6: The main message from this figure seems to be that sea ice is melting in summer. This is trivial.
Fig. 7: c) It seems the colour bar shows nINP because it is written on top of it. Replace and label colourbar with units. Add a minimal explanation what can be interpreted from the patterns.
Fig. S1: Define ground types in the figure caption.

Fig. S2: why are some points connected by lines and others not? Homogenize all precipitation scales and nINP at the same temperatures. -18°C and -22°C plot in first column are switched. Second column last plot DFPC instead of FPC.
Fig. S3: use same fLand scale for all DFPC and WT-CRAFT subfigures and same nINP for same temperatures.
Fig S4, S5: Include chlorophyll as fifth land type in S1. Remove figures.
Fig. S6: use same colour-scale range for all subfigures. Use Fig. S1 map design to facilitate comparison. It seems regions where more trajectory points (Fig.S1) pass, also show higher CWT. This points to a problem with the small dataset size for this analysis.

**Tables**

The robustness of correlations in Tables 2 a, b and 3 would be clearer when shown as scatterplots. Due to the small size and structure of the data used, the derived linear correlation coefficients might be strongly biased by few outlier data points and be therefore misleading.
Scatterplots help to visually judge correlations. Person's R is sensitive to the data distribution and the R value can be generally misleading. Scatterplots of fLand and nINP would be helpful to investigate these issues.

**Technical corrections**

Delete "apparently", "likely", "noteworthy", "worth highlighting" throughout the manuscript.
Line 73: icebreaker
Line 100, 168: km instead of Km
Line 104: Section instead of Par
Line 115: define TSP, define OD
Line 132, 142: per $m^3$ not per $m^{-3}$
Line 140: replace super-microliter with 3uL
Line 230: define $\bar{D}$
Line 239: nINP instead nIPN
Line 339: $p<0.05$ instead $p<0.5$

---

## Referee Comment (RC2) · Anonymous Referee #2 · 21 Oct 2020

**Review of "Condensation and immersion freezing Ice Nucleating Particle measurements at Ny-Ålesund (Svalbard) during 2018: evidence of multiple source contribution" by Rinaldi et al.**

General Comment:

In this manuscript the INP concentration in Ny-Ålesund (Svalbard) was evaluated in two different seasons and in two different ice nucleation modes, using two different offline techniques. Given the high importance of the Arctic and the low number of studies focusing on INPs, the present study is useful for the ice nucleation, aerosol, and cloud microphysics communities. Although the present results are very valuable, is not completely clear how this study differs from previous studies conducted at the same Arctic location. The manuscript is poorly written as the authors did not pay attention to several important details as listed below. I encourage the authors to improve the quality of the manuscript taking into account the Major, Minor, and Technical comments.

Major Comments:

1. I do not see a clear difference between the content of the Abstract and the Conclusions. These two sections need to be different with the Abstract including more concise and quantitative information.
2. The English needs to be significantly improved. The way the document is written makes it very difficult to read it in some parts. Also, the document seems disorganized, and therefore, I encourage the authors to improve this part. The authors need to be more precise, more quantitative, and improve the statistical analysis.
3. There is key information missing in the text (e.g., technical details, references, units, correlation coefficients, etc.). See below.
4. Lines 94-96: What is different or what is novel in the present study compared to Wex et al. (2019) and Hartmann et al. (2019)?

Minor Comments:

Line 43: Add a reference after "amplification".

Line 46: Add a reference after "detail". How about Abbatt et al. (2019)?

Line 53: "sufficient numbers". Please be clear.

Line 56: "transport dynamics". Please be clear.

Line 57: Add a reference after "budgets".

Line 69: Define "T".

Line 77: "tripling INP". Please be clear.

Line 88: "evidencing order of magnitude wise increase". Please be clear.

Line 89: Add a reference after "ice".

Line 99: "The aerosol sampling was performed at the Gruvebadet observatory". Add a map showing it.

Lines 110-111: "The sampling generally started in the morning, during the spring campaign, while it started typically in the afternoon during the summer campaign". What is the reason for this?

Lines 110-111: Add a Table with the details of each sample from both techniques (e.g., initial time, final time, date, etc.).

Lines 113-119: When where the samples collected with the WT-CRAFT?

Line 114: "0.2 μm pore size". Brand? Model?

Line 115: Define "TSP".

Lines 118-119: If the flow rate of the WT-CRAFT is 150 lpm and the flow rate for the DFPC is 38.3 lpm, why the samples from the former one was 4 days and for the later one just 3-4 hours?

Line 121: The authors need to provide a brief description of the method.

Lines 128-129: I would rather add a small paragraph indicating how good is the agreement of the DFPC data compared to other techniques.

Lines 141-142: "in a known volume". Specify the volume.

Line 141-142. This is not very clear.

Line 151: No parametrizations were derived in this study.

Line 152: "concentration of ice nucleating particles ($n$INP)". This was defined in Line 69.

Line 164: "1.95 g cm$^{-3}$". Add a reference.

Line 167: "air temperature, T". This was first used in Line 69.

Line 173: "GVB". Define it.

Line 173-174: "on filters collected". The authors need to be clear on what filters and how the particle were collected.

Line 177: "$C_2O_4^{-2}$". Fix it.

Line 179: "$Mg^{2+}$, $Ca^{2+}$". Fix it.

Line 211 and along the text: The authors used "at Ny-Ålesund", "GVB", and "Gruvebadet". Please be consistent.

Line 217-219: "$t$", "$L$", "$C_t$" and "$D_{ijt}$" should be in italics.

Lines 241-242: "The observed offset may derive from the different time resolutions of the sampling for INP analyses, as well as from uncertainties in sampling activities and/or measurement uncertainties". How about the particle size analyzed in both techniques? The pore size of the filters used is different in each technique.

Line 242: In the Hiranuma et al. (2015) paper the 2 techniques were no used.

Line 244: Line 43: Add a reference after "questionable".

Lines 250: Are you sure the authors clearly distinguished between the 2 modes in Wex et al. (2014)?

Line 252: "different aerosol types yielded different results". Again, is it not the size measured by both techniques? Depending on the aerosol type, their size distribution changes, and therefore, the particle collection efficiency of each technique.

Lines 257-259: "median 115". From Figure 1, the value seems to be close to 90 $m^{-3}$ instead of 115 $m^{-3}$.

Line 259: "33-135 (median 77), 18-107 (45) and 6-66 (20) $m^{-3}$". This is not shown in Figure 1. Please add them to the Figure.

Line 264: "24-9082". What is the reason of such large variability?

Line 274: "range 5-10, 10-30 and 30-70". Units are missing.

Lines 281-182: "we can conclude that the results of the present study are generally consistent with literature". Add literature data to Figure 1.

Line 282: Add Wex et al. (2019) data to Figure 1.

Line 294: "similar". By how much?

Lines 300-302: I don't get what is unique here.

Line 304: "shows the bimodal activation with a hump feature at T above -15 °C.". Really?

Lines 304-305: "may be due to marine biogenic aerosols". Are not you sampling in a marine environment? Why is this surprising?

Line 311: "suggesting that the dominant INP sources may be located at long distances". This is rather speculative. Make a link with the evidence you show later.

Lines 313-314: "likely resulting from the activation of local sources after snow and ice melting". Why coarse particles can be transported long distances in spring and not in summer.

Line 317: Mason et al. (2016) found a large contribution from the coarse particles with most of the samples collected in Alert during the spring.

Line 319: The Creamean et al. (2018) samples were collected in spring.

Lines 321-322: Should "$cm^{-3}$" be "$m^{-3}$"?

Line 325: "at lower temperatures". Should it be higher?

Line 332: Add a reference after "melting".

Line 354: "at GVB (2012)". Add the corresponding paper.

Line 360: Why wind direction was not included?

Line 361: "were often associated to a reduction". It is not very obvious from the Figure. This is a qualitative conclusion.

Line 361: "the exception of precipitation events". Add the $r^2$.

Lines 362-371: I don't think this is really necessary as it adds too little to the discussion and does not help at all to support the data.

Line 372: "covariate". Add the $r^2$.

Line 374: "more accentuated" and "significant correlations". Add the $r^2$.

Line 385: "showed a significant". Add the $r^2$.

Lines 388-389: "mainly related to long-range transport of anthropogenic aerosol particles from lower latitudes (Arctic haze)". No evidence provided.

Line 396: "for all the activation temperatures". Just 2 temperatures are shown in the Figure.

Lines 396-397: "with the exception of the coldest one (T = -25 °C)". This information is not provided.

Lines 398-399: "only a minority of samples (<50%)". 30-40% is a minority?

Lines 399-401: I don't get it.

Lines 409-410: "suggesting that the INP population over the Arctic in summer originates from a combination of mineral dust and marine aerosol particles". If I understood correctly, this figure contains the data from both summer and spring, therefore, such conclusion cannot be drawn from the data reported in the figure. Please separate the data sets into summer and spring.

Line 417: "*n*INPDFPC correlated". Add $r^2$ and *p*.

Line 422 and 424: "anticorrelation" and "significant". Add $r^2$.

Figure 2. It seems you are comparing apples with oranges. Why are the y axis scale different? This figure and its discussion seems to be useless.

Figure 4. The authors need to separate them between summer and spring.

Technical comments:

1. Line 62: Delete the "dot" after "temperatures".
2. Line 100, Line 168: "Km" should be "km".
3. Lines 116-117: "diaphragm pump". Please add the details about the pump.
4. Line 272: "immersion mode freezing". Be clear.
5. Line 273: ")." Fix it.
6. Line 280: "etc...". It should be one dot.
7. Line 282: "Both the datasets discussed". Be clear.
8. Line 287: "while both our datasets". Be clear.
9. Line 293: "ranged 0.4-15 and 2-40 m-3". At what temperatures?
10. Line 361: Figure S2 is called before Figure S1.
11. Line 436: Figure S3 is called before Figure S1.
12. The citing format is wrong and needs to be fixed (e.g. missing spaces between references and when multiple references are cite, they are not organized chronologically).
13. The format of the units if not uniform. For example in some cases the authors used "X ° C" but in other cases "X°C" is used. Please be consistent.
14. Figure S1. Change the color of the "snow lines" as it is not clearly distinguishable from the white background.
15. How much time passed from sampling until the actual INP analyses? Add this information to the text.
16. Figure S2. The nINP in the middle left panel should be in blue. The nINP in the bottom left panel should be in black. Add more details to the Figure caption.
17. Figures S2. I am not sure if it makes sense to correlate precipitation and INP concentration when using the WT-CRAFT based on the low time resolution i.e., 4 days.

---

## Author Comment (AC1) · 29 Mar 2021

*Response to Referee #1*

First of all, the authors thank the referee for submitting helpful and meaningful comments, which lead to improvements and clarifications within the manuscript.

Below, we provide our point-by-point responses. For clarity and easy visualization, the Referee's comments (**RC**) are shown from here on in black. The authors' responses (**AR**) are in blue color below each of the referee's statement. In addition to the responses to referees' comments, we further modified the manuscript to increase its clarity and readability. Abstract and conclusions were mostly rewritten. The Section on the ice nucleation active site density ($n_s$) was removed; $n_s$ was substituted by the Activate Fraction (AF) parameter in the discussion. The Results section was re-organized for major clarity and separated from the Discussion Section. All the changes can be checked in the track change version of the manuscript, where the new text is highlighted in yellow color. We introduce the revised materials in green color along/below each one of your response (otherwise directed to the Track Changes version manuscript). All references are available in the end of this AR document.

**RC:** *General comments*

Ambient measurements from Spitzbergen during spring and summer 2018 are reported and analysed. The dataset is a welcome addition to the growing body of ice nucleation concentration measurements from the Arctic region. For the analysis the authors correlate the measured INP concentrations to bulk particle properties, season, meteorology and airmass trajectory. (1) Unfortunately, the analysis is not well motivated by hypotheses, and the results not presented clearly. Additionally, I suspect that (2) the set of measurements is too small to perform a robust analysis and the found correlations might be random. By ignoring this, (3) the authors got mislead to overinterpretations and speculative conclusions. The interpretation of data (e.g. concerning INP size and land vs. marine contributions) agrees with previous studies and (4) no new insights are obtained. The one interesting finding is that there was no seasonal variation observed in 2018. The authors need to be more specific in their descriptions, quantitative within reason for the interpretation and visualize their findings clearer to turn this manuscript into a valuable contribution to the field of ambient ice nucleation measurements.

**AR:** The authors appreciate these general remarks and constructive criticisms regarding our manuscript by Referee #1. We found these as invaluable guidance. We believe that the hypotheses and analyses in the revised manuscript are robust and insightful. We have limited but very good data, which are statistically valid. We admit that we have made some insufficient discussions, leading some of our data interpretations in an original manuscript to be speculative. Based on the peer-review comments, we removed/modified them to motivate the research. To allay the reviewer's concerns and mitigate any misgivings, the authors have decided to change the title of manuscript to "**Ice-nucleating particle concentration measurements from Ny-Ålesund during the Arctic Spring-Summer in 2018**", reflecting our changes and articulate what is truly presented in the revised version paper. We have also revised our abstract as well as the conclusion to reflect all of our major revisions (please read the Track Changes version paper). Below, we provide our point-by-point responses in hopes of our manuscript being considered for another review by the reviewer.

*Specific comments*

Major changes are suggested for each section. Minor corrections or clarification requests at specific line numbers are listed below.

**RC:** **Structure**

The manuscript could be structured better. Adding a Discussion section instead of including the discussion with the results would help the organization of the paper. Currently the Result section has mixed-in discussion, interpretation, literature comparison and some method description. Isolating the Results, visualizing and explaining them more specifically would be helpful to judge the interpretation.

**AR:** We thank the reviewer for the suggestion. We added a Discussion Section in order to improve the manuscript organization and clarity. The Results section now focus more on describing our data analysis and observation. In our new Discussion Section, we detailed: Interpretation of $n\text{INP}_{\text{DFPC}}$ and $n\text{INP}_{\text{WT-CRAFT}}$ discrepancy, Interpretation of seasonal variability of $n\text{INP}$, Sources of INPs in Ny-Ålesund etc.

*RC:* **Citations**

A high number of citations are given, but it is often unspecific what information can be found in which citation or why several citations are listed. Better integration in the text would be helpful instead listing multiple citations at the end of a sentence.

*AR:* The authors agreed and kept only most relevant references in the revised manuscript.

*RC:* **Title**

The title does not fit the manuscript. There was never doubt that multiple sources contribute INP at different temperatures. Something along the line of "Concentration of ice nucleating particles measured at Ny-Ålesund during 2018" would be more accurate.

*AR:* This is a good suggestion. The authors took the referee's word for it, and our title now reads, "Ice-nucleating particle concentration measurements from Ny-Ålesund during the Arctic Spring-Summer in 2018."

*RC:* **Abstract**

The abstract uses wordy, euphemistic language. In the interest of clarity, the tone should be revised to be strait forward. At the moment, the abstract consists of too many unsupported statements and vague conclusions that are incomprehensive before reading the manuscript in detail.

*AR:* The authors concur. By incorporating with all review comments, the authors offer concise and straightforward abstract by revising almost entire abstract. Please see the Track Changes version of the manuscript.

*RC:* Line 15: INP sources, INP concentrations and ice nucleation properties should be distinguished clearly. They are not the same.

*AR:* The authors agree. We now carefully distinguished INP source, aerosol particle source, INP concentration ($n$INP) and INP properties throughout the manuscript.

*RC:* Line 15: INP sources are unknown not "inadequately understood"

*AR:* The authors decided to remove this sentence and offer more detailed discussion in the main manuscript Sect. 4.

*RC:* Line 18: specify what properties were characterized

*AR:* We meant INP concentrations ($n$INP), and the text is updated accordingly.

*RC:* Line 22: why are the temperatures (-15°C, -18°C, -22°C) probed with the DFPC not evenly spaced?

*AR:* The temperature ($T$ hereafter) boundaries of DFPC are determined (limited) by the device operational capability (Rinaldi et al., 2017; Rinaldi et al., 2019). The upper and lower boundary $T$s of -15 and -22°C are the highest and lowest $T$s we can be confident for. The $T$ point of -18°C was chosen as an arbitrary intermediate step between the two boundary temperatures. We find no scientific reason to consider evenly spaced ice activation temperatures. We rephrased this $T$ range to "temperatures ($T$s) of -15 to -22 °C".

*RC:* Line 24-25: The dependence on ice nucleation mode is not investigated in the manuscript and a sampling issues is a more reasonable cause for the discrepancy.

*AR:* We respectfully disagree. Perhaps, our inadequate explanation of sampling method has misread the referee. The authors clarified that the flow of 150 lpm represents a total flow through a common TSP inlet, but a diverged flow to our polycarbonate filter sampler was only on average 5.4 lpm in Sect. 2.1. We will address potential sampling issues below in this document.

*RC:* Line 26: specify how the increase in coarse INP was observed.

*AR:* The text was modified as follows: "DFPC measured $n$INPs for a set of filters collected through two size-segregated inlets: one for transmitting particulate matter less than 1 μm (PM$_1$) and another for that of less than 10 μm aerodynamic diameter (PM$_{10}$). Overall, $n$INP$_{PM10}$ measured by DFPC ranged from 3 to 185 m$^{-3}$ at temperatures ($T$s) of -15 to -22°C. On average, the super-micrometer INP ($n$INP$_{PM10}$ - $n$INP$_{PM1}$) accounted for

approximately 20-30% of $n$INP$_{PM10}$ in spring and increased markedly in summer. In particular, it contributed 45% at $T$ of -22°C and 65% at $T$ of -15°C). This increase trend of super-micron INP fraction towards summer suggests an important role of super-micrometer aerosol particles as the source of Arctic INPs".

*RC:* Line 27: Explain why increase in coarse INP fraction suggests local sources.
*AR:* The authors decided to exclude this part from our abstract and provide detailed discussion in Sects. 3.2 and 3.6 (please see the Track Changes version paper).

*RC:* Line 28: Speculative. INP active at -15°C are not exclusively biological particles. The source of the particles active at -15°C in this study are unknown.
*AR:* The referee is right. We decided to exclude this sentence and to avoid similar sentences throughout the manuscript.

*RC:* Line 30: specify "distinct behaviours of particles"
*AR:* We admit that this part sounds confusing. Upon reformulation of the manuscript, this part became unnecessary. We have deleted this part from abstract.

*RC:* Line 31: How was the inter-annual variability of local INP sources previously considered? Specify the evidence for an inter-annual variability based on the current dataset and all available data from Spitzbergen. Consider that the 2012 data set is only 12 measurements and not a strong dataset to compare to.
*AR:* The authors realize that these information fit better in the main manuscript, and we address it in Sects. 3.4 and 4 (please see the Track Changes version paper).

*RC:* Line 33: It seems trivial that the INP population can be contributed by terrestrial and marine sources on an island.
*AR:* The authors agree. We reduced the discussion of it in our abstract.

*RC:* Line 35: specify nucleation ability. Higher particle load can explain higher INP concentrations without increased ice nucleation activity.
*AR:* Ice nucleation efficiency would be more appropriate word to be used. We decided to discuss this in detail in Sect. 3.3 and 3.4, and thus removed this part from our abstract.

*RC:* Line 37-40: requires reading the paper to understand this description.
*AR:* We agree. We now introduced only general remarks, for which the reader does not need to refer to our main manuscript: "Our spatiotemporal analyses of satellite retrieved Chlorophyll-a as well as spatial source attribution indicates the maritime INPs are expected at GVB from the seawaters surrounding the Svalbard archipelago and/or close to Greenland and Iceland".

*RC:* **Introduction**
The introduction is not tailored enough to the subject of this study. To make the introduction more effective at explaining and putting into perspective what follows, I suggest to focus on: ice nucleation mechanisms (condensation and immersion mode), INP-cloud interaction in the Arctic without going into detail on radiative effects, previous INP measurements in the Arctic and what has been learned about potential sources, tends, dependencies of nINP in the Arctic. A lot of literature is currently discussed in the Result section. Better to include a concise discussion of relevant literature to the introduction to develop the hypotheses which are then addressed in this study. Currently literature review is used inefficiently in the Result section to point to similar conclusions found elsewhere in literature.
*AR:* Ice nucleation mechanisms are now discussed in Lines 54-61. The Arctic INP-cloud interaction and its importance is briefly introduced by citing Murray et al. (2021). Previous studies of $n$INP are now summarized in a tabular form (Table 1). Most articles used in the result section are now merged in the Introduction section as suggested. We clarified our study motivation at the end of the Introduction Section (Lines 99-108): "In the present study, we contribute to fill the present gap of INP observations in the Arctic

environment, investigating *n*INP and potential sources at the ground level site of GVB (Svalbard), through spring and summer time measurements, by two INP quantification techniques, representing immersion and condensation freezing. We hypothesized that the *n*INP variability at a single *T* can be explained by differences in freezing modes. Recent modeling simulation and remote sensing studies suggest immersion freezing is the most relevant heterogeneous ice nucleation mechanism in mixed-phase clouds, which are prevalent in the Arctic (Hande and Hoose, 2017; Westbrook and Illingworth, 2011). The key to verify this in the Atlantic sector of the Arctic depends on a multitude of ambient INP measurements with a combination of trustful INP measuring systems at wide heterogeneous ice-nucleating conditions. Finally, we investigate the ice nucleation efficiency of Arctic aerosol particles represented by the activated fraction (AF), which provides further insight into the seasonal trend of ice nucleation efficiency besides concentration data".

**RC:** Line 46: provide reference and explain how aerosol affect cloud properties
**AR:** Murray et al., 2021 has been added to concisely direct the reader to the negative cloud-phase feedback in the Arctic.

**RC:** Line49-57: specify how the numerous local processes and feedbacks interact to affect structure, phase and persistence of clouds in the Arctic. Beyond what is generally true for INP-cloud interactions, explain why Arctic clouds are sensitive to INP concentrations. Provide a reference for the uncertainty associated with nINP, eg. DeBoer et al., 2018.
**AR:** This paragraph is now completely revised with only relevant references. Please see the Track Changes version of the manuscript.

**RC:** Line 59: the references are not "recently"
**AR:** The authors agree. We rephrased this sentence to: "Sea water has been identified to be a source of ice active organic matters (Knopf et al., 2011; Wang et al., 2015; Wilson et al., 2015), which are transferable to the atmosphere within sea spray particles (e.g., McCluskey et al., 2017)."

**RC:** Line 61-62: specify "most" ice nucleation processes. It would be good to introduce ice nucleation mechanism in more detail. Line 63-64: It is not generally true that biogenic INP nucleate ice at temperatures above -15°C. It is a bit of a stretch from ice nucleation properties to rain.
**AR:** With additional references, we have rephrased this sentence to: "Mineral particles are dominant immersion and condensation mode INPs typically below -20°C according to Fig. 13 in Hoose and Mohler (2012), with an exception of K-feldspar, which facilitates ice nucleation at much higher *T*s when compared to other mineral compositions (Atkinson et al., 2013). Further, biogenic INPs tend to support formation of ice at *T*s relatively higher than abiotic INPs (Murray et al., 2012), even though there is a considerable variation in ice nucleation efficiency within biotic INPs (Kanji et al., 2017)", to clarify our points. In addition, ice nucleation mechanisms are now introduced in the third paragraph of the Introduction section.

**RC:** Provide a more coherent explanation and provide the link to INP in the Arctic region.
**AR:** This is a good suggestion. We now compiled 14 previous Arctic *n*INP studies in Table 1 and associated text (Lines 70-98).

**RC:** Line 67-86: Specify that the literature review is separated into condensation and immersion mode measurements as well as separated into short and year around observations.
**AR:** This is also a good suggestion. Please see our new Table 1. We have enlisted previous literature according to condensation vs. immersion.

**RC:** Line 67: specify "short periods of time"
**AR:** All study time periods are now listed in Table 1. As seen, it is typically in the order of several months. As these time spans vary from study to study, we decided not to use the word of short periods of time in text. Thanks for catching this.

*RC:* Line 72: add some information how Hartmann et al. confirmed this.
*AR:* The authors decided to conduct detailed discussion of Hartmann et al. in Sect. 4 (Discussion), Lines 580-581.

*RC:* Line 74: How did Bigg et al. identify the Ocean was the main source? There was a third Arctic cruise in 2001 and a more recent expedition in 2017. They are reported in Welti et al., 2020.
*AR*: The authors added the requested details (Lines 76-77). Welti et al., 2020 is now included in the new Table 1.

*RC:* Line 76-86: Mention at what temperatures Mason et al., 2016, Si et al., 2018, Creamean et al., 2018, Irish et al., 2019 reported data. This section seems to contradict line 61-64 where it is argued that INP active at T>-15°C are biological and not mineral dust.
*AR:* The requested information was added in the new Table 1. We have revised any unappropriated statement regarding biological particles.

*RC:* Line 88: Quantify the increase. Contrast to the fact that often ambient $n$INP measurements scatter within 1-2 order of magnitude within less than a day, highlighting that caution should be used when interpreting variations smaller than one order of magnitude in such dataset.
*AR:* The authors thank the referee for sharing thoughts, and we agree. An order magnitude discrepancy is not by all means an acceptable margin or any sort of magic numbers. We have carefully removed all of the one order of magnitude discussions from the manuscript.

*RC:* Line 90: A time-series showing measured nINP at -15°C as function of DOY from all the listed measurements would be a helpful addition to illustrate the discussion and to show if the Arctic region as a whole experiences seasonal variations or if these are local phenomena.
*AR:* We thank the reviewer, but we believe that we made sufficient discussion in the revised manuscript in comparison to Wex et al. (2019). Specifically, the authors have provided a comparison with the seasonal evolution of $n$INP presented by Wex et al. (2019) at *Ts* of -15 and -18°C in the SI (Fig. S3).

*RC:* Line 94: As your literature review exemplifies, there is no general "gap" of INP measurements in the Arctic.
*AR:* We agree and disagree. It is deemed to be inconclusive yet at least. In any case, our $n$INP data for multi-seasons along with a rich set of baseline data from the GVB station (including but not limited to the dataset presented in this work) would be crucial for future verification of more rigorous modeling closure study to examine temporal trends of the Arctic $n$INP. As presented in our work, we find both agreement and disagreement between our $n$INPs from this study and $n$INPs measured in previous studies. Filling that gap completely is beyond the scope of the current work. However, it is an imperative future task, and we believe that our findings of condensation vs. immersion in the Arctic $n$INPs as well as non-substantial seasonal variability in $n$INP are invaluable to report.

*RC:* **Methods**
The method of how INP concentrations are determined with the WT-CRAFT starting from the air volume sampled through filter to counting the number of frozen aliquots, should be explained in more detail, focusing on how nINP can be derived step by step from sample volume, water volume, droplet volume. How the preparation is done practically is of secondary interest.
*AR:* This is a valid question. The authors initially intended to include some of these details (omitted concerning the manuscript length). All details regarding WT-CRAFT are now incorporated in our revised manuscript. The authors should have clarified that only a subset of 150 lpm from the common inlet was directed towards our WT-CRAFT filter sampler. We hope these array the referee's misgivings.
Lines 127-137: "Aerosol particles were collected using 47 mm membrane filters (Whatman, Track-Etched Membranes, 0.2 µm pore). Briefly, aerosol particle-laden air was drawn from a central total suspended particulate (TSP) inlet with a constant average inlet flow of 5.4 lpm (± 0.2 lpm standard deviation). We note

that the TSP inlet is custom made, and is designed to operate with isokinetic and laminar flow at 150 lpm. From the central inlet, an 8 mm outside diameter stainless steel tube was directly connected to the filter sampler to intake a subset of air flow. More detailed conditions of our filter sampling, including sampling time stamps, air volume sampled through filter cross section, and the resulting HPLC water volume used to suspend aerosol particles for WT-CRAFT analysis, are summarized in Table S1. Below the filter sampler, the filtered-air was constantly pumped through a diaphragm pump (KnF, IP20-T). A critical orifice was installed upstream of the pump to ensure a constant volume flow rate and control the mass flow rate through the sampling line. A typical sampling interval was approximately 4 days with only one exception (i.e., 8 days for the sample collected starting on 26 May 2018)".

**Table S1**. Summary of sampling conditions for filters collected for WT-CRAFT.

| Sample ID | Filter Sampling Ref Start Time | Filter Sampling Ref End Time | Flow Rate | Total Flow (optimized for 50% of filter) | Suspension water volume (First frozen drop = 0.001 INP L$^{-1}$) |
|---|---|---|---|---|---|
| | DAT_UTC | DAT_UTC | LPM | L | mL |
| NYA_GVB_01 | 4/16/2018 17:00 | 4/20/2018 10:00 | 5.1 | 13617.0 | 2.8 |
| NYA_GVB_02 | 4/20/2018 14:40 | 4/24/2018 14:40 | 5.1 | 14601.6 | 3.1 |
| NYA_GVB_03 | 4/24/2018 18:20 | 4/28/2018 16:00 | 5.5 | 15314.5 | 3.1 |
| NYA_GVB_04 | 4/29/2018 13:30 | 5/2/2018 16:15 | 5.6 | 12445.9 | 2.4 |
| NYA_GVB_05 | 5/2/2018 16:20 | 5/6/2018 14:37 | 5.5 | 15471.9 | 3.3 |
| NYA_GVB_06 | 5/6/2018 14:45 | 5/10/2018 13:00 | 5.4 | 15325.1 | 3.3 |
| NYA_GVB_07 | 5/10/2018 13:10 | 5/14/2018 11:05 | 5.6 | 15890.7 | 3.3 |
| NYA_GVB_08 | 5/14/2018 11:15 | 5/18/2018 7:50 | 5.5 | 15179.0 | 3.1 |
| NYA_GVB_09 | 5/18/2018 8:00 | 5/22/2018 8:28 | 5.5 | 15917.0 | 3.3 |
| NYA_GVB_10 | 5/22/2018 8:30 | 5/26/2018 11:33 | 4.8 | 14263.2 | 3.0 |
| NYA_GVB_21 | 5/26/2018 11:45 | 6/3/2018 18:30 | 5.5 | 32883.2 | 6.9 |
| NYA_GVB_22 | 6/3/2018 18:35 | 6/7/2018 17:20 | 5.5 | 15576.9 | 3.3 |
| NYA_GVB_23 | 6/7/2018 17:24 | 6/11/2018 17:35 | 5.4 | 15668.3 | 3.3 |
| NYA_GVB_24 | 6/11/2018 17:40 | 6/15/2018 16:24 | 5.4 | 15218.9 | 3.2 |
| NYA_GVB_25 | 6/15/2018 16:28 | 6/19/2018 19:05 | 5.4 | 15887.1 | 3.3 |
| NYA_GVB_26 | 6/19/2018 19:09 | 6/23/2018 19:16 | 5.3 | 15152.8 | 3.2 |
| NYA_GVB_27 | 6/23/2018 19:20 | 6/27/2018 13:55 | 5.3 | 14525.0 | 3.0 |
| NYA_GVB_28 | 6/27/2018 14:00 | 7/1/2018 16:40 | 5.4 | 16013.6 | 3.3 |
| NYA_GVB_16 | 7/1/2018 16:50 | 7/5/2018 17:20 | 5.5 | 15792.2 | 3.3 |
| NYA_GVB_17 | 7/5/2018 17:25 | 7/9/2018 17:22 | 5.4 | 15587.1 | 3.3 |
| NYA_GVB_18 | 7/9/2018 17:27 | 7/13/2018 18:24 | 5.4 | 15662.3 | 3.3 |
| NYA_GVB_19 | 7/13/2018 18:33 | 7/17/2018 16:43 | 5.3 | 15071.4 | 3.2 |
| NYA_GVB_20 | 7/17/2018 16:52 | 7/21/2018 15:55 | 5.3 | 15241.3 | 3.2 |
| NYA_GVB_11 | 7/21/2018 16:02 | 7/25/2018 16:31 | 5.4 | 15602.3 | 3.3 |
| NYA_GVB_12 | 7/25/2018 16:38 | 7/29/2018 15:07 | 5.4 | 15391.3 | 3.2 |
| NYA_GVB_13 | 7/29/2018 15:14 | 8/2/2018 18:39 | 5.5 | 16254.6 | 3.4 |
| NYA_GVB_14 | 8/7/2018 15:55 | 8/11/2018 14:05 | 5.4 | 15382.1 | 3.2 |
| NYA_GVB_15 | 8/11/2018 14:12 | 8/15/2018 17:36 | 5.4 | 16177.4 | 3.4 |

*RC:* In section 2.3 it should be clarified (by a short explanation at the beginning of each subsection) for what purpose the measurement or analysis is performed or used in the context of this paper. Here it would be helpful to already know from the introduction what the aim of the analysis is or what hypotheses are going to be tested with these data. Give context in the introduction section: why are you investigating ground type, trajectories, chlorophyll,…

*AR:* We thank the reviewer for the suggestion. We revised the manuscript accordingly. For instance, Sect. 2.3.4 now starts with "In order to investigate the sources that contributed to INPs (i.e., maritime vs. terrestrial), we performed the 5-day back trajectory analysis,…". Sect. 2.3.5 now starts with "Satellite retrieved chlorophyll-a fields were used to track the evolution of oceanic biological activity in the Arctic ocean during the study period".

*RC:* Line 99: Refer to the location as Gruvebadet station throughout the paper and introduce the abbreviation (GVB) here.

*AR:* Introduced. Thanks.

*RC:* Line 100: Point to fig. S1 showing the location on a map.
*AR:* This is a valid suggestion. A new Fig. 1 is produced, and the GVB location is now shown.

*RC:* Line 102: Can Longyearbyen in the SE of the GVB station contribute to the aerosol population?
*AR:* The distance of the sampling site to Longyearbyen, the main settlement of Svalbard with about 2000 people living and working there, is more than 100 km. Dekhtyareva et al (2016) investigated the potential impact of the activities taking place there on the measurements in Ny-Ålesund. Using 3D backward trajectories, they found that during spring NOx values were higher for long range transport cases (defined as those in which the airmasses come from latitudes below 70°N). During summer, mainly due to meteorological patterns, they concluded that it is unlikely that the pollution generated in Longyearbyen reaches Ny-Ålesund. Based on these remarks and considering the distance and the small dimension of the settlement, we would not expect that substantial contributions of this point source in our measurements at GVB.

*RC:* Line 110: Is filter overloading an issue in the clean Arctic air? The WT-CRAFT filters were sampled longer, with higher flow and on filter with smaller pore size. Where these filters potentially overloaded?
*AR:* Filter overloading is an issue only for the DFPC technique. DFPC analyzes aerosol particles collected on filters directly; therefore, it was necessary to avoid coalescence of ice crystals while processing condensation freezing experiments. For these reasons, the upper limit of sampled volume through the DFPC filter cross section was optimized to comply with the INP quantification range of ca. 50-1500 INPs per filter.
This overloading concern is not an issue for immersion freezing measurements. If necessary, dilutions of stock suspensions (i.e., aerosol particles suspended in HPLC water) can be assessed. Furthermore, we monitored the flow passing through the cross section of the WT-CRAFT filter while sampling. Between the beginning and the end of each sampling, the flow deviation was <5% for individual samples. With typically <100 p/ccm particle load and ~5.4 lpm of sampling flow rate (See revised texts in Sect. 2.1), we do not expect any particle overloading conditions.

*RC:* Line 115: Pumping 150lpm through filter with 0.2um pore size creates a huge pressure drop. Can you comment on how sampling was possible without fracturing the filter? How was the flow monitored? In this setup the filter probably acted as flow limiter rather than the critical orifice before the pump. An overestimation of the sample flow would explain the offset between WT-CRAFT and DFPC.
*AR:* For clarity, 150 LPM is the flow rate of the central laminar flow sampling inlet at GVB. From the central inlet, only a small amount of flow (~5.4 lpm) was bypassed to the WT-CRAFT filter sampler. The text has been substantially modified to clarify what was truly done at GVB in Sect. 2.1. Please see our Track-Changed manuscript.

*RC:* Line 116: specify pump model
*AR:* Specified.

*RC:* Line 118: Is filter overloading (line 110) an issue for 4-day samples? The volume sampled is more than 100-times larger for WT-CRAFT than for the DFPC filters.
*AR:* Clarified above, and all total sampled air volume for each WT-CRAFT sample is provided in Table S1.

*RC:* Line 127: Specify how uncertainties in T and Sw convert into uncertainties in nINP.
*AR:* We considered temperature uncertainties of 0.2°C and 0.1°C for air and filter, respectively. These uncertainties determines an uncertainty of 0.02 on the calculated $S_w$. We evaluated the effect of such variation in $S_w$ on the final number of counted INPs by extrapolating the results of $n$INP as a function of $S_w$, obtained by Belosi et al. (2018) for different aerosol particles.
The text was modified accordingly (Lines 151-153).

*RC:* Line 128f: Has a systematic difference between condensation and immersion mode ice nucleation been observed in these inter-comparisons?

*AR:* This is a valid question. The authors now clarified the raised point in our revised Sect. 4. Please, refer to sub-Sect. 4.1 for details.

*RC:* Line 132: The large sample volume of over 800m3 would allow to detect approximately 100-times lower INP concentrations than 1 m-3. Why was the analysis not performed in the full range?
*AR:* As clarified above, the sampled air volume is much smaller than the said number (see Table S1).
The WT-CRAFT measurement was not conducted below -25°C because we observed that non-negligible amount of field blank and HPLC-grade pure water droplets (>3% of 70 droplets) could freeze at below -25 °C for this study. WT-CRAFT was operated inside of the ventilated fume hood with air flow filtered by HEPA. This effort is to follow the setup of the original NIPR-CRAFT, which is used in a clean booth. Regardless of similar experimental procedures used in both CRAFT systems, this limitation of measureable temperature > -25 °C persisted for WT-CRAFT. While the reason of this limitation is unknown, more insightful description of WT-CRAFT and its capabilities are now available in Vepuri et al. (2021).

*RC:* Line 133: State how the two CRAFT systems are different. All I could find in Hiranuma et al., 2019 was that they used different sizes of droplets. This is not an instrumental difference.
*AR:* This is a valid question. Cont'd on our previous response;
**Camera:** We employ a combination of an Opti-Tekscope OT-M HDMI microscope camera and a Logitech c270 camera to correctly capture the transition of droplet brightness/contrast to opaque ice with 30 fps time resolution with a reasonable pixel resolution as well as magnification (if needed).
**Droplet holding plate:** We use a thin (<5 mm) polished aluminum plate to warrant an efficient thermal cooling and to make sure the Cryo-cooler system temperature is equivalent to the temperature measured at the surface of the plate within known uncertainties.
**Isolation to the lab air**: WT-CRAFT is operated in a vertical clean bench (LABCONCO, Purifier®). All droplet preparations (70 x 3µL) were conducted in the clean bench to minimize the chance of contamination from the lab air.
We have clarified these in our revised Sect. 2.2.2. Please see the track change manuscript.

*RC:* Line 135: Explain how the uncertainty in ice nucleation efficiency is derived.
*AR:* The uncertainties of temperature, ± 0.5 °C, stems from a sensor manufacturer reported uncertainty (TGK, SN-170N) The uncertainty in ice nucleation efficiency in WT-CRAFT are and ±23.5% according to Hiranuma et al. (2019, i.e., Table S2). Note that our ice nucleation uncertainty was estimated based on the average standard deviation across the examined temperature ($T$ > -25 °C) for known composition (microcrystalline cellulose), which reasonably matches with 95% confidence intervals of individual measurements (i.e., Eqn. 3.21 of Schiebel, 2017).

*RC:* Line 136: repetition, delete
*AR:* Deleted. Thanks for catching this.

*RC:* Line 139: provide camera model specifics
*AR:* Provided and specified.

*RC:* Line 140: define how INP concentrations are derived.
*AR:* Defined in **Section 2.2.3** as follows:
For the WT-CRAFT analysis, we first computed the $C_{INP}(T)$ value, which is the nucleus concentration in HPLC suspension (L$^{-1}$ water) at a given $T$ as described in Vali (1971). This $C_{INP}(T)$ value was calculated as a function of unfrozen fraction, $f_{unfrozen}(T)$ (i.e., the ratio of number of droplets unfrozen to the total number of droplets) as:

$$C_{INP}(T) = -\frac{\ln(f_{\text{unfrozen}}(T))}{V_d} \qquad (1)$$

in which, $V_d$ is the volume of individual droplets (3 µL). Next, we converted $C_{INP}(T)$ to $n$INP($T$). The

cumulative $n$INP per unit volume of sample air, described in DeMott et al. (2017), was estimated as:

$$nINP(T) = C_{INP}(T) \times DF \times \frac{V_l}{V_{air}}$$
(2)

where DF is a serial dilution factor (e.g., DF = 1 or 10 or 100 and so on). The sampled air volume ($V_{air}$) and the suspension volume ($V_l$) are now provided in Table S1.

**RC:** Line 141f: specify water volumes, for 1 INP per m3 that would be 90mL for 4 day samples and 180mL for 8 day samples.
**AR:** Now given in **Table S1**.

**RC:** Line 143f: specify water volume used for soaking
**AR:** Now given in **Table S1**.

**RC:** Line 144: specify how mechanical vibration was applied. By sonication?
**AR:** We soaked each polycarbonate filter in a sterilized falcon tube with HPLC water on the VWR vortex mixer. This point is now clarified in the main text. No sonication was applied not to damage polycarbonate filter as well as particles suspended in water.

**RC:** Line 145: How were droplets prepared?
**AR:** Manual pipetting in a clean bench.

**RC:** Line 146: The method by how much the sample was diluted is not explained clearly. Specify the dilution water volume and how dilution was considered for the derivation of INP concentrations.
**AR:** Simple serial dilution (x10 and/or x100) as described in Vepuri et al. (2021) was applied. The derivation of INP concentrations including dilution factor is now explained in Sect. 2.2.3.

**RC:** Line 149: How was the stitching performed? At what temperature were the spectra stitched?
**AR:** The IN measurements from the undiluted and diluted runs were merged by taking the lower $n_{INP}$ values, which exhibit smaller CL95% error, for the overlapped $T$ region (Vepuri et al., 2021). This procedure was employed not to have erroneous jumps – we also attempted to take median or max numbers for overlapping regions, but we found that the proposed procedure gives the least stair case like spectrum. Stitching spectra does not depend on the temperature. Instead, we merge our spectra of an original stock and diluted suspension in the way we make sure the following three criteria are met:
(1). Gap is within a factor of few
(2). Two spectra match within CI95%, within T uncertainty or a combination of both
(3). Two spectra match within 23.5%, within T uncertainty or a combination of both
We now cite Vepuri et al. (2021) in our manuscript.

**RC:** In Fig.1 a jump in max concentration appears at -23°C. Is this the range of the diluted measurements? To show the "absence of failure" of this technique, it would be helpful to show the individual measurements in Fig.1 and not just the range.
**AR:** With the procedure explained above for merging spectra, we merged all spectra in a consistent manner. As per request of the review, we show individual spectra for the reviewer mentioned. The authors respectfully wish that the reviewer finds our procedure is reasonable.

[Figure]

RC: Line 153f: Derivation of nINP must be defined clearer. Dividing the number of INP by the total sample volume is incorrect. The concentration is calculated from the filtered air volume, dilution water volume, droplet volume and number.
AR: We clarified this point now in Sect. 2.2.3.

RC: Line 154: As you state on line 379, only a small, T-dependent fraction of ambient aerosol are INP. Therefore, dividing the number of INP by the bulk particle surface has no physical meaning for a heterogeneous aerosol population. I recommend changing this approach to deriving only the fraction of particles that are ice active, by dividing the INP concentration by the particle concentration.
AR: We thank the reviewer for the suggestion. We have decided to follow the reviewer suggestion, limiting to present and discuss activated fraction (AF) data - see our revised Sect. 3.3 and 3.4. Our conclusions did not change.
In the future, long-term nINP monitoring by an online instrument (e.g., Möhler et al., 2021) may allow the authors to further evaluate ice nucleation efficiency of the Arctic aerosol particles.

RC: Line 161: specify APS measurement range
AR: The APS measurements range is from 0.5 to 20 micrometers. We modified the text as follows:
"An Aerodynamic Particle Sizer (APS) model TSI 3321 for the diameters between 0.5 and 20 micrometers."

RC: Line 163: what are the references pointing at? How were the size distributions averaged for the sampling interval of the filters?
AR: The citations refer to papers reporting more details on the cited instruments. The average aerosol number concentration, for each INP sample, was obtained by averaging all the aerosol number concentration data points falling within the filter sampling interval. The text was modified adding this detail (Lines 215-219):
"The number concentration in the resulting overlapping range was taken from the SMPS data as SMPS provides more size bins. At the end, commutative aerosol particle counts of SMPS and APS were considered as a total aerosol particle number concertation. To compare with nINP and to calculate the AF, the particle number concentrations at 10 minutes time resolution were averaged over each filter sampling period".

RC: Line 164: What substance is assumed for a density of 1.95 gcm-3? Mineral dust and sea salt have higher densities.
AR: We considered a mixture of different substances, including lighter compounds like methanesulfonic acid and nss-Sulphates, besides sea-salt and dust. Based on the findings from Lisok et al 2016 on the chemical characterization of the aerosol at the same site, we estimated the value of 1.95 gcm$^{-3}$. We changed the sentence as follows;
"The aerodynamic diameters measured by the APS were corrected to the volume equivalent diameters using an average particle mass density equal to 1.95 g cm-3, assuming a mixture of different substances based on

the findings from Lisok et al. (2016) and a dynamic shape factor of 1. The number concentration in the resulting overlapping range was taken equal to that from the SMPS".

*RC:* Line 173: Where was this analysis performed? While handling during the analysis is relevant, handling before and after sampling, storage and transport are equally important and could be described.
*AR:* The authors clarified these in Lines 228-230: "The filters were handled with care (working under a class 100 laminar flow hood by personnel wearing powder free latex gloves to minimize potential contamination) throughout the sampling and offline analysis at the University of Florence. After sampling the filters were stored and shipped at -20°C".

*RC:* Line 181-196 (Section 2.3.4.): Very similar to the text in Wex et al., 2019. Sentences in line 191-195 are copied from Sec. 2.7 in Wex et al., 2019. It is difficult to understand without consulting the original description. Section 2.3.4. should be rewritten entirely, explaining more clearly how ground types were categorized and how trajectories were merged to the filter sampling intervals. I suggest (instead of the analysis in 2.3.5.) to include high chlorophyll concentration as a fifth ground type in this analysis. Additionally, precipitation along the trajectories should be considered.
*AR:* The authors apologize for extending the assessment to comment towards raised questions. The relevant text has been substantially updated in Sect. 2.3.4. Please refer to the track change manuscript.
While the introduction of the high CHL class may be valid and one way for analysis, we took an alternative approach following previously published articles as these have been well-established (Rinaldi et al., 2013; O'Dowd et al., 2015; Mansour et al., 2020a; Mansour et al., 2020b). The approach proposed by the reviewer would presume that we can define a CHL threshold associated to emission of INPs. We do not have such a knowledge of the biological processes leading to production of marine INPs. On the contrary, working on the correlation evidences eventually present relations between phytoplankton activity and INP concentration, without arbitrary assumptions. One caveat is that we unfortunately cannot incorporate with the occurrence of precipitations along the considered BTs in our model.

*RC:* Line 201: state the temporal resolution of the dataset.
*AR:* Stated as: "The Level-4 product is available globally at ~4 km spatial resolution and daily time resolution."

*RC:* Line 202: Shift the description of how INP concentrations and chlorophyll maps were merged from the Result section to here. The DFPC summer data consists of only 17 measurement days and 3 are excluded because land influence, leaving 14 data points. Demonstrate that correlations are robust by showing some scatterplots of grid cells with a strong correlation as a supplement.
*AR:* We thank the reviewer for the suggestion, but we believe that this explanation fits in this particular part and increases a clarity of logical flow.
It is impossible to check visually all the regressions that form the correlation maps discussed in the manuscript as each map is composed of 651,508 pixels, of which between 30,724 (~5%) and 85,829 (~13%) present a positive and significant correlation, according to the considered delay time, from 0 to 26 day. To meet the reviewer's request, which is legitimate, we focused on three evidenced sea regions characterized by systematic high correlation between INP and CHL (Figure 7) and we divided, within each region, the significant and positively correlating pixels into three categories: High, Medium, and Low correlating, according to the distribution of the correlation coefficient. Then we selected randomly 6 pixels within each category, per each region, of which we plotted the results of the INP vs CHL regression analysis, for a total of 54 scatter plots. Careful investigation of the randomly selected scatter plots show a variety of conditions regarding the robustness of the investigated correlation, with generally robust correlations, in the majority of the cases not distorted (or influenced) by one single (or a few) points, which we consider a prove of the robustness of the obtained correlation maps.
The scatterplots have been added as an Appendix to this AR document and to the Supporting Material.

*RC:* Line 203: Explain why a relationship between INP and chlorophyll concentration is expected.
*AR:* We have added this lines as explanation: "Recent literature (Wilson et al., 2015; Knopf et al., 2011; Wang et al., 2015) has showed that sea-spray organics can nucleate ice being potentially important INPs in the clean

marine atmosphere. Mansour et al. (2020b) evidenced that *n*INP over the North Atlantic Ocean follows the patterns of marine biological activity as traced by surface CHL concentration".

*RC:* Line 203: Excluding the samples with land input is mentioned several times. Elaborate why this is important.
*AR:* We are trying to investigate the relation between INPs and marine biological activity: it is reasonable to exclude from the dataset the samples for which we have a clear evidence of a terrestrial influence. This has been made clearer in the text "to focus only on INPs potentially originated from the sea".

*RC:* Line 212, 213: specify, concentration of INP
*AR:* This is the general description of a general chemometric approach; it is valid for INPs or for any other atmospheric concentration. We explained in the following lines that we applied it to INP concentrations.

*RC:* Line 217: Specify how many trajectories were used and demonstrate that this is a large enough sample to draw conclusions. Looking at the figures it seems that higher CWT is found where more trajectories passed.
*AR:* We thank the reviewer for evidencing the limit of the proposed CWT solution: indeed, we have verified that a significant correlation can be observed between the CWT results presented in the original Figure 7c and the number of BT endpoints in each cell. This is shown in the plot below (left).

[Figure]

[Figure]

Weighted (Eq. 2)                                        Not weighted

This effect, which should not be present in the outcome of the CWT analysis and requires a correction, is not related to the number of back-trajectories or of samples, rather to the weighting approach described in Equation 2. Indeed, the above (right) plot shows that this correlation disappears if the weighting step is excluded from the CWT analysis.
Weighting based on the number of passages over one cell has the aim of avoiding that cells with a low number of passing back trajectories (typically cells that are at the borders of the domain) are considered of the same importance as cells characterized by many passages, for which the CWT value is statistically more robust. If one cell has only a back trajectory endpoint, its CWT will be determined only by one INP sample. On the contrary, a cell crossed by many trajectories will have a CWT which derives from the weighted averaging of many samples. Weighting the cells by the number of passing endpoints is common practice in applying the CWT method (Cheng et al. 2013; Hsu et al., 2003; Jeong et al. 2011), even though the weighting step may also be excluded (Bycenkiene et al., 2014).
In the original manuscript, we derived the Weighting correction scheme (Eq. 2) from Masiol et al. (2019a, b), selecting it among different examples found in literature, because the correction is based on intrinsic properties of the dataset (i.e., the distribution of the number of endpoints within the cells), which makes the choice of the W scheme less subjective. Nevertheless, we recognize that the weighting criteria are evidently too strong and are responsible of the effect evidenced by the reviewer (high CWT is associated to high number of passing trajectories). To check the effect of the W correction on the overall source location approach, we report below also the unweighted CWT results (i.e., the CWT map before applying the W

correction). The same main source regions result from both plots (indicated by black circles to guide the eye), while the main difference is observed in the most marginal south-east zone of the domain, where potentially high CWT values (in the uncorrected plot) are down weighted (in the left plot) because of the low number of endpoints determining it. In conclusion, the application of the W correction does not modify substantially the results evidencing the same source regions in both plots.

[Figure]

Few trajectories at the border of the domain: not robust statistics

Considering the evident limits of the solution presented in the original manuscript, we elaborated a new solution for the revised manuscript. In this new solution, we adopted a softer Weighting correction for cells with a low number of endpoints and we increased the number of BTs considering two trajectories per day to be associated to the corresponding INP concentration of the day (total BTs = 28, total end points = 3388 of which 2184 endpoints passing at low altitudes (< 500m)). This doubles the number of BTs enhancing the statistics of the CWT solution. Obviously, we cannot increase the number of INP samples available for the analysis, but we note that Hsu et al. (2003) successfully applied CWT to datasets consisting of 22 and 30 samples, which is not far from the dimension of our dataset.

The following Table presents a comparison between the new UNWEIGHTED and WEIGHTED solutions. As it is clearly shown by the third column, no one of the solutions present a correlation between CWT and the number of endpoints in a cell (N); furthermore, they show a general agreement regarding the major identified sources, demonstrating that the weighting approach does not shape the CWT maps, but it only clears the solution by removing the less representative cells. Finally, the identified sources are generally the same as evidenced in the previous version of the CWT analysis (original manuscript), showing that the solutions are robust and independent on the number of BTs deployed.

[Figure]

[Figure]

Finally, we note that, in our work, CWT is used only qualitatively to evidence the location of the most probable source regions of INPs over the selected marine domain (to provide a comparison term for the results of the spatio-temporal correlation with CHL). For this purpose, also a limited dataset as the present one might be sufficient, as shown in the general consistency of the "W corrected" and "not corrected" plots above. On the contrary, to use the same approach quantitatively, for instance by comparing the relative strengths of the evidenced source regions, by comparing their CWT values, a more extended dataset would certainly be necessary.
The text has been modified reporting the new CWT solution (Lines 290-295 and 504-511).

*RC:* Line 227: Justify that longer residence time in a grid box is related to higher INP concentration. I would expect high windspeed to generate more particles, but also less endpoints at the location because the trajectory moves faster. Discuss assumptions made for this analysis.
*AR:* Longer residence time does not translate in higher CWT values as the parameter Dij appears both at the numerator and at denominator. For instance, if only one trajectory passes in a grid cell, the resulting CWT for that cell will be equal to the INP concentration measured at the sampling point at time of arrival of that single back trajectory. This will happen independently on the residence time. In fact, in this case the formula will be
CWT = C * Dij / Dij = C (in this case, C = *n*INP).
The residence time is only used to "weight" the relative contribution of each back-trajectory in determining the final CWT value of a cell. In other words, if a cell is crossed by multiple trajectories, the final CWT will be influenced more by the concentration (C) associated to the trajectories that stay over the cell for more time, following a classical "weighted averaging" approach.

*RC:* Line 227: What uncertainties are avoided by weighting? Motivate the application of a weighting factor. This methodology makes no sense to me and needs a clearer explanation.
*AR:* Explained above. Thanks for bringing this up – the authors admit that the sentence in the manuscript was not clear; in the new manuscript it was reformulated as follows:

"In order to reduce the impact of grid cells containing a low number of endpoints, for which the calculation of the CWT is statistically less robust, the CWT values were multiplied by a weighting factor ($W_{ij}$) according to Eq. (4).

$$W_{ij} = 1 \text{ (if } D_{ij} \geq median), W_{ij} = 0.8 \text{ (if } 3 < D_{ij} < median), \text{ and } W_{ij} = 0 \text{ (if } D_{ij} \leq 3) \tag{4}$$

The introducing of the weighing factor reduces the number of considered cells to 203"

*RC:* **Results**

Line 237-238, 244ff: Discuss different ice nucleation modes in the introduction section. Remove here.
*AR:* This is a good suggestion. We moved the ice nucleation mode discussion to the third paragraph of the introduction section.

*RC:* Line 240: specify "sharper"
*AR:* Corrected using "steeper".

*RC:* Line 241: Provide a more detailed explanation how "time resolution" and "sampling activities" can explain these differences. Calculate how much of the difference can be explained by the uncertainty introduced by the ice nucleation analysis and how much from uncertainties in sample volume.

*AR:* The uncertainties involved in WT-CRAFT immersion efficiency analysis and sampling flow rate are ±23.5% and ±3.7%. Those for DFPC are 30% and <10%. These 'systematic errors' would not be able to explain the difference we observed.

More in detail, in the manuscript we enlisted some parameters that may have contributed to the observed discrepancy. For instance, we are comparing samples with different time resolutions (4 days vs 4 hours). This could explain some discrepancy in the resulting INP concentrations by the two techniques, if a strong diurnal gradient was present in the INP concentration. Honestly, we believe that this alone could never explain the observed discrepancy, considering also the absence of a day-night cycle during the Arctic summer. We considered that some uncertainties in the sampling flow rates (see above) could also have contributed a small fraction of the discrepancy, but no flowmeter can be so off as to generate differences of 8 times. Finally, we highlight that the discrepancy is temperature dependent, which could not be justified by sampling volume uncertainties alone.

We have made this clearer in the revised version of the manuscript (please, refer to sub-Sect. 4.1).

*RC:* Line 242: unclear what the references point at

*AR:* These references have been removed.

*RC:* Line 242f: Elaborate based on what it is a valid assumption that the ice nucleation mode generates the observed difference of higher nINP from condensation than immersion mode.

*AR:* Discussed below.

*RC:* Line 247: Vali 1975 is a better reference for ice nucleation modes

*AR:* We used Pruppacher and Klett (2010) and Vali et al. (2015), as reference for the ice nucleation modes.

*RC:* Line 247-255: This section is speculative. Provide an explanation how the different mechanisms can exert an influence on nINP and why in particular on mixed particles. Much more probable would be an uncertainty in the sample volume.

*AR:* We echo that a sampling volume uncertainty can hardly explain the observed difference. Furthermore, if a sampling volume would be a source of the issue, it would impact the INP concentration across the assessed temperatures. However, the observed difference varies depending on temperature, which implies that any systematic errors might not be the cause of such a T dependent trend.

We have modified this part, that is now included in the Discussion Sect., as reported above.

*RC:* Line 257ff (Sec. 3.2): For a field study as this, aiming to learn something about the abundance and nature or source of INP, I would consider the differences in concentration of minor importance. Focus should be on the big picture, on trends while being cautious not to overinterpret the data.

*AR:* We agree with the reviewer that the inter-comparison between DFPC and WT-CRAFT is not the focus of the manuscript, and we limit to acknowledge its existence and to show that it does not affect any general feature observed by the two measurements (seasonal trend, relative time series). We also now carefully choose the word to discuss agreement with previous measurements.

*RC:* Line 264: give concentration ranges at -15°C, -18°C, -22°C to compare to DFPC instead. Specify what can be learned from these concentration ranges.

*AR:* The difference in concentration between DFPC and WT-CRAFT is described in detail in the previous paragraph and now visualized in Fig. 2.

*RC:* Line 265: repetition from introduction line 67-68.

This part was merged with the Introduction.

*RC:* Line 272: It is implied that Borys, 1983, Bigg 1996, Bigg 2001 did not measure in the immersion mode. This should be clarified. I recommend merging the literature review here into the introduction.
*AR:* Indeed, Borys (1983) measured using a dynamic processing chamber, similar to DFPC, while Bigg (1996 and 2001) used a static thermal diffusion chamber. Such information was added in the revised introduction section.

*RC:* Line 265-290: Consider presenting the comparison to literature in form of a table and to shift it into a Discussion section. Point out and discuss any systematic differences between marine and land influenced data from the Arctic region at specific temperatures.
*AR:* Now Table 1 reports a compilation of previous INP observations in the Arctic.

*RC:* Line 279-280: Explain how parameters intervene with INP concentrations. Specify what is meant by "particle activation modality". Quantify the conclusion that the data are generally consistent to literature.
   *AR:* This is a valid suggestion. The sentence was modified as follows: "We note that the comparison to these past studies is only qualitative given the great variability of parameters that could influence $n$INP (e.g., different instruments, locations, season, weather conditions, aerosol particle size distribution, ice nucleation mode, etc.). Regardless, both the DFPC and WT-CRAFT datasets fairly overlap with the $n$INP results reported in Wex et al. (2019), especially for $T$s below -15°C. The authors showed $n$INP previously measured at the same GVB station, during spring and summer 2012. The comparison between the $n$INP data from this study relative to Wex et al. (2019) can be seen in Fig. 2. While this figure provides only a qualitative comparison as two studies examined different aerosol particles collected in different years, we found several interesting agreements and disagreements. First, at $T$ = -22°C, Wex et al. (2019) report a very narrow concentration range (27-33 m$^{-3}$), resulting from only three samples, while DFPC and WT-CRAFT measurements span a much wider range (ca. 3-200 m$^{-3}$). The upper limit of observable $n$INP in Wex et al. (2019) was roughly 40 m$^{-3}$, depending on the volume of air sampled onto the analysed filters. On the contrary, the data ranges are in good agreement for $T$s over -18 to -15°C. Finally, the data from Wex et al. (2019) span over a wider range (ca. 10$^{-1}$ – 10 m$^{-3}$) than WT-CRAFT ones (1-3 m$^{-3}$) for $T$> -15°C. The difference in the lower limit of the observations is due to different detection limits of WT-CRAFT (1 m$^{-3}$) and Wex et al. (2019) immersion freezing (ca. 10$^{-1}$ m$^{-3}$) measurements".

The authors also show a part of our new Fig. 2 for clarity.

[Figure]

Figure 2: Ambient $n$INP as a function of the activation $T$ measured at GVB during 2018 by DFPC and WT-CRAFT. DFPC data are divided in spring (blue) and summer (red) samples, while WT-CRAFT data are color coded according to the sampling date. (a) PM$_{10}$ (DFPC) and TSP (WT-CRAFT) data.

*RC:* Line 282: Quantify "reasonable agreement"
*AR:* Discussed above.

*RC:* Line 285: Quantify "overlaps well"
*AR:* Discussed above.

*RC:* Line 286: Quantify "wider range"
*AR:* Discussed above.

*RC:* Line 289: What other factors can explain the differences? It would be helpful to specify the upper and lower detection limits of the methods used here for a comparison to Wex et al., 2019.
*AR:* Discussed above.

*RC:* Line 291-299: Wide reached and speculative. Sec. 3.7.2. does not provide quantitative evidence on the contribution of continental particles.
*AR:* We have removed this sentence from the text: "The significantly lower INP concentrations observed over the remote North Atlantic Ocean are likely due to the lack of continental particles, which we will show play an important role in the Arctic atmosphere" The remaining part is just a neutral comparison between present and previous DFPC measurements.

*RC:* Line 299: Name the locations of the high-altitude and coastal measurements in Rinaldi et al., 2017, 2019.
*AR:* Done. The revised version of the text is; "If we compare with recent measurements performed at lower latitudes by DFPC, $n$INP over the Arctic was lower than those observed in continental European sites (San Pietro Capofiume, Po Valley, Italy; Belosi et al. (2017) and Rinaldi et al. (2017)), but comparable or even higher with respect to those observed at high altitudes (Monte Cimone, Norther Apennines, Italy; Rinaldi et al. (2017)) or at a Mediterranean coastal location (Capogranitola, Southern Sicily; Rinaldi et al. (2019))".

*RC:* Line 300: Ice formation is usually observed at -15°C in filter based INP measurements and not unique. It is also present in dust rich environments. Provide references for examples showing otherwise.
Line 303: Specify the "special feature"
Line 304: This is the only reference to Fig.2. The figure is not relevant and can be removed.
*AR:* We decided to remove the part of the text referred to by the above three comments (L300-306 of the old version), together with previous Fig. 2.

*RC:* Line 309: Two size ranges do not qualify as "size distribution".
*AR:* For clarity, we rephrased it to; "…allowed to investigate in fine (< 1 μm) and coarse (>1 μm) INPs".

*RC:* Line 309: Instead of Table 1, provide a figure showing a scatterplot of INP concentrations measured on PM1 versus PM10 filter in the same time interval (day). All 3 temperatures can be included. Use different colours for spring and summer data.
*AR:* We have added the requested plot in the supporting material (Figure S1). We would like to keep the Table (now Table 2) in the manuscript. The authors consider these are important data to report in the main text. The Table was updated including WT-CRAFT data sorted by season.

[Figure]

Figure S1. Scatter plot between PM$_1$ and PM$_{10}$ $n$INP at T of -22°C (a), -18°C (b) and -15°C (c).

*RC:* Line 311: Specify why long distance to source is suggested. Quantify "long distance".

*AR:* The authors meant aerosol transport over scales of hundreds to thousands of kilometers, when it is not possible to be more precise on the source distance, by "long distance" or "long range" transport. We have added the explanation in the text; "A small contribution from coarse INPs characterized the spring campaign (~20%), suggesting that the dominant INP sources may be located at long distances (scale of the order of 100s-1000s km), with consequent depletion of the largest particles during transport, due to their higher gravitational deposition velocities. This result is consistent with previous works highlighting the contribution of long range transport from lower latitudes during the Arctic spring (Shaw, 1995; Heidam et al., 1999; Stohl, 2006)".

*RC:* Line 314: To substantiate this interpretation, compare to by how much the concentration of particles in the coarse fraction change from spring to summer, based on the measured size distribution.

*AR:* We have added the suggested analysis. The text was modified as follows; "While these coarse INP fraction estimation, presented in Table 2, involves substantial uncertainties, the same trend is inferred by the particle size distribution measurements, which show a significant (p<0.01) enhancement of coarse particles contribution in summer (median 30%) with respect to the spring time (median 16%) (Fig. S2)". Moreover, we complementing this statement by presenting the seasonal coarse fraction aerosol particles contribution increase in Fig. S2.

*RC:* Line 316: Speculative, the coarse particles could be dust particles. It is not clear to what "above considerations" this is liked to.

*AR:* This part was reformulated to remove speculation; "The increase of coarse INP contribution, from spring to summer time, is progressively more pronounced with increasing activation T. A similar coarse fraction dominated INP population was reported by Mason et al. (2016) for measurements performed between 29 March to 23 July 2014 at the Alert Arctic station, with increasing coarse INPs contribution as a function of the activation T. Our results are unique compared to past studies as our measurements and data support the increase of coarse INP contribution during the meteorological season transition from spring to summer with increasing activation T."

*RC:* Line 308: PM1 data is not depicted in any figure but used in the analysis. Include DFPC PM1 data to Fig.1 and Fig.3.

*AR:* PM1 INP data are now reported in the revised Fig. 2 and also in Fig. 4.

*RC:* Line 308-318: the difference between PM1 and PM10 samples is not obvious from this section. Please provide a figure showing both time series together at -15°C, -18°C, -22°C as well as a scatterplot comparing PM1 to PM10 INP concentrations.

*AR:* Now provided in Fig. 2, 4 (time series) as well as Fig. S2 (scatter plots).

*RC:* Line 319-325: INP concentrations from PM1 and PM10 should be compared to particle concentrations <1um and <10um including all smaller sizes instead of only supermicrometre particles. Otherwise the comparison is not objective and only implies that INP concentrations were similar for both cut-offs and the difference is introduced by the choice through what size range was divided.

*AR:* This is precisely what we did. We compared PM1 INPs with sub-micrometer particle number and COARSE INPs (obtained by difference: PM10 – PM1) with super-micrometer particle number, providing the relative activated fractions. Finally, when discussing the AF of PM10 samples, we have considered the whole range 0.5-10 μm for the aerosol particle number. This part was moved to the new "Activated Fraction" Section (3.3).

*RC:* Line 327-332: shift to Discussion or introduction. Show how the current data compares to trends found in other studies, e.g., Wex et al., 2019 by plotting the data (at -15°C, -18°C, -22°C) as a function of DOY into the same time-series.

*AR:* We have added a further plot to the Supporting Material (see below) addressing the referee's point. Unfortunately, Wex et al. measured only three points (all in spring) at T=-22°C, so the only temperatures at which the comparison was possible for the three methods are T =-15 and -18°C.
In calculating the Summer/Spring ratios, we based on the threshold that appears evident from Wex et al. (2019): DOY 150.

[Figure]

Figure S3. Seasonal evolution of nINP in this study (GVB, 2018) compared to the results by Wex et al. (2019), here indicated as W19, obtained at GVB in spring-summer 2012.

*RC:* Line 333-343: The main findings need to be worked out clearer in this section. Listing a lot of factors at random temperatures in the text is not helpful to understand the situation. Show that the small dataset can be used to determine robust trends. Factors on the order of 2 are small and should not be overinterpreted. Scattering within a season is much higher.
*AR:* We would like to retain these discussions as they exhibit statistical significance according to the standard t test. To increase the clarity, we have modified the text as follows; "Interestingly, our 2018 time series data in Fig. 4 do not indicate a clear seasonal increase in ambient *n*INP from spring to summer. A comparison between the seasonal trends in this study and from Wex et al. (2019) can be found in Fig. S3. For the DFPC data, a statistically significant ($p < 0.01$) *n*INP reduction (by a factor 1.5) was found at *T* of -22°C, passing from the spring campaign (April) to the summer period (July), while no significant ($p > 0.05$) difference was observed for *T*s of -15 and -18°C.

The time series of *n*INP measured by WT-CRAFT agrees with the DFPC one if we consider only the periods in which the two sampling activities were run in parallel: a statistically significant ($p < 0.05$) reduction by a factor 1.6 is observed at -22°C and no significant differences can be appreciated at -15 and -18°C. On the other hand, considering the whole WT-CRAFT data extent, a statistically significant ($p < 0.05$) increasing *n*INP seasonal trend was observed but only for *T*s between -17.5 and -21.5°C. Even in these cases, the spring to summer enhancement ratios did not exceeded a factor of three. We notice that such variations are smaller than the variability of *n*INP observed within one season. A primary peak in *n*INP was observed by WT-CRAFT during June, at *T*s lower than *T* = -17°C (Figs. 4a and 4b). Further, the increase was visually notable in this case: the average *n*INP during June was up to ~3 (*T* = -20°C) times higher than the average of the rest of the measurement period. As can be seen in Figs. 4a and 4b, a second peak of *n*INP can be observed at the end of the WT-CRAFT measurement period, with the last sample presenting the highest concentrations of all the campaign for many activation *T*s. Further discussion of the *n*INP-AF relationship during this specific period is provided below."

*RC:* Line 337: What is the reason for limiting the WT-CRAFT dataset to the same period?

*AR:* The reason is simply to show consistency within the two datasets. By all means, we are not limiting the WT-CRAFT dataset. We first show agreement with the DFPC dataset considering only the overlapping periods, then we present the trends in the whole WT-CRAFT dataset.

*RC:* Line 340: Why only at -17.5°C and -21.5°C? A plot showing the individual measured Tspectra would be helpful to show how relevant this increase is.
*AR:* Not only at -17.5 and -21.5°C but between these two temperature boundaries. For clarity, we substituted "within" with "between". Individual WT-CRAFT datapoints are now presented in Fig. 2 (color coded by sampling date), and we state that the Summer/Spring ratio is within a factor of three in the revised text.

*RC:* Line 340: quantify "clear nINP peak"
*AR:* We re-worded it as follows; "A primary peak in $n$INP was observed by WT-CRAFT during June, at $T$s lower than $T$ = -17°C (Figs. 4a and 4b). Further, the increase was visually notable in this case: the average $n$INP during June was up to ~3 ($T$ = -20°C) times higher than the average of the rest of the measurement period".

*RC:* Line 344: Why was the last sample excluded?
*AR:* In the revised version we report the difference between the June peak and the rest of the WT-CRAFT observations without excluding any sample. The results do not change. Furthermore we added the following caveat at the end of Sect. 3.4:
We note that the AF data of WT-CRAFT in August is not available due to the lack of SMPS-APS data (maintenance reason). Thus, whether the increase of $n$INP detected by WT-CRAFT in August (i.e., the last two data point sin Figs. 4a and 4b) corresponds to the enhancement of ice nucleation efficiency or absolute aerosol particle concentration remains uncertain.

*RC:* Line 345: It is an often-misinterpretation of DeMott et al., 2010. The concentration of particles >0.5um are simply used to parameterize INPs of all size, not an actual size fraction of them.
*AR:* We agree with the reviewer, and this sentence is excluded from the manuscript.

*RC:* Line 346: This is incorrect. Aerosol were not more ice active, there were only more INP.
*AR:* We agree with the reviewer - an abundance of INPs and an IN ability/efficiency of aerosol particles are two different things. This sentence is excluded from the manuscript.

*RC:* Line 347-350: Speculative. Maybe the nINP is higher because more activity at the station towards the end of sampling.
*AR:* Any activities at the station were carefully recorded, and we do not have any records of suspicious sources of INPs towards the end of our sampling.

*RC:* Line 352: How was the statistical significance of a seasonal trend determined?
*AR:* The trend significance was evaluated by checking the statistical significance of the Pearson regression between INP concentration and time.

*RC:* Line 353: Quantify "peaked mainly"
*AR:* The magnitude of the June peak is already quantified before; we added the peak/baseline ratio in parenthesis: "(up to 3 times higher concentration that the rest of the measurements)".

*RC:* Line 355: Speculative. One order of magnitude scattering occurs also on short timescales.
*AR:* The authors agree. We decided to clarify our points by referring to what the referee pointed out as well as Schrod et al. (2020); "Such results are more in line with the flat trends reported by Schrod et al. (2020). The observed discrepancy between current and aforementioned past studies may be indicative of the inter-annual variability of meteorological conditions and aerosol particle sources determining the ambient $n$INP. Nonetheless, the number of $n$INP. observations in the Arctic and their temporal coverage remains limited to derive general conclusions on the $n$INP trends."

*RC:* Line 360-371: Explain the relevance of scavenging values for the interpretation of measurements here or delete these lines.
*AR:* We agree with the reviewer on the limited relevance of this part and decided to delete it.

*RC:* Line 372: Covariance with particle concentration was not shown. This could be an interesting addition to discuss the ice active particle fraction.
*AR:* We did not report the result of the correlation analysis between $n$INP and particle number concentration as there were mainly non-significant results. Only the few significant results have been included and described in the text.

*RC:* Line 374: Quantify "even more accentuated".
*AR:* We modified the text as follows; "During summer, no correlation at all was observed between $n$INP and particle number (R between -0.13 and -0.25)."

*RC:* Line 375-376: Sentence fragment.
*AR:* Changed to; "It is, however, important to note that previous studies from different regions report various results about the correlation between INP and particle number concentration: a correlation is often reported with the number concentration of aerosol particles larger than 0.5 μm (DeMott et al., 2010; DeMott et al., 2015; Mason et al., 2015; Schwikowski et al., 1995); in other cases, a complete lack of correlation has been documented (Richardson et al., 2007; Rogers et al., 1998), which is not surprising considering that INPs are only a small fraction of total particles".

*RC:* Line 376-383: Unclear what this discussion is aiming for. Clarify main point.
*AR:* This part just aims at setting our results in the perspective of previous measurements. By all means, our results may not be conclusive – that is what we meant to infer.

*RC:* Line 385ff (Sec. 3.6) Converting nINP to ns doesn't yield new insights. As stated in line 379, INP are only a small fraction of total particles and the total surface area from all different particle types is not related to the number of INP. I recommend deleting the section and Fig. 4 and Fig.5. Instead include the ice active particle fraction at different temperatures and the spring-summer contrast.
*AR:* The authors agree. We have removed the active site density section and discussed only AF data. But, our overall conclusions did not change. The following comments, therefore, refer to the text that was excluded in part.

*RC:* Line 389: Repetition of line 72
*AR:* Yes, thanks for noticing. We believe that this is an important point and worth being echoed.

*RC:* Line 397: Explain why significance is not found for -16°C. Contradicting results in a narrow T-range could indicate that this analysis is not robust.
*AR:* This is not a contrasting result; for T=-16°C we have still a positive correlation, but with an R value just below the significance threshold.

*RC:* Line 399f: The difference indicates that there is no general trend.
*AR:* The general trend is a slight increase of $n_s$ passing from spring to summer. The fact that we see the maximum increase at different temperatures for the two datasets suggests that aerosol particles may respond differently to the activation modes, as hypothesized previously. Nevertheless, we admit that highlighting this would be speculative.

*RC:* Line 408: Quantify "substantial good agreement"
*AR:* The agreement was within a factor of 2.5. Nevertheless, this section was removed as we used AF to describe the ice nucleation efficiency of aerosol particles instead of the ice nucleation active site density.

*RC:* Line 409f: The aerosol population at GVB is a mixture of many particle types and only a tiny fraction acts as INP. Interpreting ns compared to ns from well constrained particle types is speculative.
*AR:* This sentence is removed. We now point out "that INPs are only a small fraction of total particles" in Line 427.

*RC:* Line 417: Show scatterplots of significant correlations in the supplement.
*AR:* The authors believe that the tabular form of data presentation with our concise explanation is adequate and sufficient. Showing > 30 plots (considering 3 temperatures and two size ranges) for one season would be cumbersome. We provided some representative and meaningful snapshot scatter plots to respond to a later comment (please see below).

*RC:* Line 421: Explain why and how the results are in line with what considerations.
*AR:* We thank the reviewer for pointing out this unclear paragraph. The text was modified accordingly:
"In order to investigate the potential sources of the INPs at GVB, a correlation analysis was performed between both $n$INP datasets and the atmospheric concentration of chemical tracers routinely measured at the station. During the spring campaign, $n$INP correlated with tracers of long range transported anthropogenic aerosol particles such as nitrates, non-sea-salt-sulfate and non-sea-salt-potassium (Table 3). Indeed, Udisti et al. (2016) associated spring time non-sea-salt-sulfate at GVB to long range transported anthropogenic sources. The authors also showed that the production of biogenic non-sea-salt-sulfate from the sea is relevant only in summer time. The spring time peak of anthropogenic aerosol transport from lower latitudes is often referred to as the Artic haze (Shaw, 1995). A general tendency to anticorrelation with sodium and chlorine was also observed in both the size classes, though only $PM_1$ is statistically significant ($p<0.05$)". Further discussion of these findings is provided in the new Discussion Section (4.3).

*RC:* Line 422: If these are general tendencies they should agree with the PM10 data as well. Explain why the analysis is limited to PM1".
*AR:* All the INP concentration data anticorrelate with Na and Cl in springtime, but the $PM_{10}$ ones have a Pearson correlation coefficient (R) below the significance threshold corresponding to the 95% confidence interval. In the case of Na (as an examples), R is -0.61 for $PM_1$ and -0.49 for $PM_{10}$ at T=-22°C; similarly, it is -0.59 for $PM_1$ and -0.36 for $PM_{10}$, at T=-18°C; finally it is -0.60 for $PM_1$ and -0.25 for $PM_{10}$, at T=-15°C. This shows a general tendency to anticorrelation between the tested variables, even though the result is clearer when considering only the fine fraction. Considering the whole size spectrum ($PM_{10}$) may include different sources, with different relations with sea spray, resulting in a less clear correlation.

*RC:* Line 423: Quantify "Less clear"
*AR:* The text was modified as follows; "The only significant relations observed from the analysis of the summer DFPC data was for T = -15°C…"

*RC:* Line 427: Explain why these elements are not good tracers for the soil type. What would be good tracers for the local mineralogy of the soil?
*AR:* This sentence was admittedly speculative and thereby removed. In the future, the measurement by polarization lidar etc. (Mamouri and Ansmann, 2015; 2016) may provide the better insight of dust and $n$INP.

*RC:* Line 415-429 (Sec. 3.7.1): Add a conclusion, lesson learned from this exercise.
*AR:* This requested "summary" is now included in the new Discussion Section 4.3.

*RC:* Line 430-441 (Sec. 3.7.2) Suggest some arguments why a larger land fraction (residence time) of a trajectory should linearly correlate to the INP concentration at -15°C. Why not at lower temperatures as well? The distance of land contact to the receptor, time past, precipitation formation along the trajectory and source strength in different land locations should make a large difference.
*AR:* We did not address that the effect of land contact is evident only at -15°C. Table 3 and Figure S3 shows that this effect is evident for all the three probed temperatures. Indeed, for the WT-CRAFT dataset, for T=-15°C the correlation is positive but not significant, conversely to the other two reported temperatures.

We are aware that this is a simplified model; nevertheless, we believe that it provides an idea of the broad effect of different land cover types on the INP concentration.

*RC:* Line 433: Figure S1 is more informative than Fig.6 to show the overpassed ground types. I suggest changing Fig.6 for Fig.S1. In addition, Fig. S1 is referred to more often than Fig. 6 later in the manuscript.
*AR:* This is a good suggestion. We have merged the two Figures into the new Figure 6.

*RC:* Line 436: A scatterplot showing fLand versus nINP instead of timeseries would be more helpful than Fig.S3 and Table 3, to show the influence of land sources.
*AR:* We will address this comment with greater detail and examples below.

*RC:* Line 435-438: If a fLand effect is found at -15°C it could be evidence against biological INPs dominating nINP at this temperature.
*AR:* We disagree with this comment. As biological particles or fragments can also derive from soil dust, a snow-free land can act as a source of biological INPs.

*RC:* Line 440: specify what the "outcome" is and provide an overall conclusion from Sec. 3.7.2.
*AR:* The following remark was added in the new Discussion Section (Lines 590-594): "This analysis points out that both marine and terrestrial sources may contribute to the INP population in the study area, with land sources showing a potential for dominating the INP pool, due to the higher ice activity of mineral dust and soil particles. On the contrary, marine sources may be significant, even though marine INPs are intrinsically less ice active, because of the extension of ice-free sea waters during the Arctic summer. This has implications also for the future balance between terrestrial and marine INP sources in a warming Arctic (Murray et al., 2021)."

*RC:* Line 443-482 (Sec. 3.7.3) Specify that this analysis was performed using 14 datapoints from PM1 DFPC. It should be demonstrated that the limited dataset yields robust correlations with CHL. Show some scatterplots. I suggest to include high CHL regions as a ground condition, subdividing the sea category, and include it in the analysis of sec. 3.7.2.
*AR:* Discussed above. In the new version, we have added the following caveat (Lines 594-602): "The major limitation of our spatio-temporal correlation analysis and of the INP spatial source attribution approach (CWT) is the low number of samples available. This limits the time representativeness of the dataset and increases the uncertainty of the outputs. Nevertheless, the consistency of the two independent approaches (spatio-temporal correlation analysis and CWT source location) provides a certain measure of credibility to the presented results. For this reason, we consider the above as an implication that the marine biota may be a source of INPs in the Arctic. Nevertheless, further studies, based on more robust datasets, are necessary to confirm this result and to achieve a more quantitative understanding of the relative importance of marine vs. terrestrial INP sources over the Arctic. In particular, online INP quantification methods have the potential to provide better suitable data for this kind of statistical approaches and will certainly contribute to clarify INP sources over the Arctic in the future"

*RC:* Line 443-459: The hypothesis and description of how INP and CHL maps are correlated fits better to the Method section.
*AR:* Yes, it would. But, as the method itself is stabled and citable (Mansour et al., 2020a and b), we intentionally keep the content close to the Results section for the reader.

*RC:* Line 451: Why are trajectories with land contact excluded and why only some? A short land contact can have a large impact on nINP.
*AR:* Here we are trying to show that outside the main INP inputs from land, the background INP concentration has a relation with marine biological activity. Therefore, we have to exclude land influenced samples that would generate a spurious signal in the correlation analysis, being not related to marine sources. We have

excluded samples with fLand>=10% as we did not observe an effect on the INP concentration for samples with lower fLand. Indeed, apart one sample with fLand = 8%, which does not show any anomalous INP concentration, the samples with fLand>0 have values between 1 and 2%, which can be considered negligible. Removing the sample with fLand=8% does not substantially alter the observed correlations with CHL.

*RC:* Line 460: The time lag doesn't make sense to me. Why would the aerosol generating, biochemical process not change location in 6 days or 16 days? The movement of the surface water should be considered.

*AR:* The time-lag approach is often used in oceanographic studies (for instance, Volpe et al. (2012), which concluded that phytoplankton biomass and surface heat content anomalies are related with a roughly 5-month time-lag in the Mediterranean Sea). This approach has been derived from oceanography and was applied for the first time on atmospheric studies by Rinaldi et al. (2013) and O'Dowd et al. (2015). They observed a time-lag between CHL concentrations from satellite measurements and organic matter enrichment in the ambient marine aerosol. Later on, a delay time (4 to 10 days) between changes in sea-spray chemical composition and CHL peaks was observed in controlled laboratory experiments conducted in a wave tunnel (Lee et al., 2015; Wang et al., 2015). Furthermore, McCluskey et al. (2017) demonstrated 4-day time-lag between ice nucleating particles (INPs) activation in sea spray aerosol and CHL concentration as a part of the National Science Foundation Center for Aerosol Impacts on Climate and the Environment (CAICE) experiment. These systematic laboratory studies showed short time lag (which may not completely reflect the reality), but demonstrated that a delay exists between the patterns of CHL evolution during a bloom and the observed effects of biological activity on sea-spray aerosol. Finally, Mansour et al. (2020a, b) introduced the use of source identification algorithms (e.g., CWT, as used here, or PSCF) to support the results of the spatio-temporal correlation analysis between in-situ aerosol parameters and surface CHL concentration fields.

In the spatio-temporal correlation analysis between in situ aerosol parameters and surface CHL fields, considering the movement of surface waters is not important. Our analysis evidences a potential relation between the biological activity occurring X days before the aerosol sampling and aerosol properties (in this case INP concentration) at time of aerosol sampling (t_zero), independently on where the bioproducts originating from said biological activity might be at t_zero.

The assumption that the bioproducts of algal activity, responsible for ejection of marine INPs to the atmosphere, do not move substantially from the production region ,within the considered time span, is relevant only when comparing the results of the spatio-temporal correlation analysis with source location results by CWT. This assumption is reasonable: typical surface ocean motion for the Arctic summer is < 5 cm/s (e.g., Lumpkin and Johnson, 2013), corresponding to less than 4 km/day, which is almost negligible considering the resolution of the pixels (~100 Km) in our CWT analysis. Furthermore, Lehahn et al. (2014) showed that an algal bloom can be confined and stable for as long as ca. 30 days in the North Atlantic Ocean (which is characterized by faster currents than the Arctic Ocean).

Hence, the authors believe that our approach is reasonable as supported by previous studies.

*RC:* Line 464: justify why 6- and 16-day time-lag was selected

*AR:* The reason is explained in the following lines (484-486): "The maps in Fig. 7 were selected because they clearly show high correlation regions in the seawaters surrounding the Svalbard archipelago (time-lag 6 days), close to the Greenland coast (time-lag 14 and 16 days) and to the northeast of Iceland (time-lag 16 days)."

*RC:* Line 477-478: Explain how this can be seen in Fig. 7c?

*AR:* We have added a new plot for major clarity (Figure 8b), showing the intersection between the correlating regions, from the spatio-temporal correlation analysis, and the source regions of marine INPs identified by CWT. This new map confirms substantial agreement in identifying the most likely INP sources in the study domain between the two approaches. We highlight that the two approaches are totally independent and based on different principles; the agreement between the two supports the reliability of the derived conclusions and indirectly also supports the reliability of the correlation analysis, even if conducted with a limited number of samples.

*RC:* Line 479-482: Consistency is not obvious. There seems to be even more negative correlations. The pattern looks random. I would expect some high productive areas based on ocean currents and biological factors that do not change rapidly.

*AR:* We have added a new plot showing clearly spatial consistency between positively correlating regions and CWT identified regions. Regarding the existence of negatively correlating regions, we specify that our approach (Mansour et al., 2020a, b) is based on the assumption that if a marine aerosol component is biogenic is should follow the patterns of biological activity (tracked by CHL). For this reason, we focus on positively correlating sea regions in the spatio-temporal correlation analysis. In the lag-correlation approach, an inverse correlation cannot be explained by a physical mechanism: if we assume the aerosol is biogenic, its concentration can only increase with increasing algal activity (positive correlation); therefore, we attribute the observed negative correlations to the relative patterns of CHL in different sea-regions. In other words, if we identify a positively correlating area in Region X (supported by the source spatial location approach, like CWT in this study) and a negatively correlating area in Region Y, we assume that the correlation (negative) in Region Y is due only to the fact that the CHL pattern in Region Y anticorrelate with the CHL pattern of Region X.

*RC:* **Conclusion**

Avoid euphemistic language.

*AR:* In the revised manuscript, the authors made sure to put conclusive statements and some future work suggestions in this section.

*RC:* Line 484-489: It seems that the paper gains little by including the WT-CRAFT dataset. It is only marginally relevant to discuss seasonality in sec. 3.4. In all other sections it is only mentioned that the data agrees with what was seen from analysing DFPC data. It there was a dependency on ice nucleation mechanism (condensation, immersion) it seems not to make a difference on size and source of INP.

*AR:* The authors now clarify the specifications and capability of WT-CRAFT in the sections above according to the reviewer's inputs. We sincerely hope that our revisions remove misgivings of the reviewer regarding the 150 lpm "stack" flow rate etc. We believe both DFPC and WT-CRAFT results are important to derive our main conclusions, and we keep all INP measurements data as originally presented.

*RC:* Line 490-493: More, larger INP in summer seems contradictory to the absence of a seasonal trend.

*AR:* We have revised the Conclusion Section completely. Now we discuss the INP concentration seasonal trend, the AF one and the seasonal evolution of the contribution of coarse INPs. Briefly, we see only a modest seasonal increase in the INP concentration and only limited to a certain T range. The AF, instead, shows a clearer increase at each probed T. Finally, we address the clear and significant increase in the coarse INP fraction contribution. We do not see any contradiction in the above conclusions: INP number concentration is rather constant passing from spring to summer, while the relative contribution of fine and coarse INPs changes with the season.

*RC:* Line 497: inter-annual variability is a trivial statement. The question is how large the variation is and why it happens.

*AR:* Given the limited knowledge on INP sources in the Arctic environment, it would unfortunately not be feasible to address the reasons behind the discrepancy between the seasonal trends observed in 2012 and 2018. Regarding the magnitude of the spring to summer concentration increase with respect, for instance to day by day variability, we believe that the new version of the manuscript is more quantitative.

Instead, in Sects. 4.2 and 4.3, we added the following statements as for the future study topic:

"Future application of long-term online INP measurements (e.g., Möhler et al. (2021)) may allow shedding light on the seasonal evolution of $n$INP at GVB and over the Arctic in general."

"In particular, online INP quantification methods have the potential to provide better suitable data for this kind of statistical approaches and will certainly contribute to clarify INP sources over the Arctic in the future."

*RC:* Line 499: Explain the importance of this study in detail.

*AR:* The revised Conclusions Sect. explains this in a better way. One of our major findings is related to the different behaviour of aerosol particles sampled at GVB under different ice nucleation modes:

"We considered many factors that could potentially explain the observed difference (Sect. 4) and conclude that the different ice nucleation mechanisms probed by the two techniques (condensation freezing, for DFPC, and immersion freezing, for WT-CRAFT) is an undeniable reason. While differences in the sampling resolution and overall measurement uncertainties have partly contributed to the observed offset, it seems conclusive to address there is ice nucleation mode dependent INP propensity at GVB in 2018 at least. Any future investigations regarding INP compositions and more controlled-study focusing on condensation vs. immersion freezing on identified compositions will lead to further findings to settle this issue".

Another significance of our work is the observation of no substantial seasonal variation in *n*INP accompanied by generally higher ice nucleation efficiency (AF) and a clear enhancement in the contribution of coarse INPs in summer with respect to spring:

"This study also offered unique data examining the seasonality of INPs in the Arctic with respect to *n*INP and AF. Both condensation and immersion INP datasets did not indicate a marked *n*INP seasonal trend. We report a statistically significant spring to summer enhancement in *n*INP only for a narrow range of $T_s$ (-17.5 to -21.5°C) and the associated *n*INP enhancement never exceeded a factor of three. On the other hand, the AF of atmospheric aerosol particles from GVB presents a statistically significant spring to summer increase almost independent on the probed *T*, reaching up to ca. 6 times at *T* of -19°C. A clear seasonal evolution of the super-micrometer INP contribution was observed by DFPC. Such contribution was around 20% in spring (with the highest 32% at -15 °C) and increasing markedly in summer and at high $T_s$ (45% at *T* of -22°C and 65% at *T* of -15°C)".

Finally, we also provide evidence for the different contribution of local vs. long range, natural vs. anthropogenic and terrestrial vs. marine aerosol particle sources to the INP burden at the study location, contributing to improve our current understanding of INP dynamics over the Arctic.

*RC:* Line 502-505: This is a weak conclusion. There is no reason mentioned to assume that only one source contributes INP at all temperatures in the spectra.

*AR:* As evidenced in the revised Introduction, there is not a general consensus in literature on the prevalence of terrestrial or marine sources of INPs over the Arctic. This is partly attributable to the scarce coverage of observational data and may also result from the complexity of the Arctic environment. In this work, we present convincing evidence of the contribution of both source types, even though we are still far from a quantitative understanding of their contributions to the Arctic INP burden. This is probably not the major finding of our work, nevertheless it is a reasonable conclusion worth to be evidenced as it contributes to an open literature debate.

*RC:* Line 506-508: The relation has not been proven without doubt. It is a speculative interpretation.

*AR:* In the revised text, the robustness of our approach has been discussed with more detail. Although longer datasets would be desirable for future investigations of the relationship existing between INPs and the marine biological activity, we believe that our results are robust enough to support some connection between the marine biota and atmospheric INPs during the Arctic summer.

*RC:* **Figures**
The provided figures do not support the content of the manuscript. Fig.2 for example can be deleted is only referred to in a side note and supplementary figures are referred to more often than the actual figures included to the manuscript.

*AR:* We agree. Our former Fig. 2 was removed.

Helpful figures could include:
1. Temperature spectra with all nINP measured with WT-CRAFT and all datapoints (PM1 and PM10) measured with DFPC, with colour code for DOY of measurement and 3 symbols to differentiate the techniques.

*AR:* Provided in Fig. 2.

2. Timeseries of nINP data at -15C, -18C, -22C as function of day of year including the 12 data points from Ny-Alesund in Wex et al, 2019,
*AR:* Thanks for this useful suggestion. We present the DFPC and WT-CRAFT time series in Fig. 4. The suggested comparison with Wex et al. (2019) is now added in the Supplementary Material (Fig. S3).

3. Scatterplot of PM1 vs. PM10 nINP measured with DFPC
*AR:* Thanks for this useful suggestion. The suggested figure is now added in the Supplementary Material (Fig. S1).

4. Time series showing activated fraction of particles (nINP divided by number of particles in PM1 for DFPC, in PM10 for DFPC and CRAFT) at -15C, -18C, -22C.
*AR:* Added instead of the $n_s$ time series, including PM1 (Fig. 5).

5. Fig. S1
*AR:* We merged the old Figure 6 with Fig. S1 obtaining the new Fig. 6.

*RC:* Fig.1: Showing the individual measurements would be more informative than only median, min, max. Please change the figure accordingly and indicate summer, spring, PM1, PM10 samples in different colours and symbols. Indicate the detection limits of the DFPC and WT-CRAFT.
*AR:* All incorporated accordingly. To increase visibility for the reader, we have separated Fig. 1 into two panels:

[Figure]

Figure 1. INP atmospheric concentration as a function of the activation temperature measured at GVB during 2018 by DFPC and WT-CRAFT. DFPC data are divided in spring (blue) and summer (red) samples, while WT-CRAFT data are color coded according to the sampling date. (a) $PM_{10}$ (DFPC) and TSP (WT-CRAFT) data. (b) $PM_1$ data(available only for DFPC). For comparison purposes, the data from Wex et al. (2019), which refer to PM10 samples, are also reported in plot (a). Data for Wex et al. (2019) were downloaded from the repository associated to the publication at….

*RC:* Fig.2: Comparing it to Fig. 4a in Irish et al., 2017 did not make it obvious how it was adapted. What assumptions are made to overlap the two y-axes (INP in water and INP in air)? This figure can be deleted.
*AR:* we appreciate the reviewer's suggestion. We agree and removed this figure as suggested.

*RC:* Fig.3: It needs to be specified how the measurement uncertainty is determined from the sample volume and the analysis. It appears several times throughout the paper and is important.
*AR:* We thank the reviewer for recapping this point. We addressed this point according to the reviewer's suggestions in the revised version. We added INP_PM1 time series and removed the lines between data points in the WT-CRAFT time series (see Fig. 4 of the revised text).

*RC:* Fig. 4: It would be more informative to scale the nINP with the total aerosol number.

This would show that INP are not from the bulk aerosol population but rare exceptions.
Summer, spring and PM1, PM10 can be contrasted.
*AR:* We plotted the AF as requested in a similar way as the revised Figure 2. For major clarity, we divided the Figure in three panels (new Fig. 3).

*RC:* Fig.5: redundant to Fig.4 no new information in this figure. Remove.
*AR:* This Figure was substituted with the time series of AF.

*RC:* Fig.6: The main message from this figure seems to be that sea ice is melting in summer.
This is trivial.
*AR:* We merged the old Figure 6 with Fig. S1 obtaining the new Figure 6.

*RC:* Fig. 7: c) It seems the colour bar shows nINP because it is written on top of it. Replace and label colourbar with units. Add a minimal explanation what can be interpreted from the patterns.
*AR:* Corrected.

*RC:* Fig. S1: Define ground types in the figure caption.
*AR:* A reference to the appropriate text Section was added in the caption.

*RC:* Fig. S2: why are some points connected by lines and others not? Homogenize all precipitation scales and nINP at the same temperatures. -18°C and -22°C plot in first column are switched. Second column last plot DFPC instead of FPC.
*AR:* Lines between points have the only aim of guiding the eye and evidencing better the similarities between the plotted time series (fLand and *n*INP). Data below detection limit are not represented as markers in the plot and result in the broken line noted by the reviewer.

*RC:* Fig. S3: use same fLand scale for all DFPC and WT-CRAFT subfigures and same nINP for same temperatures.
*AR:* We have chosen the scales that makes the plots clearer. Land contribution is much different between DFPC and WT-CRAFT samples because of the different periods covered by the two sampling activities.

*RC:* Fig S4, S5: Include chlorophyll as fifth land type in S1. Remove figures.
*AR:* We would like to keep this figure to clarify our points. We have already answered about the reviewer's proposal of changing our approach above.

*RC:* Fig. S6: use same colour-scale range for all subfigures. Use Fig. S1 map design to facilitate comparison. It seems regions where more trajectory points (Fig.S1) pass, also show higher CWT. This points to a problem with the small dataset size for this analysis.
*AR:* We have answered to these comments above. The new version of the CWT results does not present raised issues.

*RC:* **Tables**
The robustness of correlations in Tables 2 a, b and 3 would be clearer when shown as scatterplots. Due to the small size and structure of the data used, the derived linear correlation coefficients might be strongly biased by few outlier data points and be therefore misleading. Scatterplots help to visually judge correlations. Person's R is sensitive to the data distribution and the R value can be generally misleading. Scatterplots of fLand and nINP would be helpful to investigate these issues.
*AR:* The authors believe that the tabular form of data presentation with our concise explanation is adequate and sufficient. Showing > 30 plots (considering 3 temperatures and two size ranges) for one season would be cumbersome. Below we report some meaningful examples, taken from the highest correlations we observed in spring time between INP concentrations and chemical tracers of anthropogenic aerosols, on which we based some of our main conclusions on INP sources (INP concentrations are in $m^{-3}$ while chemical species are expressed in µg $m^{-3}$).

[Figure]

The situation is different for the relationship between INP concentration and ground types along the sampled air mass. In this case, all the more evident correlations are necessarily driven by outliers, which are the few samples presenting non-negligible concentrations of the fLand value (generally resulting in minimum fSea values)! For this reason we decided to present the results in terms of paired time series (Figure S3 of the old version) together with the correlation coefficients of Table 3. Figure 3 shows that each time fLand has a positive peak, this is associated with a nINP increase. Below we report scatterplots of the most interesting cases.

DFPC_summer:

[Figure]

WT-CRAFT_all data:

**RC: Technical corrections**
Delete "apparently", "likely", "noteworthy", "worth highlighting" throughout the manuscript.
***AR:*** *removed.*
***RC:*** Line 73: icebreaker ***AR:*** corrected
***RC:*** Line 100, 168: km instead of Km ***AR:*** corrected
***RC:*** Line 104: Section instead of Par ***AR:*** corrected/deleted
***RC:*** Line 115: define TSP, define OD ***AR:*** defined
***RC:*** Line 132, 142: per m3 not per m-3 ***AR:*** corrected
***RC:*** Line 140: replace super-microliter with 3uL ***AR:*** replaced
***RC:*** Line 230: define $D$ ***AR:*** the symbol does not appear in the updated version as the formulation of the weighting criteria has been changed
***RC:*** Line 239: nINP instead nIPN ***AR:*** corrected
***RC:*** Line 339: p<0.05 instead p<0.5 ***AR:*** corrected

**APPENDIX**: Scatter plots of randomly selected pixels relative to the spatio-temporal correlation analysis between *n*INP and satellite retrieved surface CHL concentration.

[Figure]

Figure A1: Scatter plots between $n\text{INP}_{\text{PM1}}$ sampled at GVB and CHL at pixels selected randomly within seawaters surrounding the Svalbard archipelago.

**Category**

[Figure]

Figure A2: Same as Figure A1, but for seawaters close to the Greenland coast.

[Figure]

Figure A3: Same as Figure A1, but for seawaters to the northeast of Iceland.

**References used in this AR**

Atkinson, J. D., Murray, B. J., Woodhouse, M. T., Whale, T. F., Baustian, K. J., Carslaw, K. S., Dobbie, S., O'Sullivan, D., and Malkin, T. L.: The importance of feldspar for ice nucleation by mineral dust in mixed-phase clouds, Nature, 498, 355-358, 10.1038/nature12278, 2013.

Belosi, F., Piazza, M., Nicosia, A., and Santachiara, G.: Influence of supersaturation on the concentration of ice nucleating particles, Tellus Series B-Chemical and Physical Meteorology, 70, 10.1080/16000889.2018.1454809, 2018.

Belosi, F., Rinaldi, M., Decesari, S., Tarozzi, L., Nicosia, A., and Santachiara, G.: Ground level ice nuclei particle measurements including Saharan dust events at a Po Valley rural site (San Pietro Capofiume, Italy), Atmospheric Research, 186, 116-126, 10.1016/j.atmosres.2016.11.012, 2017.

Bigg, E. K.: Ice forming nuclei in the high Arctic, Tellus Series B-Chemical and Physical Meteorology, 48, 223-233, 10.1034/j.1600-0889.1996.t01-1-00007.x, 1996.

Bigg, E. K. and Leck, C.: Cloud-active particles over the central Arctic Ocean, Journal of Geophysical Research-Atmospheres, 106, 32155-32166, 10.1029/1999jd901152, 2001.

Borys, R. D.: The effects of long-range transport of air pollutants on Arctic cloud-active aerosol, Atmospheric Science, Colorado State University, Fort Collins, Colorado, USA, 367 pp., 1983.

Bycenkiene, S., Dudoitis, V., and Ulevicius, V.: The Use of Trajectory Cluster Analysis to Evaluate the Long-Range Transport of Black Carbon Aerosol in the South-Eastern Baltic Region, Advances in Meteorology, 10.1155/2014/137694, 2014.

Cheng, I., Zhang, L., Blanchard, P., Dalziel, J., and Tordon, R.: Concentration-weighted trajectory approach to identifying potential sources of speciated atmospheric mercury at an urban coastal site in Nova Scotia, Canada, Atmospheric Chemistry and Physics, 13, 6031-6048, 10.5194/acp-13-6031-2013, 2013.

Dekhtyareva A., Edvardsen K., Holmén K., Hermansen O. & Hansson H.-C.: Influence of local and regional air pollution on atmospheric measurements in Ny-Ålesund, International Journal of Sustainable Development and Planning, 11, 4, 578–587, 2016.

DeMott, P. J., Prenni, A. J., Liu, X., Kreidenweis, S. M., Petters, M. D., Twohy, C. H., Richardson, M. S., Eidhammer, T., and Rogers, D. C.: Predicting global atmospheric ice nuclei distributions and their impacts on climate, Proceedings of the National Academy of Sciences of the United States of America, 107, 11217-11222, 10.1073/pnas.0910818107, 2010.

DeMott, P. J., Prenni, A. J., McMeeking, G. R., Sullivan, R. C., Petters, M. D., Tobo, Y., Niemand, M., Mohler, O., Snider, J. R., Wang, Z., and Kreidenweis, S. M.: Integrating laboratory and field data to quantify the immersion freezing ice nucleation activity of mineral dust particles, Atmospheric Chemistry and Physics, 15, 393-409, 10.5194/acp-15-393-2015, 2015.

DeMott, P. J., Hill, T. C. J., Petters, M. D., Bertram, A. K., Tobo, Y., Mason, R. H., Suski, K. J., McCluskey, C. S., Levin, E. J. T., Schill, G. P., Boose, Y., Rauker, A. M., Miller, A. J., Zaragoza, J., Rocci, K., Rothfuss, N. E., Taylor, H. P., Hader, J. D., Chou, C., Huffman, J. A., Poschl, U., Prenni, A. J., and Kreidenweis, S. M.: Comparative measurements of ambient atmospheric concentrations of ice nucleating particles using multiple immersion freezing methods and a continuous flow diffusion chamber, Atmospheric Chemistry and Physics, 17, 11227-11245, 10.5194/acp-17-11227-2017, 2017.

Hande, L. B. and Hoose, C.: Partitioning the primary ice formation modes in large eddy simulations of mixed-phase clouds, Atmospheric Chemistry and Physics, 17, 14105-14118, 10.5194/acp-17-14105-2017, 2017.

Heidam, N. Z., Wahlin, P., and Christensen, J. H.: Tropospheric gases and aerosols in northeast Greenland, Journal of the Atmospheric Sciences, 56, 261-278, 10.1175/1520-0469(1999)056<0261:tgaain>2.0.co;2, 1999.

Hiranuma, N., Adachi, K., Bell, D. M., Belosi, F., Beydoun, H., Bhaduri, B., Bingemer, H., Budke, C., Clemen, H. C., Conen, F., Cory, K. M., Curtius, J., DeMott, P. J., Eppers, O., Grawe, S., Hartmann, S., Hoffmann, N., Hohler, K., Jantsch, E., Kiselev, A., Koop, T., Kulkarni, G., Mayer, A., Murakami, M., Murray, B. J., Nicosia, A., Petters, M. D., Piazza, M., Polen, M., Reicher, N., Rudich, Y., Saito, A., Santachiara, G., Schiebel, T., Schill, G. P., Schneider, J., Segev, L., Stopelli, E., Sullivan, R. C., Suski, K., Szakall, M., Tajiri, T., Taylor, H., Tobo, Y., Ullrich, R., Weber, D., Wex, H., Whale, T. F., Whiteside, C. L., Yamashita, K., Zelenyuk, A., and Mohler, O.: A comprehensive characterization of ice nucleation by three different types of cellulose particles immersed in water, Atmospheric Chemistry and Physics, 19, 4823-4849, 10.5194/acp-19-4823-2019, 2019.

Hoose, C. and Mohler, O.: Heterogeneous ice nucleation on atmospheric aerosols: a review of results from laboratory experiments, Atmospheric Chemistry and Physics, 12, 9817-9854, 10.5194/acp-12-9817-2012, 2012.

Hsu, Y. K., Holsen, T. M., and Hopke, P. K.: Comparison of hybrid receptor models to locate PCB sources in Chicago, Atmospheric Environment, 37, 545-562, 10.1016/s1352-2310(02)00886-5, 2003.

Jeong, U., Kim, J., Lee, H., Jung, J., Kim, Y. J., Song, C. H., and Koo, J. H.: Estimation of the contributions of long range transported aerosol in East Asia to carbonaceous aerosol and PM concentrations in Seoul, Korea using highly time resolved measurements: a PSCF model approach, Journal of Environmental Monitoring, 13, 1905-1918, 10.1039/c0em00659a, 2011.

Kanji, Z. A., Ladino, L. A., Wex, H., Boose, Y., Burkert-Kohn, M., Cziczo, D. J., and Kramer, M.: Overview of Ice Nucleating Particles, Ice Formation and Evolution in Clouds and Precipitation: Measurement and Modeling Challenges, 58, 10.1175/amsmonographs-d-16-0006.1, 2017.

Knopf, D. A., Alpert, P. A., Wang, B., and Aller, J. Y.: Stimulation of ice nucleation by marine diatoms, Nature Geoscience, 4, 88-90, 10.1038/ngeo1037, 2011.

Lee, C., Sultana, C. M., Collins, D. B., Santander, M. V., Axson, J. L., Malfatti, F., Cornwell, G. C., Grandquist, J. R., Deane, G. B., Stokes, M. D., Azam, F., Grassian, V. H., and Prather, K. A.: Advancing Model Systems for Fundamental Laboratory Studies of Sea Spray Aerosol Using the Microbial Loop, Journal of Physical Chemistry A, 119, 8860-8870, 10.1021/acs.jpca.5b03488, 2015.

Lehahn, Y., Koren, I., Schatz, D., Frada, M., Sheyn, U., Boss, E., Efrati, S., Rudich, Y., Trainic, M., Sharoni, S., Laber, C., DiTullio, G. R., Coolen, M. J. L., Martins, A. M., Van Mooy, B. A. S., Bidle, K. D., and Vardi, A.: Decoupling Physical from Biological Processes to Assess the Impact of Viruses on a Mesoscale Algal Bloom, Current Biology, 24, 2041-2046, 10.1016/j.cub.2014.07.046, 2014.

Lisok, J., Markowicz, K. M., Ritter, C., Makuch, P., Petelski, T., Chilinski, M., Kaminski, J. W., Becagli, S., Traversi, R., Udisti, R., Rozwadowska, A., Jefimow, M., Markuszewski, P., Neuber, R., Pakszys, P., Stachlewska, I. S., Struzewska, J., and Zielinski, T.: 2014 iAREA campaign on aerosol in Spitsbergen - Part 1: Study of physical and chemical properties, Atmospheric Environment, 140, 150-166, 10.1016/j.atmosenv.2016.05.051, 2016.

Lumpkin, R. and Johnson, G. C.: Global ocean surface velocities from drifters: Mean, variance, El Nino-Southern Oscillation response, and seasonal cycle, Journal of Geophysical Research-Oceans, 118, 2992-3006, 10.1002/jgrc.20210, 2013.

Mamouri, R. E. and Ansmann, A.: Estimated desert-dust ice nuclei profiles from polarization lidar: methodology and case studies, Atmospheric Chemistry and Physics, 15, 3463-3477, 10.5194/acp-15-3463-2015, 2015.

Mamouri, R. E. and Ansmann, A.: Potential of polarization lidar to provide profiles of CCN- and INP-relevant aerosol parameters, Atmospheric Chemistry and Physics, 16, 5905-5931, 10.5194/acp-16-5905-2016, 2016.

Mansour, K., Decesari, S., Bellacicco, M., Marullo, S., Santoleri, R., Bonasoni, P., Facchini, M. C., Ovadnevaite, J., Ceburnis, D., O'Dowd, C., and Rinaldi, M.: Particulate methanesulfonic acid over the central Mediterranean Sea: Source region identification and relationship with phytoplankton activity, Atmospheric Research, 237, 10.1016/j.atmosres.2019.104837, 2020a.

Mansour, K., Decesari, S., Facchini, M. C., Belosi, F., Paglione, M., Sandrini, S., Bellacicco, M., Marullo, S., Santoleri, R., Ovadnevaite, J., Ceburnis, D., O'Dowd, C., Roberts, G., Sanchez, K., and Rinaldi, M.: Linking Marine Biological Activity to Aerosol Chemical Composition and Cloud-Relevant Properties Over the North Atlantic Ocean, Journal of Geophysical Research-Atmospheres, 125, 10.1029/2019jd032246, 2020b.

Masiol, M., Squizzato, S., Rich, D. Q., and Hopke, P. K.: Long-term trends (2005-2016) of source apportioned PM2.5 across New York State, Atmospheric Environment, 201, 110-120, 10.1016/j.atmosenv.2018.12.038, 2019a.

Masiol, M., Squizzato, S., Cheng, M. D., Rich, D. Q., and Hopke, P. K.: Differential Probability Functions for Investigating Long-term Changes in Local and Regional Air Pollution Sources, Aerosol and Air Quality Research, 19, 724-736, 10.4209/aaqr.2018.09.0327, 2019b.

[revised manuscript text omitted]

Wilson, T. W., Ladino, L. A., Alpert, P. A., Breckels, M. N., Brooks, I. M., Browse, J., Burrows, S. M., Carslaw, K. S., Huffman, J. A., Judd, C., Kilthau, W. P., Mason, R. H., McFiggans, G., Miller, L. A., Najera, J. J., Polishchuk, E., Rae, S., Schiller, C. L., Si, M., Temprado, J. V.,

Whale, T. F., Wong, J. P. S., Wurl, O., Yakobi-Hancock, J. D., Abbatt, J. P. D., Aller, J. Y., Bertram, A. K., Knopf, D. A., and Murray, B. J.: A marine biogenic source of atmospheric ice-nucleating particles, Nature, 525, 234-+, 10.1038/nature14986, 2015.

Whale, T. F., Wong, J. P. S., Wurl, O., Yakobi-Hancock, J. D., Abbatt, J. P. D., Aller, J. Y., Bertram, A. K., Knopf, D. A., and Murray, B. J.: A marine biogenic source of atmospheric ice-nucleating particles, Nature, 525, 234-+, 10.1038/nature14986, 2015.

---

## Author Comment (AC2) · 29 Mar 2021

*Response to Referee #2*

First of all, the authors thank the referee for submitting helpful and meaningful comments, which lead to improvements and clarifications within the manuscript.

Below, we provide our point-by-point responses. For clarity and easy visualization, the Referee's comments (**RC**) are shown from here on in black. The authors' responses (**AR**) are in blue color below each of the referee's statement. In addition to the responses to referees' comments, we further modified the manuscript to increase its clarity and readability. Abstract and conclusions were mostly rewritten. The Section on the ice nucleation active site density ($n_s$) was removed; $n_s$ was substituted by the Activate Fraction (AF) parameter in the discussion. The Results section was re-organized for major clarity and separated from the Discussion Section. All the changes can be checked in the track change version of the manuscript, where the new text is highlighted in yellow color. We introduce the revised materials in green color along/below each one of your response (otherwise directed to the Track Changes version manuscript). All references are available in the end of this AR document.

General Comment:**RC:** In this manuscript the INP concentration in Ny-Ålesund (Svalbard) was evaluated in two different seasons and in two different ice nucleation modes, using two different offline techniques. Given the high importance of the Arctic and the low number of studies focusing on INPs, the present study is useful for the ice nucleation, aerosol, and cloud microphysics communities. Although the present results are very valuable, is not completely clear how this study differs from previous studies conducted at the same Arctic location. The manuscript is poorly written as the authors did not pay attention to several important details as listed below. I encourage the authors to improve the quality of the manuscript taking into account the Major, Minor, and Technical comments.

**AR:** The authors appreciate these general remarks regarding our manuscript by Referee #2. Below, we provide our point-by-point responses. To reflect our changes and articulate what is truly presented in the revised version paper, the authors have decided to change the title of manuscript to "**Ice-nucleating particle concentration measurements from Ny-Ålesund during the Arctic Spring-Summer in 2018**". We admit that we have made some insufficient discussions, leading some of our data interpretations in an original manuscript to be speculative and unclear. Based on the peer-review comments, we removed/modified them to motivate the research as described below in our individual responses.

Major Comments:

**RC:** 1. I do not see a clear difference between the content of the Abstract and the Conclusions. These two sections need to be different with the Abstract including more concise and quantitative information.

**AR:** We have revised our abstract as well as the conclusion to reflect all of our major revisions (please see the Track Changes version paper).

**RC:** 2. The English needs to be significantly improved. The way the document is written makes it very difficult to read it in some parts.

**AR:** The revised manuscript has been carefully checked by the authors.

**RC:** Also, the document seems disorganized, and therefore, I encourage the authors to improve this part. The authors need to be more precise, more quantitative, and improve the statistical analysis.

**AR:** We have improved the manuscript organization by adding a Discussion Section. Moreover, we have reorganized the Results Section for clarity. Now the Results Section reports only the results of our observations and elaborations, while implications of the results are addressed in the Discussion.

**RC:** 3. There is key information missing in the text (e.g., technical details, references, units, correlation coefficients, etc.). See below.

**AR:** Thank you for pointing this out. The authors found it as invaluable guidance. We considered it as addressed below.

*RC:* 4. Lines 94-96: What is different or what is novel in the present study compared to Wex et al. (2019) and Hartmann et al. (2019)?

*AR:* The revised Abstract and Conclusions Sect. explain the novelty and significance of this work in a better way. In this study we present parallel observations of immersion and condensation INPs, which was never achieved in the Arctic. One of our major findings is indeed related to the different behaviour of aerosol particles sampled at GVB under different ice nucleation modes:

"We considered many factors that could potentially explain the observed difference (Sect. 4) and conclude that the different ice nucleation mechanisms probed by the two techniques (condensation freezing, for DFPC, and immersion freezing, for WT-CRAFT) is an undeniable reason. While differences in the sampling resolution and overall measurement uncertainties have partly contributed to the observed offset, it seems conclusive to address there is ice nucleation mode dependent INP propensity at GVB in 2018 at least. Any future investigations regarding INP compositions and more controlled-study focusing on condensation vs. immersion freezing on identified compositions will lead to further findings to settle this issue".

Another significance of our work is the observation of no substantial seasonal variation in *n*INP accompanied by generally higher ice nucleation efficiency (AF) and a clear enhancement in the contribution of coarse INPs in summer with respect to spring:

"This study also offered unique data examining the seasonality of INPs in the Arctic with respect to *n*INP and AF. Both condensation and immersion INP datasets did not indicate a marked *n*INP seasonal trend. We report a statistically significant spring to summer enhancement in *n*INP only for a narrow range of *Ts* (-17.5 to -21.5°C) and the associated *n*INP enhancement never exceeded a factor of three. On the other hand, the AF of atmospheric aerosol particles from GVB presents a statistically significant spring to summer increase almost independent on the probed *T*, reaching up to ca. 6 times at *T* of -19°C. A clear seasonal evolution of the super-micrometer INP contribution was observed by DFPC. Such contribution was around 20% in spring (with the highest 32% at -15 °C) and increasing markedly in summer and at high *Ts* (45% at *T* of -22°C and 65% at *T* of -15°C)".

Finally, we also provide evidence for the different contribution of local vs. long range, natural vs. anthropogenic and terrestrial vs. marine aerosol particle sources to the INP burden at the study location, contributing to improve our current understanding of INP dynamics over the Arctic.

Minor Comments:

*RC:* Line 43: Add a reference after "amplification".
*AR:* The authors agree. Serreze and Barry (2011) is now added.

*RC:* Line 46: Add a reference after "detail". How about Abbatt et al. (2019)?
*AR:* This is a valid suggestion. Murray et al. (2021) is now added.

*RC:* Line 53: "sufficient numbers". Please be clear.
*AR:* The authors clarified the sentence by rephrasing it to, "Thus, the presence of aerosol particles that can trigger heterogeneous ice nucleation (ice-nucleating particles, INPs, hereafter) in the Arctic atmosphere can potentially have substantial impacts on precipitation formation, cloud radiative properties and climate (Solomon et al., 2018; Murray et al., 2021)".

*RC:* Line 56: "transport dynamics". Please be clear.
*AR:* We meant to say "For these reasons, the current inadequate understanding of INP sources, transport and removal processes in the Arctic region…" After the revision, the authors found this part is irrelevant to our introduction. Thus, we decided to remove this sentence.

*RC:* Line 57: Add a reference after "budgets".
*AR:* deBoer et al. (2014) and Morrison et al. (2012) would be appropriate references. However, for the same reason addressed above, we decided to delete this sentence.

*RC:* Line 69: Define "T".
*AR:* Defined for temperature (*T*).

*RC:* Line 77: "tripling INP". Please be clear.
*AR:* We have rephrased this sentence to "Conen et al. (2016) measured *n*INP at a coastal mountain observatory in Northern Norway. During the summer, the authors observed that *n*INP (*T* of -15°C) in oceanic air tripled after about one day of passage over land".

*RC:* Line 88: "evidencing order of magnitude wise increase". Please be clear.
*AR:* The authors decided to use more straightforward language (Line 92-95): "In particular, Wex et al. (2019) observed an increase of *n*INP of more than one order of magnitude from spring to summer (e.g., ~14 times at T=-15°C) at GVB in 2012. Tobo et al. (2019) focused on two field campaigns held at Mt. Zeppelin, in July 2016 (six samples) and March 2017 (seven samples). They report *n*INPs at -20°C of about 0.01 L$^{-1}$ in spring and about 0.1 L$^{-1}$ in summer".

*RC:* Line 89: Add a reference after "ice".
*AR:* Wex et al. (2019) and Santl-Temkiv et al. (2019) are now added.

*RC:* Line 99: "The aerosol sampling was performed at the Gruvebadet observatory". Add a map showing it.
*AR:* This is a good suggestion. It is now offered in our new Fig. 1.

*RC:* Lines 110-111: "The sampling generally started in the morning, during the spring campaign, while it started typically in the afternoon during the summer campaign". What is the reason for this?
*AR:* This is a valid question. The variation in our sampling start time stems from only logistical reasons (e.g., not to disturb other activities taking place in each season). We now clarify this point in the text as "The sampling generally started in the morning during the spring campaign, while it started typically in the afternoon during the summer campaign (in coordination with other scheduled activities at GVB)."
In addition, we have carefully assessed the daily variation of aerosol particle concentrations in Spring and Summer (please see the figure below). As seen in the figure, the aerosol particle concentrations are consistent within the 25-75% percentile range. We hope the referee finds our method reasonable.

[Figure]

*Figure: Daily profiles of particle number concentration at GVB during spring (April) and summer (July) 2018.*

*RC:* Lines 110-111: Add a Table with the details of each sample from both techniques (e.g., initial time, final time, date, etc.).
*AR:* As suggested, the WT-CRAFT sampling details are now available in Table S1.

*RC:* Lines 113-119: When where the samples collected with the WT-CRAFT?
*AR:* Thanks for asking this. The authors initially intended to include some of these details (omitted concerning the manuscript length). All details regarding WT-CRAFT are now incorporated in our revised manuscript. We have clarified these in Sect. 2.1; "For the application of West Texas Cryogenic Refrigerator Applied to Freezing

Test system (WT-CRAFT) analysis, a total of 28 samples were collected from April 16 to August 15, 2018. Aerosol particles were collected using 47 mm membrane filters (Whatman, Track-Etched Membranes, 0.2 µm pore). Briefly, aerosol particle-laden air was drawn from a central total suspended particulate (TSP) inlet with a constant average inlet flow of 5.4 lpm (± 0.2 lpm standard deviation). We note that the TSP inlet is custom made, and is designed to operate with isokinetic and laminar flow at 150 lpm. From the central inlet, an 8 mm outside diameter stainless steel tube was directly connected to the filter sampler to intake a subset of air flow. More detailed conditions of our filter sampling, including sampling time stamps, air volume sampled through filter cross section, and the resulting HPLC water volume used to suspend aerosol particles for WT-CRAFT analysis, are summarized in Table S1. Below the filter sampler, the filtered-air was constantly pumped through a diaphragm pump (KnF, IP20-T). A critical orifice was installed upstream of the pump to ensure a constant volume flow rate and control the mass flow rate through the sampling line. A typical sampling interval was approximately 4 days with only one exception (i.e., 8 days for the sample collected starting on 26 May 2018)".

*RC:* Line 114: "0.2 µm pore size". Brand? Model?
*AR:* Whatman, Track-Etched Membranes, 0.2 µm pore (added in Line 128).

*RC:* Line 115: Define "TSP".
*AR:* Defined - total suspended particulate (TSP). Now in Line 129.

*RC:* Lines 118-119: If the flow rate of the WT-CRAFT is 150 lpm and the flow rate for the DFPC is 38.3 lpm, why the samples from the former one was 4 days and for the later one just 3-4 hours?
*AR:* For clarity, 150 LPM is the flow rate of the central laminar flow sampling inlet at GVB. From the central inlet, only a small amount of flow (~5.4 lpm) was bypassed to the WT-CRAFT filter sampler. The text has been substantially modified to clarify what was truly done at GVB in Sect. 2.1. Please see our Track-Changed manuscript. The authors should have clarified that only a subset of 150 lpm from the common inlet was directed towards our WT-CRAFT filter sampler. We hope these array the referee's misgivings.

*RC:* Line 121: The authors need to provide a brief description of the method.
*AR:* Added in Lines 143-148.

*RC:* Lines 128-129: I would rather add a small paragraph indicating how good is the agreement of the DFPC data compared to other techniques.
*AR:* As we point out in the revised version manuscript, the agreement depends on the analyzed aerosol particle type. For this reason, instead of addressing generally the agreement of DFPC with other techniques, we discuss in the manuscript the agreement of DFPC with WT-CRAFT in the present and previous deployments. Please, refer to the new Sect. 4.1 for details.

*RC:* Lines 141-142: "in a known volume". Specify the volume.
*AR:* Now provided in Table S1.

*RC:* Line 141-142. This is not very clear.
*AR:* The authors now clarified our INP estimation method in Sect. 2.2.3. We optimized our suspension water volume in the way the first frozen droplet correspond to 1 INP m$^{-3}$.
For the WT-CRAFT analysis, we first computed the $C_{INP}(T)$ value, which is the nucleus concentration in HPLC suspension (L$^{-1}$ water) at a given $T$ as described in Vali (1971). This $C_{INP}(T)$ value was calculated as a function of unfrozen fraction, $f_{unfrozen}(T)$ (i.e., the ratio of number of droplets unfrozen to the total number of droplets) as:

$$C_{INP}(T) = -\frac{\ln(f_{\text{unfrozen}}(T))}{V_d} \qquad (1)$$

in which, $V_d$ is the volume of individual droplets (3 $\mu$L). Next, we converted $C_{INP}(T)$ to $n\text{INP}(T)$. The cumulative $n$INP per unit volume of sample air, described in DeMott et al. (2017), was estimated as:

$$nINP(T) = C_{INP}(T) \times DF \times \frac{V_l}{V_{air}}$$
  (2)

where DF is a serial dilution factor (e.g., DF = 1 or 10 or 100 and so on). The sampled air volume ($V_{air}$) and the suspension volume ($V_l$) are now provided in Table S1.

*RC:* Line 151: No parametrizations were derived in this study.
*AR:* The referee is right. We only did estimation rather than parameterization. The new heading of this Section reads as "2.2.3 Derivation of INP atmospheric concentrations".

*RC:* Line 152: "concentration of ice nucleating particles (nINP)". This was defined in Line 69.
*AR:* Corrected.

*RC:* Line 164: "1.95 g cm-3". Add a reference.
*AR:* We have added the requested reference (Lines 212-214). Now the text reads: "The aerodynamic diameters measured by the APS were corrected to the volume equivalent diameters using an average particle mass density equal to 1.95 g cm$^{-3}$, assuming a mixture of different substances based on the findings from Lisok et al. (2016) and a dynamic shape factor of 1".

*RC:* Line 167: "air temperature, T". This was first used in Line 69.
*AR:* Corrected.

*RC:* Line 173: "GVB". Define it.
*AR:* Defined. Line 15.

*RC:* Line 173-174: "on filters collected". The authors need to be clear on what filters and how the particle were collected.
*AR:* This part was extended as follows: "The chemical analysis of major and trace ion species, used in this work as aerosol source tracers, was accomplished on Teflon filters (PALL Gelman) collected at GVB by means of a TECORA Skypost sequential sampler equipped with a PM10 sampling head and operating at 2.3 m$^3$ h$^{-1}$ (EN 12341)".

*RC:* Line 177: "C2O4-2". Fix it.
*AR:* Fixed.

*RC:* Line 179: "Mg2+, Ca2+". Fix it.
*AR:* Fixed.

*RC:* Line 211 and along the text: The authors used "at Ny-Ålesund", "GVB", and "Gruvebadet". Please be consistent.
*AR:* Corrected, we now referred to the sampling location as GVB through the whole manuscript.

*RC:* Line 217-219: "t", "L", "Ct" and "Dijt" should be in italics.
*AR:* Thanks for noticing, they have been checked and corrected accordingly.

*RC:* Lines 241-242: "The observed offset may derive from the different time resolutions of the sampling for INP analyses, as well as from uncertainties in sampling activities and/or measurement uncertainties". How about the particle size analyzed in both techniques? The pore size of the filters used is different in each technique.
*AR:* The filter pore size does not influence the lower size cut-off of the sampled particles for the reason described below. In the process of particles filtration from air, particles smaller than the nominal filter porosity are retained onto the filters, differently from what happens in fluid. This happens because the filters capture particles by different mechanisms: inertial impact, interception and Brownian diffusion. As an

example Willeke and Baron (1993) show that cellulose acetate/nitrate membranes (0.45 micron porosity) capture particles in all the aerosol size spectrum with an efficiency range of >99.8 - >99.99 at face filtration velocities 1-100 cm s$^{-1}$, respectively. Therefore, the difference in the filter nominal pore size might not substantially impact our INP results and explain the observed INP concentration difference. The authors appreciate the referee for bringing up this point though. While large particles are typically assumed to act as active INPs for their surface, a potential contribution of small soluble particles cannot be ruled out. People in the INP community should keep this in our mind.

*RC:* Line 242: In the Hiranuma et al. (2015) paper the 2 techniques were no used.
*AR:* The authors meant to refer to uncertainties in each technique, which are reported over these two inter-comparison papers (DFPC in both and WT-CRAFT in H19). Nonetheless, the authors agree that H15 is not necessary here. It has been removed.

*RC:* Line 244: Line 43: Add a reference after "questionable".
*AR:* We have extended the discussion on ice nucleation modes, mainly in the Introduction. Now the manuscript reads (Lines 59-61): "The distinction between condensation-freezing and immersion-freezing is still matter of debate (Dymarska et al., 2006). Nevertheless, the recent results of Wex et al. (2014) and Hiranuma et al. (2015) suggest that they might be the same process".

*RC:* Lines 250: Are you sure the authors clearly distinguished between the 2 modes in Wex et al. (2014)?
*AR:* Yes. We have double-checked the referred publications:
In Wex et al. (2014) we read: "The above-described results support the hypothesis that condensation and immersion freezing (i.e., the ice nucleation of an insoluble core immersed in a haze particle or in a diluted droplet) might basically be the same process, with the only distinction that a freezing point depression has to be accounted for in the subsaturated regime (i.e., for the haze particles)."
In Hiranuma et al. (2015): "Two types of immersion freezing experiments are presented. One set of experiments was designed to fully activate droplets before ice formation (that is, $T_{droplet}$>-10 °C), whereas another set was aimed to examine immersion mode freezing at or during droplet formation (that is, $T_{droplet}$~$T_{IN}$). The good agreement between the two approaches (see Fig. 2) demonstrates that the ice-nucleating efficiency is similar for immersion and condensation freezing for MCC, supporting the idea that those two mechanisms are in fact the same."

*RC:* Line 252: "different aerosol types yielded different results". Again, is it not the size measured by both techniques? Depending on the aerosol type, their size distribution changes, and therefore, the particle collection efficiency of each technique.
*AR:* Discussed above.

*RC:* Lines 257-259: "median 115". From Figure 1, the value seems to be close to 90 m-3 instead of 115 m-3.
*AR:* The old version of Figure 1 referred to the whole DFPC dataset, while in the text we presented the data divided in spring and summer campaign. The revised Figure, Figure 2, presents the DFPC data divided by season. So everything should be consistent now. Our apologies for inconsistency and confusion.

*RC:* Line 259: "33-135 (median 77), 18-107 (45) and 6-66 (20) m-3". This is not shown in Figure 1. Please add them to the Figure.
*AR:* In the revised version, all the single data points are presented in the Figure (Fig. 2).

*RC:* Line 264: "24-9082". What is the reason of such large variability?
*AR:* It is due to the outlier data in the last sample of the campaign. This is discussed later on in Sect. 3.4.

*RC:* Line 274: "range 5-10, 10-30 and 30-70". Units are missing.
*AR:* Added. This part was removed and merged with the Introduction section (Lines 70-98).

*RC:* Lines 281-182: "we can conclude that the results of the present study are generally consistent with literature". Add literature data to Figure 1.
*AR:* We have decided to extend this discussion – we now provided a summary of past results of Arctic *n*INPs and used instruments and *T* ranges along with all references in our new Table 1. Associated discussion also appears in Sect. 3.1.

*RC:* Line 282: Add Wex et al. (2019) data to Figure 1.
*AR:* This is a good suggestion. Added accordingly to Fig.2.

*RC:* Line 294: "similar". By how much?
*AR:* This sentence was removed as discussing the agreement of DFPC and other immersion freezing techniques (apart WT-CRAFT) is beyond the purposes of this manuscript. For reference, the agreement reported in McCluskey et al. (2018), between DFPC and CSU cold stage, is within 2.5 times.

*RC:* Lines 300-302: I don't get what is unique here.
Line 304: "shows the bimodal activation with a hump feature at T above -15 °C.". Really?
Lines 304-305: "may be due to marine biogenic aerosols". Are not you sampling in a marine environment? Why is this surprising?
*AR:* The authors agree that this part is too speculative. Based on the suggestion made by Referee #1, the part of the text, which the above 4 comments refer to, and our former Fig. 2 were removed.

*RC:* Line 311: "suggesting that the dominant INP sources may be located at long distances". This is rather speculative. Make a link with the evidence you show later.
*AR:* This part was revised, and we believe the clarity is improved (Lines 340-344): "A small contribution from coarse INPs characterized the spring campaign (~20%), suggesting that the dominant INP sources may be located at long distances (scale of the order of 100s-1000s km), with consequent depletion of the largest particles during transport, due to their higher gravitational deposition velocities. This result is consistent with previous works highlighting the contribution of long range transport from lower latitudes during the Arctic spring (Shaw, 1995; Heidam et al., 1999; Stohl, 2006)".

*RC:* Lines 313-314: "likely resulting from the activation of local sources after snow and ice melting". Why coarse particles can be transported long distances in spring and not in summer.
*AR:* Perhaps, our previous sentence was not clear enough to infer that aerosol particles derive mainly from long distance sources at GVB in Spring, therefore there are few coarse particles. In summer, on the other hand, the contribution of coarse particles is dominant, which is hardly reconcilable with long-range transport. This is consistent with literature, describing the phenomenon of the Arctic Haze as cited in the text. We have substantially altered our paragraph here: "A small contribution from coarse INPs characterized the spring campaign (~20%), suggesting that the dominant INP sources may be located at long distances (scale of the order of 100s-1000s km), with consequent depletion of the largest particles during transport, due to their higher gravitational deposition velocities. This result is consistent with previous works highlighting the contribution of long range transport from lower latitudes during the Arctic spring (Shaw, 1995; Heidam et al., 1999; Stohl, 2006). During the summer campaign, a significant ($p<0.005$) increase of the contribution of coarse INPs was observed (i.e., 65% at $T$ = -15°C), resulting from the contribution of locally emitted aerosol particles (see Sect. 3.6) in part from the surface exposed to the air after snow and ice melting. While these coarse INP fraction estimation, presented in Table 2, involves substantial uncertainties, the same trend is inferred by the particle size distribution measurements, which show a significant ($p<0.01$) enhancement of coarse particles contribution in summer (median 30%) with respect to the spring time (median 16%) (Fig. S2). The increase of coarse INP contribution, from spring to summer time, is progressively more pronounced with increasing activation $T$."

*RC:* Line 317: Mason et al. (2016) found a large contribution from the coarse particles with most of the samples collected in Alert during the spring.

*AR:* Mason et al. (2016) present the results of measurements performed at Alert between 29 March and 23 July 2014. This period covers both spring and summer. Unfortunately, no information on the seasonal evolution of the fine and coarse INPs is provided. In any case, the analogy with the size distribution in our summer samples is noteworthy and we reported it in our manuscript. We have modified the revised version as follows (Lines 351-354), "A similar coarse fraction dominated INP population was reported by Mason et al. (2016) for measurements performed between 29 March to 23 July 2014 at the Alert Arctic station, with increasing coarse INPs contribution as a function of the activation *T*. Our results are unique compared to past studies as our measurements and data support the increase of coarse INP contribution during the meteorological season transition from spring to summer with increasing activation *T*".

*RC:* Line 319: The Creamean et al. (2018) samples were collected in spring.
*AR:* Thank you for this useful comment. We now mention it in our Sect. 3.3; "Analogously, Si et al. (2018) and Creamean et al. (2018) reported a higher ice nucleation efficiency for super-micrometer particles sampled at Arctic stations. The above cited papers report data collected in both summer (Si et al., 2018) and spring (Creamean et al., 2018)."

*RC:* Lines 321-322: Should "cm-3" be "m-3"?
*AR:* $cm^{-3}$ is the correct unit as particle number concentration is several times higher than INP concentration.

*RC:* Line 325: "at lower temperatures". Should it be higher?
*AR:* Thanks for noting this. In any case, the sentence was removed.

*RC:* Line 332: Add a reference after "melting".
*AR:* Santl-Temkiv et al. (2019) is now added.

*RC:* Line 354: "at GVB (2012)". Add the corresponding paper.
*AR:* This part was merged in the new Discussion Section; the proper citation was added.

*RC:* Line 360: Why wind direction was not included?
*AR:* The back-trajectories used for the spatial attribution of INP sources fully cover the requested variable (wind direction). For this reason, we decided not to consider wind direction in the meteorology analysis.

*RC:* Line 361: "were often associated to a reduction". It is not very obvious from the Figure. This is a qualitative conclusion.
*AR:* We have added a quantitative consideration on the significance of the correlation. In addition, the Pearson R values and corresponding significance levels (P) where added in each plot of Figure S5.

*RC:* Line 361: "the exception of precipitation events". Add the r2.
*AR:* We have added it in Figure S5.

*RC:* Lines 362-371: I don't think this is really necessary as it adds too little to the discussion and does not help at all to support the data.
*AR:* The authors agree. This part has been removed as suggested.

*RC:* Line 372: "covariate". Add the r2.
*RC:* Line 374: "more accentuated" and "significant correlations". Add the r2.
*AR:* We added the Pearson's R coefficient as requested.

*RC:* Line 385: "showed a significant". Add the r2.
*AR:* Instead of R2, we are presenting the results of a T test for two groups of data. This part of the text was removed and was substituted with the new Section on the AF.

*RC:* Lines 388-389: "mainly related to long-range transport of anthropogenic aerosol particles from lower latitudes (Arctic haze)". No evidence provided.

*AR:* This part was substituted with the new Section on AF. We have considered the reviewer's comments in writing the new AF Section and the new Discussion Section. Proves that spring time aerosol particles are associated to long range transport from outside the Arctic are presented in Sections 3.2, 3.6.1 and 3.6.2. In the Discussion these findings are summarized as follows:

"The chemical tracer correlation analysis, the ground contribution analysis and the above presented considerations on the different contributions of sub- and super-micrometer INPs in spring and summer time suggest that the main sources of spring time INPs measured at GVB may be located outside the Arctic. They are deemed to derive from the lower latitude regions together with anthropogenic aerosols during the Artic haze (Heidam et al., 1999; Stohl, 2006). Conversely, the summer time aerosol particles population is more related to local (Arctic) sources".

*RC:* Line 396: "for all the activation temperatures". Just 2 temperatures are shown in the Figure.

*AR:* Now, Fig. 3 shows data for all the temperatures, and Fig. 5 reports data at 3 temperatures.

*RC:* Lines 396-397: "with the exception of the coldest one (T = -25 °C)". This information is not provided. Lines 398-399: "only a minority of samples (<50%)". 30-40% is a minority?

*AR:* Based on the comment of Referee #1, we have substantially revised this section (to activated fraction discussion), and these sentences have been removed since no longer relevant.

*RC:* Lines 399-401: I don't get it.

*AR:* The maximum increase was observed at $T$ of -19°C. This part was rephrased as "Differently from the DFPC data, the spring to summer AF increase from WT-CRAFT data had its maximum at $T$ = -19°C (5.7), with the minimum value obtained at $T$ = -25°C (1.4) (Fig. S4)" in Lines 407-408.

*RC:* Lines 409-410: "suggesting that the INP population over the Arctic in summer originates from a combination of mineral dust and marine aerosol particles". If I understood correctly, this figure contains the data from both summer and spring, therefore, such conclusion cannot be drawn from the data reported in the figure. Please separate the data sets into summer and spring.

*AR:* This part was removed.

*RC:* Line 417: "nINPDFPC correlated". Add r2 and p.

*RC:* Line 422 and 424: "anticorrelation" and "significant". Add r2.

*AR:* These info are reported in Table 2 (and 3).

*RC:* Figure 2. It seems you are comparing apples with oranges. Why are the y axis scale different? This figure and its discussion seems to be useless.

*AR:* All this part was removed.

*RC:* Figure 4. The authors need to separate them between summer and spring.

*AR:* This was done, but we reported a different metric, the activated fraction (AF) instead of $n_s$.

Technical comments:

*RC:* 1. Line 62: Delete the "dot" after "temperatures".

*AR:* Corrected.

*RC:* 2. Line 100, Line 168: "Km" should be "km".

*AR:* Corrected.

*RC:* 3. Lines 116-117: "diaphragm pump". Please add the details about the pump.

*AR:* Added (KnF, IP20-T).

*RC:* 4. Line 272: "immersion mode freezing". Be clear.

*AR:* Good point - all details on the technique used and $T$ ranges are now compiled and reported in Table 1.

*RC:* 5. Line 273: ").". Fix it.

*AR:* Fixed. We have divided sentences with "." with some additional information – "Conen et al. (2016) measured *n*INP at a coastal mountain observatory in Northern Norway. During the summer, the authors observed that *n*INP (*T* of -15°C) in oceanic air tripled after about one day of passage over land. Both marine and terrestrial INP sources were identified by Creamean et al. (2018) in the Northern Alaskan Arctic during spring".

*RC:* 6. Line 280: "etc...". It should be one dot.

*AR:* Corrected.

*RC:* 7. Line 282: "Both the datasets discussed". Be clear.

*AR:* Corrected. both the DFPC and WT-CRAFT datasets

*RC:* 8. Line 287: "while both our datasets". Be clear.

*AR:* Corrected.

*RC:* 9. Line 293: "ranged 0.4-15 and 2-40 m-3". At what temperatures?

*AR:* Instead of providing individual temperatures, we now enlisted concentrations and *T* ranges for all previously published data in Table 1.

*RC:* 10.Line 361: Figure S2 is called before Figure S1.

Fixed

*RC:* 11.Line 436: Figure S3 is called before Figure S1.

Fixed

*RC:* 12.The citing format is wrong and needs to be fixed (e.g. missing spaces between references and when multiple references are cite, they are not organized chronologically).

*AR:* We apologize for not checking on this. It is now all fixed.

*RC:* 13.The format of the units if not uniform. For example in some cases the authors used "X ° C" but in other cases "X°C" is used. Please be consistent.

*AR:* Corrected.

*RC:* 14.Figure S1. Change the color of the "snow lines" as it is not clearly distinguishable from the white background.

*AR:* Modified (the Figure is the new Fig. 6).

*RC:* 15.How much time passed from sampling until the actual INP analyses? Add this information to the text.

*AR:* DFPC analyses where completed within December 2018 (spring samples) and February 2019 (summer ones). All WT-CRAFT measurements were completed by July 5th 2019. The info was added to the text (Sections 2.2.1 and 2.2.2). We note that Beall et al. (2020) recently found a decrease in $n_{INP}$ depending on the storage method/*T*s and suggested correction factors for the *T* range of -7 to -17°C. As both DFPC and WT-CRAFT analysed $n_{INP}$ beyond that T-range, we did not apply any correction for this study. This discussion is now given in Sect. 4.1.

*RC:* 16.Figure S2. The nINP in the middle left panel should be in blue. The nINP in the bottom left panel should be in black. Add more details to the Figure caption.

*AR:* The Figure was corrected and the caption updated.

*RC:* 17.Figures S2. I am not sure if it makes sense to correlate precipitation and INP concentration when using the WT-CRAFT based on the low time resolution i.e., 4 days.

*AR:* We agree and in fact we did not derive strong conclusion from that, we just limited to show the data for completeness of information. The correlation is indeed clearer with DFPC data.

**References used in AR**

[revised manuscript text omitted]

---

## Referee Report (RR1)

**2nd Referee report on "Ice-nucleating particle concentration measurements from Ny-Ålesund during the Arctic Spring-Summer in 2018" by Rinaldi et al.**

The authors have addressed most of the previous comments and improved the manuscript. However, I would like to suggest some further revisions to improve the manuscript before it can continue the review process.

**Main comment**

The number of samples this study is based on is small and might not allow to characterize the population of INP. It needs to be highlighted in the abstract and conclusion that results are preliminary because of the small sample size. Findings in the abstract and conclusion should be limited to the strong signals that are expected to be reproduceable in future investigations. For less clear results it should be stressed that more observations are needed.
The limitations of the applied analysis due to the size of the available dataset should be stated clearly in each section. This includes stating the number of measurements used and the assumption on the structure of the data, e.g. normal distribution of nINP (not typical) to compare spring to summer concentrations or contribution of coarse and fine particle fraction to nINP.
Because correlation analysis is prominent throughout the paper, I suggest to add a subsection detailing the statistical analysis, including how significance in dependence of number of samples is determined (t-test), to the Method Section. This would help to understand, e.g. why correlation coefficients are high and why 0.5 is significantly non-zero for DFPC but 0.8 can be not significant for WT-CRAFT (Tabs.3 and 4).

**Specific comments**

Line 27f, 101f, 609ff.: The difference in nINP due to two different ice nucleation modes does not emerge from the data. To make this conclusion, filter samples from one sampling setup should be analysed with both DFPC and WT-CRAFT. More plausible are differences in the samples used by the two methods.

Line 29: name the "several important indications"

Line 29f: This is inconsistent with what is reported on line 419ff

Line 31: specify "subset of our data"

Line 33: Explain how higher AF can be interpreted as larger freezing efficiency of large particles.

Line 34: inconsistent to line 417 where it is stated that no clear relation emerges between nINP and meteorology.

Line 48: what other mechanisms beside Bergeron-Findeisen play an important part?

Line 56, 57: Clarify the difference between water vapour deposition and water vapour condensation.

Line 60f.: If condensation and immersion are the same process how can the difference in nINP between DFPC and WT-CRAFT be explained by it? To make this case, provide evidence that the two mechanisms can be different.

Line 96: Why are Santl-Temkiv et al. and Wex et al. cited for the interpretation of data in Tobo et al. and not the Tobo et al. paper itself?

line 104f: Explain why INP measurements are the key to verify that immersion freezing is the most relevant ice nucleation mechanism in Arctic mixed-phase clouds and how such measurements can be used to do that.

Line 117ff.: state the number of samples collected in spring, summer, PM1, PM10.

Line 128ff.: on line 131 it says the 5.4lpm are a subset of the airflow through the TSP inlet. Specify the total flow through the inlet and describe more clearly how the 5.4lpm are extracted from the higher total flow. Was there a pressure drop in the flow from which the subset flow was taken? Was this considered to determine the sample flow? Was there an online measurement of the flow through the membrane filter (how are flow rates in Tab.S1 measured) and why did the flow vary from 4.8-5.6lpm between samples?

Line 165: what is meant by "reasonably matches"? The 95% confidence interval is inherent in the cited formula.

Line 178: point to sec. 2.2.3. for details on how nINP are estimated

Line 179: mention already here that half of each filter was used and explain how the used water volume was calculated. Volumes given in Tab.S1, row 2-6 are off by -0.1 to 0.2ml from calculated values.

Line 202ff: I couldn't find the size-range in Kanji et al., 2017. Pruppacher & Klett, 2010 section 9.2.3.2 suggest 0.1um as a lower size limit, which I would recommend. The choice needs to be motivated better. How does the lower size limit effect the results in sec. 3.3?

Line 215ff: Clarify what size-range was used to calculate AF. From line 203 it would seem that only APS data (0.5-10um) was used.

Line 246: provide a reference for the marine boundary layer height in the Arctic.

Line 270: It could be already mentioned here that nINP at -15°C, PM1 are used for this exercise.

Line 280: to be consistent throughout the paper, consider using nINP instead of Ct for the INP concentration.

Line 295: It could be explained here how the CHL correlation analysis and CWT were overlayed (Fig.8b).

Line 303: Can you provide an interpretation of the difference in slope? It could suggest that there is a specific type of high temperature INP, that was only detected in the DFPC measurement. Could storing the filters at different conditions cause such an effect?

Line 304ff: Consider referring to Tab.2 instead of listing the concentrations.

Line 307: give number of samples instead of "<50%".

Line 311: give T-range for the nINP, see comment on Tab.1

Line 312ff: Can the higher nINP from condensation mode measurements compared to immersion mode be confirmed from the studies listed in Tab.1?

Line 320: give number of samples that span the 3-200m-3 nINP range.

Line 323ff: The difference in nINP depending on the sensitivity of methods indicates that the INP population is highly variable and the variation is underestimated at the detection limits of methods.

Line 325: Clarify for what the nINP range is substantial. Below what concentration would it be negligible?

Line 328: give nINP at -22°C to compare to the range at -22°C on line 325.

Line 333ff: add nINP at -22°C measured at the cited locations

Line 344: As I understand a normal distribution is assumed to obtain the difference at p<0.005. I doubt that the dataset (<20 measurements per season) is large enough to obtain a valid distribution to perform statistics.

Line 346: From Tab.2 it can be seen that he median nINP,PM10 is the same in summer and spring. The nINP,PM1 is lower in summer thus contradicting the conclusion that coarse INP from exposed surface cause the difference. The higher nINP,PM1 in spring could indicate INP from the arctic haze.

Line 347: on line 344 the contribution of coarse INP is described as significant to a high level of certainty. How were the "substantial uncertainties" considered to estimate the significance of an increase in coarse INP contribution?

Line 349: On the PM1 filters not only particles larger than 0.5um are sampled, as was assumed for the fractions shown in Fig. S2. I suggest using the full size-range below 1um for the fine particle concentration to calculate the change in the particle population.

Line 350: The coarse INP fraction doesn't dominate. Looking at Tab.2, the fine fraction makes up 80% of nINP in spring and 50% in summer.

Line 352: As mentioned on line 350, Mason et al. also reported a spring to summer dataset, making the results not "unique".

Line 360: Repeat here how coarse and fine AF are calculated, i.e. particle size-range used and nINP coarse as difference of PM10-PM1 nINP. Also mention it in the caption of Fig.3. What particle size-range was used for the WT-CRAFT (sampling through a TSP inlet) AF?

Line 360, 414, 544, 573, 579, 585, 587: Ice nucleation efficiency/ability/activity/capability was not measured in this study, but concentrations of INP. Ice nucleation efficiency requires knowing the concentration of the ice active species. To be more consistent replace them throughout the paper with nINP or AF where appropriate.

Line 362: the difference would be much larger if fine particles would not be limited to particles larger than 0.5um.

Line 363ff: instead of listing AF, point to Fig.3. Disentangle if the AF is governed by changes in particle concentration or nINP.

Line 373: mention what particle size-ranges Si et al. used.

Sec.3.4: This section needs to be structured better. It is currently unclear what the important results are.

Line 387, 391ff: factors of 1.5 or 1.6 are very small differences to draw conclusions. Additionally, the number of samples could be too small to determine the underlying nINP distributions on which the comparisons in this section are based on.

Line 395: State the number of June samples and explain how it was determined that they represent a significant peak.

Line 409ff: The effect of particle concentration and nINP need to be disentangled. The absence of a change in nINP with season points to the particle concentration causing the change in the AF.

Line 430: the 50-120nm size range mentioned here provides additional evidence against the chosen lower size limit of 500nm for this study.

Line 462, Fig.6: The trajectories with land contact are not visible in Fig.6. Removing the not used grey trajectories could help.

Line 468f: specify that the -15°C, PM1 dataset was used. It is unclear why the coarse and fine data is an advantage for DFPC. There is no comparison between coarse and fine nINP shown here, except Figs. S7 and S8. Do these maps confirm Mansour et al. and McCluskey et al.?

Line 472f: Has the analysis been tried for -18°C and -22°C? It is mentioned on line 494ff that experiments showed INP active at -22°C are generated.

Line 483: Provide an interpretation of significant, negative correlated areas. Are they caused by elevated CHL without the expected nINP response?

Line 487: see comment on Fig.7.

Line 534ff: There could be an increase in biological, high T INP from growing biota on CFPC filter during storage at room T.

Line 538: It could be added that similar air volumes were collected on the filter for CFPC and WT-CRAFT.

Line 549: It could be expected that soluble compounds suppress ice nucleation. Why would the opposite be observed for these samples?

Line 552: A detailed comparison is not needed. To claim sensitivity of arctic nINP on the ice nucleation mode, exemplary DFPC filter need to be analysed with the WT-CRAFT method and vice versa. Otherwise it is speculative.

Line 566: It seems plausible that due to small number of samples, the INP population was not well characterized in aforementioned studies, leading to misinterpretation.

Line 578: state which results

Line 591: where is shown that land sources have this potential? PM10 nINP do not change much even when more land and open sea are along trajectories and PM1 nINP decrease.

Line 591, 592: at what T have mineral dusts a higher activity and at what T are marine INP less active?

Line 598: specify the results, e.g. "location of hot spots for marine INPs"

Line 610f: The presented analysis does not support this conclusion. A more plausible reason could be differences in the DFPC and WT-CRAFT samples. As mentioned above, to arrive at this conclusion the same samples must be analysed with both methods. Replace "undeniable" by "potential".

Line 616: The dataset is not unique, other studies have collected Arctic INP covering different seasons e.g. Manson et al., Schrod et al.

Line 624: how did the back-trajectories show this separation? They were done for summer only.

Line 629: repetition of line 625

Tab.1: Consider making 3 columns showing the nINP at -15°C, -18°C, -22°C, relevant for this study, instead of the current last two columns. A minus sign is missing in Bigg, 1996 T range.

Tab.2: Subtracting 1-3 standard deviations from the average values gives negative concentrations. nINP can not be negative, the real variation is clearly asymmetric. Consider reporting the range and average instead and refer to the table instead of listing these values in the text.

Tab.3, 4: Scatterplots for all 39 significant correlations reported in these tables could be shown in the supplement and investigated for outliers to exclude false positives.

Fig.3: mention the particle size range used to calculate AF in a), b) c)

Fig.4: indicate what is shown in a), b), c) in the caption. Specify that uncertainties can be found in sec.2.2.1 (CFPC) and 2.2.2 (WT-CRAFT)

Fig.6: consider removing grey trajectories.

Fig.7: specify which region belongs to which time lag. Consider overlaying the trajectories as thin lines to indicate upwind locations and clarify the choice of regions.

Fig. 8 b): consider showing the trajectories corresponding to the 14 INP samples as thin lines

Tab. S1: Column headers "Total Flow (optimized for 50% of filter)" and "Suspension water volume (First frozen drop=0.001 INP L-1)" are unclear. Give a description in the legend and rename the columns, e.g. "air volume" and "suspension water volume". Check the calculation of the suspension volumes.

Fig.S3: consider using the same y-axis scale for subplots showing the same temperature. Even if there are only 3 spring datapoints in Wex et al. at this T, add -22°C for DFPC and WT-CRAFT as the seasonal change is discussed in the main text. Give a description of the subplots a)-f) in the figure legend.

Fig. S4: Describe y-axis in the figure legend. Is PM1 or PM10 shown for DFPC? Mark the increase in AF due to the decrease in particle concentration from spring to summer (= particle conc. summer/ particle conc. spring) as horizontal line. Spring to summer AF increase close to this line indicate no change in nINP.

**Technical corrections**

Provide more links/references between sections and to tables and figures to navigate the paper.

Line 27:  remove "trustful"

Line 79: "suggested *them* to be…"

Line 200: remove "now"

Line 210: replace "side-by-side" with "in parallel".

Line 413: replace "point sin" with "points in"

Tab. 4b: remove "I" before nINP in the caption.

Fig. S1: change (a) -18°C to (b) -18°C in the second figure.

---

## Author Response (AR2)

We would like to express sincere gratitude for the referee's helpful comments. Below, we provide our point-by-point responses (in blue colour).

**2nd Referee report on "Ice-nucleating particle concentration measurements from Ny-Ålesund during the Arctic Spring-Summer in 2018" by Rinaldi et al.**

The authors have addressed most of the previous comments and improved the manuscript. However, I would like to suggest some further revisions to improve the manuscript before it can continue the review process. We appreciate these general remarks, reconsidered referee comments and revised the manuscript accordingly. Below, we provide our point-by-point responses.

Main comment
The number of samples this study is based on is small and might not allow to characterize the population of INP. It needs to be highlighted in the abstract and conclusion that results are preliminary because of the small sample size. Findings in the abstract and conclusion should be limited to the strong signals that are expected to be reproduceable in future investigations. For less clear results it should be stressed that more observations are needed. The limitations of the applied analysis due to the size of the available dataset should be stated clearly in each section. This includes stating the number of measurements used and the assumption on the structure of the data, e.g. normal distribution of nINP (not typical) to compare spring to summer concentrations or contribution of coarse and fine particle fraction to nINP.

The number of samples considered for each study is now offered in revised Tables and text. Our datasets are made based on a compilation of 33 DFPC samples (16 in spring and 17 in summer) and 28 WT-CRAFT samples. These numbers are not by far the smallest published in Arctic INP studies as can be seen in the Table below. Thus, we respectfully believe that our results are not "preliminary".

In the text, we have highlighted all the results that may be considered not fully robust because of the low number of samples. For instance, the results of the spatio-temporal correlation analysis with CHL and the output of the CWT algorithm (even though examples of applications of CWT with similar dataset dimension exist in literature (Hsu et al.; 2003)). We have also re-scaled our considerations of the different responses of Arctic aerosol particles to the ice nucleation mode, not because of the number of samples, but because more detailed intercomparisons would be necessary to address this point. As for the rest, we believe that our data are worth to be presented without any caveat on the sample dimension.

**Table.** Number of samples analysed for previous Arctic INP publications

| Reference | Location | Number of samples |
|---|---|---|
| Bigg (1996) | High Arctic (cruise) | >50 |
| Bigg and Leck (2001) | High Arctic (cruise) | >50 |
| Conen et al. (2016) | Haldde observatory, Norway | 4 |
| Mason et al. (2016) | Alert, Canada | 9 |
| Creamean et al. (2018) | Oliktok Point, Alaska | 17 |
| Si et al. (2018) | Lancaster Sound, Canada | 1 |
| Irish et al. (2019) | Multiple (cruise) | 28 |
| Santl-Temkiv et al. (2019) | Villum, Greenland | 35 |
| Si et al. (2019) | Alert, Canada | 16 |
| Tobo et al. (2019) | Mt. Zeppelin, Svalbard | 13 |
| Wex et al. (2019) | Alert, Canada | ~40 |
|  | Utqiagvik, Alaska | ~40 |
|  | Ny-Ålesund, Svalbard | 13 |
|  | Villum, Greenland | 11 |
| Welti et al. (2020) | High Arctic (cruise) | >50 |
| Schrod et al. (2020) | Mt. Zeppelin, Svalbard | >50 |
| This study DFPC | Ny-Ålesund, Svalbard | 33 |
| This study WT-CRAFT | Ny-Ålesund, Svalbard | 28 |

Hsu, Y. K., Holsen, T. M., and Hopke, P. K.: Comparison of hybrid receptor models to locate PCB sources in Chicago, Atmospheric Environment, 37, 545-562, 10.1016/s1352-2310(02)00886-5, 2003.

Because correlation analysis is prominent throughout the paper, I suggest to add a subsection detailing the statistical analysis, including how significance in dependence of number of samples is determined (t-test), to the Method Section. This would help to understand, e.g. why correlation coefficients are high and why 0.5 is significantly non-zero for DFPC but 0.8 can be not significant for WT-CRAFT (Tabs.3 and 4).
The required sub-Section has been added. Please see the revised **Sect. 2.4. Statistical data treatment**.

Specific comments
Line 27f, 101f, 609ff.: The difference in nINP due to two different ice nucleation modes does not emerge from the data. To make this conclusion, filter samples from one sampling setup should be analysed with both DFPC and WT-CRAFT. More plausible are differences in the samples used by the two methods.
We agree with the reviewer on the importance of intercomparing the two measurements. Unfortunately, it was not possible to operate the suggested comparison using our 2018 samples for the following reasons:
(1) we cannot use an identical substrate for the two INP analysis techniques. We have assessed the applicability of the cellulose membrane (optimal for DFPC) in WT-CRAFT, and we found notable background artifacts from the blank substrate below -22°C. Thus, the same filter sample cannot be shared for both INP measurement techniques.
(2) Our aerosol samplers employed different sampling periods for optimized flow conditions and estimated detection limits. The aerosol sampler for DFPC employed high flows, which were needed for the desired cut-size (see Sect. 2.1). If we conducted sampling with a longer period than what we employed, DFPC would likely suffer from filter overloading issues. Likewise, reducing the sampling time for WT-CRAFT would have created detection limit issues. As described in Sect. 2.1, the filter sampling flow for WT-CRAFT was ~5 LPM.
Nevertheless, we fully understand the concerns of the reviewers about relying on ice nucleation modes for explaining the observed difference between $n$INP$_{DFPC}$ and $n$INP$_{WT-CRAFT}$. As we clarified in the revised manuscript, it is not conclusive, and we hope to be able to further investigate this issue in future follow-up

studies. To allay further misgivings, we softened our tones and revised the text to present the particle sensitivity to different ice nucleation modes only as a potential reason. In the revised manuscript, we also discussed other potential factors for the observed discrepancy (please refer to the revised Sect. 4.1). Overall, the addressed potential factors include (1) differences in measurement uncertainties, (2) sampling apparatus, (3) sample storage protocols, (4) substrate types, (5) sampling durations and (6) ice nucleation paths (condensation vs immersion freezing).

Nonetheless, we would like to clarify here that explaining the gap is not a major point of the study. Our work extends the INP observations at GVB, contributing to filling the present lack of observations in the Arctic. Furthermore, it provides information on the AF, which was never calculated before at GVB and which was only rarely addressed at Arctic sites. One of the major findings of our work is that the seasonality of $n$INP can be significantly different from what was observed in previous studies. This finding is supported by both the INP datasets, notwithstanding the concentration gap and all the sampling differences. Our results evidence potentially a great interannual variability and the necessity for further data coverage to better understand INP dynamics over the Arctic. Furthermore, we present and discuss for the first time the seasonality of the AF, intended as a proxy of the overall ice nucleation ability of the particle population at the study site. Finally, our study reports information on the ice behaviour of fine vs coarse aerosol particles, which was addressed only in a few more Arctic studies so far. This was clarified at the end of the Introduction Section.

David, R. O., Marcolli, C., Fahrni, J., Qiu, Y., Perez-Sirkin, Y. A., Molinero, V., Mahrt, F., Brühwiler, D., Lohmann, U., and Kanji, Z. A.: Pore condensation and freezing is responsible for ice formation below water saturation for porous particles, Proceedings of the National Academy of Sciences, 116, 8184–8189, https://doi.org/10.1073/pnas.1813647116, 2019.

Wagner, R., Kiselev, A., Möhler, O., Saathoff, H., and Steinke, I.: Pre-activation of ice-nucleating particles by the pore condensation and freezing mechanism, Atmos. Chem. Phys., 16, 2025–2042, https://doi.org/10.5194/acp-16-2025-2016, 2016.

Line 29: name the "several important indications"
This sentence was reformulated for major clarity. Please refer to the revised Abstract.

Line 29f: This is inconsistent with what is reported on line 419ff
In line 29, we say that scaling $n$INP to the total particle number concentration, i.e., deriving the AF, it is possible to extract more information from the dataset. This is a true statement both in general and regarding this particular case. Further on in the abstract, we explain that the AF shows a clearer seasonal trend than $n$INP. Line 419 and following lines, treats the correlation between $n$INP and particle number concentration (which is qualitatively observable, but not statistically significant). These topics are mostly unrelated and we do not see how these parts can be contradicting each other.

Line 31: specify "subset of our data"
We specified the following: "(WT-CRAFT, between -18 and -21°C)". We have conducted the temperature-resolved correlation analysis, and this particular range of temperatures was the one showing a significant difference.

Line 33: Explain how higher AF can be interpreted as larger freezing efficiency of large particles.
This sentence was completely reformulated: "This seasonal AF trend corresponds to the overall decrease in aerosol concentration towards summer and a concomitant increase in the contribution of super-micrometre particles. Indeed, the AF of coarse particles resulted markedly higher than that of sub-micrometre ones (2 orders of magnitude)".

Line 34: inconsistent to line 417 where it is stated that no clear relation emerges between nINP and meteorology.

We do not think this is inconsistent: we intended to point out that no meteorological parameter is a major driver of the $n$INP time trend. This supports our choice of using the $n$INP vs CHL correlation as a tool for identifying potential marine sources of INPs. In this respect, the meteorological analysis is an important step of the process that leads to the conclusion that: "the summertime INP population is influenced both by terrestrial (snow-free land) and marine sources".

Line 48: what other mechanisms beside Bergeron-Findeisen play an important part?

The transformation of water between vapour, liquid and ice phase in mixed-phase clouds can occur in different ways. The Wegener-Bergeron-Findeisen (WBF) mechanism occurs only if $e_w > e > e_i$, where $e$ is the in-cloud vapour pressure, $e_w$ the vapour pressure of supercooled liquid droplets, and $e_i$ is the ice vapour pressure. Nevertheless, additional situations can be present in mixed-phase clouds. For instance, if $e > e_w > e_i$, both droplets and ice particles grow simultaneously and compete for water vapour. Korolev and Mazin (2003) showed that this condition may occur in ascending mixed-phase clouds when updraft velocity exceeds a fixed value. Fan et al. (2011) found that in the Arctic, in both single and multilayer mixed-phase clouds, the WBF process occurs in about 50% of the mix-phase regime, prevalently in the downdraft region, and in the other half of the cloud both liquid and ice grow simultaneously.

For these reasons, we believe it is correct to refer to the WBF as one of the mechanisms responsible for the relationship between lifetime and ice crystal concentration in mixed-phase clouds.

Fan et al., 2011. Representation of Arctic mixed-phase clouds and the Wegener-Bergeron-Findeisen process in climate models:    Perspectives from a cloud-resolving study J. Geophys. Res. 116, D00T07, doi:10.1029/2010JD015375.

Korolev, A. V., I. P. Mazin (2003), Supersaturation of water vapor in clouds, J. Atmos. Sci., 60(24), 2957–2974, doi:10.1175/1520-0469.

Line 56, 57: Clarify the difference between water vapour deposition and water vapour condensation.

We have reformulated the description of the deposition mode ice nucleation, making it more adherent to that by Vali et al. (2015).

"Ice formation by deposition occurs when the ambient is supersaturated with respect to ice in water-subsaturated conditions so that ice forms on an INP without prior formation of liquid".

Line 60f.: If condensation and immersion are the same process how can the difference in nINP between DFPC and WT-CRAFT be explained by it? To make this case, provide evidence that the two mechanisms can be different.

Here, we only intended to introduce what has been studied for condensation vs. immersion by referring to the two cited studies. We did not mean to conclude that condensation = immersion in the introduction section. Further assessments from the laboratory and field settings are needed to understand the similarity of ice nucleation modes and processes. Whether condensation and immersion are the same processes or not highly depends on the aerosol properties. We discuss this point in the revised Sect. 4.1.

We also rephrased the sentence in the introduction section to: "However, the recent inter-comparison study with two different organic fiber samples shows a difference between condensation freezing and immersion freezing measurements (i.e., ice nucleation efficiency of the former is higher than the latter (Hiranuma et al., 2019)). Further laboratory and field assessments are therefore necessary to understand the similarity of ice nucleation modes and processes".

Line 96: Why are Santl-Temkiv et al. and Wex et al. cited for the interpretation of data in Tobo et al. and not the Tobo et al. paper itself?

This sentence was not related to Tobo et al. (2019) specifically. We refer here to the enhancement of *n*INP reported by all the cited papers (Wex et al., 2019; Santl-Temkiv et al., 2019 and Tobo et al. 2019). In this sense, the citation is not wrong as:

- Wex et al. (2019) state: "In summer, the higher bioaerosol concentrations compared to spring indicate contributions from local and regional terrestrial and marine ice-free areas"; "these INPs can originate from both terrestrial and marine sources in the Arctic. These sources are strong in summer and weak or absent in winter, depending on the conditions on the ground".
- Santl-Temkiv et al. (2019) state that: "In summer, the higher bioaerosol concentrations compared to spring indicate contributions from local and regional terrestrial and marine ice-free areas"; "Based on this observation, we suggest that the high concentrations of INPs in the air during summer may be partially related to strengthened local terrestrial sources, in particular soil dust"
- Tobo et al. (2019) state: "This suggests that significant local sources of INPs other than marine organic materials might exist in and/or around the Svalbard region in the summer"

After all, these three papers attribute the summertime enhancement of *n*INP to local sources. We have added the reference to Tobo et al. (2019) according to the reviewer comment.

line 104f: Explain why INP measurements are the key to verify that immersion freezing is the most relevant ice nucleation mechanism in Arctic mixed-phase clouds and how such measurements can be used to do that.

We admit that this paragraph lacked logical consistency. This sentence has been removed. We have revised the entire paragraph to:

"Our study aims to add to the still scant INP observations in the Arctic environment, investigating *n*INP and potential INP sources during spring and summertime at the ground-level site of GVB. In particular, we extend the INP observations at GVB, previously only 13 samples (Wex et al., 2019), presenting the results of 61 samples investigated with two offline INP measurement techniques. We also analyze the ice nucleation efficiency of Arctic aerosol particles by calculating their activated fraction (AF). To date, only a limited number of studies provide information on INP trends scaled to the total aerosol concentration over the Arctic (Si et al., 2018). AF estimation can be understood as a simple metric indicating the ice-nucleating efficiency of particles within a specific aerosol sample (Schrod et al., 2020). In our specific case, it provides further insight, over and above the *n*INP data, into INP characteristics over the Atlantic sector of the Arctic".

Line 117ff.: state the number of samples collected in spring, summer, PM1, PM10.

The required information was added in Sect. 2.1.

Line 128ff.: on line 131 it says the 5.4lpm are a subset of the airflow through the TSP inlet. Specify the total flow through the inlet and describe more clearly how the 5.4lpm are extracted from the higher total flow. Was there a pressure drop in the flow from which the subset flow was taken? Was this considered to determine the sample flow? Was there an online measurement of the flow through the membrane filter (how are flow rates in Tab.S1 measured) and why did the flow vary from 4.8-5.6lpm between samples?

The total flow of the central sampling stack inlet was reported in the manuscript (150 LPM). Further, as described in Sect. 2.1, an 8 mm OD stainless steel tube was used as a pickup inlet for a filter sampler. All excess amount of air was drawn through other instruments or a central pump. Below we provide our sampling system schematic as well as the sampling flow measured at the beginning and end of each sampling activity in the figure and table, respectively. A TSI4100 flow meter is placed in the sampling line to check and make sure we have consistent flow on the filter cross-section. We acknowledge onsite operators who typically check our flow daily (even several times per day) to make sure the airflow is consistent. In 2018, our flow measured at the beginning and end of each sampling activity deviated on average only <2% and never

exceeded 5%. Thus, we used an average flow rate measured during individual sampling activity as a representative flow for each sample. While the pressure drop was not measured for each sampling activity, we have measured a pressure drop in the flow of our sampling system to be approximately 15 mb and 200 mb with 1.5 LPM and 9.0 LPM flow. Because we observe such a consistent airflow throughout individual samplings and no physical damages on filters after sampling, we consider our sampling is successful and valid. The observed variation in sampling airflow between samples is presumably due to air valve control. Nevertheless, we estimate a total flow through each filter by accounting for this, and we echo that the airflow was consistent for each sampling activity.

[Figure]

**Figure.** A schematic of aerosol particle sampling system at GVB. Aerosol particles were collected on a 47 mm Nuclepore filter for offline INP analysis by WT-CRAFT.

**Table.** Sampling airflow measured at the beginning and end of individual sampling activities plus the averaged flows.

| ID | Start_Date | Start_Time | Start_Air_Flow_LPM | End_Date | End_Time | End_Air_Flow_LPM | Average_Flow_LPM |
|----|-----------|-----------|--------------------|----------|----------|------------------|------------------|
| 1 | 2018-04-16 | 17:00:00 | 5.10 | 2018-04-20 | 10:00:00 | 5.10 | 5.10 |
| 2 | 2018-04-20 | 14:40:00 | 5.10 | 2018-04-24 | 14:40:00 | 5.04 | 5.07 |
| 3 | 2018-04-24 | 18:20:00 | 5.40 | 2018-04-28 | 16:00:00 | 5.50 | 5.45 |
| 4 | 2018-04-29 | 13:30:00 | 5.60 | 2018-05-02 | 16:15:00 | 5.50 | 5.55 |
| 5 | 2018-05-02 | 16:20:00 | 5.60 | 2018-05-06 | 14:37:00 | 5.34 | 5.47 |
| 6 | 2018-05-06 | 14:45:00 | 5.34 | 2018-05-10 | 13:00:00 | 5.50 | 5.42 |
| 7 | 2018-05-10 | 13:10:00 | 5.70 | 2018-05-14 | 11:05:00 | 5.58 | 5.64 |
| 8 | 2018-05-14 | 11:15:00 | 5.58 | 2018-05-18 | 07:50:00 | 5.35 | 5.47 |
| 9 | 2018-05-18 | 08:00:00 | 5.50 | 2018-05-22 | 08:28:00 | 5.50 | 5.50 |
| 10 | 2018-05-22 | 08:30:00 | 4.80 | 2018-05-26 | 11:33:00 | 4.80 | 4.80 |
| 21 | 2018-05-26 | 11:45:00 | 5.57 | 2018-06-03 | 18:30:00 | 5.46 | 5.52 |
| 22 | 2018-06-03 | 18:35:00 | 5.36 | 2018-06-07 | 17:20:00 | 5.60 | 5.48 |
| 23 | 2018-06-07 | 17:24:00 | 5.40 | 2018-06-11 | 17:35:00 | 5.46 | 5.43 |
| 24 | 2018-06-11 | 17:40:00 | 5.41 | 2018-06-15 | 16:24:00 | 5.30 | 5.36 |
| 25 | 2018-06-15 | 16:28:00 | 5.39 | 2018-06-19 | 19:05:00 | 5.35 | 5.37 |
| 26 | 2018-06-19 | 19:09:00 | 5.26 | 2018-06-23 | 19:16:00 | 5.25 | 5.26 |
| 27 | 2018-06-23 | 19:20:00 | 5.36 | 2018-06-27 | 13:55:00 | 5.33 | 5.35 |
| 28 | 2018-06-27 | 14:00:00 | 5.39 | 2018-07-01 | 16:40:00 | 5.43 | 5.41 |
| 16 | 2018-07-01 | 16:50:00 | 5.43 | 2018-07-05 | 17:20:00 | 5.48 | 5.46 |
| 17 | 2018-07-05 | 17:25:00 | 5.44 | 2018-07-09 | 17:22:00 | 5.39 | 5.42 |
| 18 | 2018-07-09 | 17:27:00 | 5.39 | 2018-07-13 | 18:24:00 | 5.38 | 5.39 |
| 19 | 2018-07-13 | 18:33:00 | 5.34 | 2018-07-17 | 16:43:00 | 5.33 | 5.34 |
| 20 | 2018-07-17 | 16:52:00 | 5.39 | 2018-07-21 | 15:55:00 | 5.30 | 5.35 |
| 11 | 2018-07-21 | 16:02:00 | 5.43 | 2018-07-25 | 16:31:00 | 5.36 | 5.40 |
| 12 | 2018-07-25 | 16:38:00 | 5.40 | 2018-07-29 | 15:07:00 | 5.46 | 5.43 |
| 13 | 2018-07-29 | 15:14:00 | 5.49 | 2018-08-02 | 18:39:00 | 5.41 | 5.45 |
| 14 | 2018-08-07 | 15:55:00 | 5.46 | 2018-08-11 | 14:05:00 | 5.43 | 5.45 |
| 15 | 2018-08-11 | 14:12:00 | 5.42 | 2018-08-15 | 17:36:00 | 5.43 | 5.43 |

Line 165: what is meant by "reasonably matches"? The 95% confidence interval is inherent in the cited formula.

Thanks for catching this. We now clarify the previously reported uncertainty estimation method of WT-CRAFT technique as:

"The uncertainty of ice nucleation efficiency estimation in WT-CRAFT was previously estimated based on the average standard deviation at $T$s > -25°C for a known composition (microcrystalline cellulose). Alternatively, as demonstrated in Vepuri et al. (2021) and Schiebel (2017), our experimental uncertainty in estimated $n$INP can be evaluated using the 95 % confidence interval method".

Line 178: point to sec. 2.2.3. for details on how nINP are estimated
Done.

Line 179: mention already here that half of each filter was used and explain how the used water volume was calculated. Volumes given in Tab.S1, row 2-6 are off by -0.1 to 0.2ml from calculated values.
To incorporate the reviewer's suggestion, we rephrased L179 to: "Prior to each WT-CRAFT experiment, we suspended particles on an individual filter sample in a known volume of ultrapure High Performance Liquid Chromatography (HPLC) grade water, in which the first frozen droplet corresponded to ≈ 1 INP m$^{-3}$ (in the range of 0.93 – 1.02 m$^{-3}$; Table S1). The HPLC water volume was determined according to Eqns. 1-2 in Sect. 2.2.3. Half of each filter was used for each WT-CRAFT experiment, the other half was saved for other and future uses"
We also decided to provide more decimal points for our data provided in the last few columns of Table S1. Note that our mL pipette used in this study has 2 decimals. Our reported $n$INP values from WT-CRAFT scale to this initial number (ca. 1 INP m$^{-3}$), so our $n$INP data at across the measured temperatures are valid. While the initial $n$INP determines our INP detection sensitivity and low detection limit, the variation between 0.93 m$^{-3}$ and 1.02 m$^{-3}$ does not substantially impact our other experimental parameters/capabilities.

**Table S1. Summary of sampling conditions for filters collected for WT-CRAFT.**

| Sample ID | Filter Sampling Ref Start Time | Filter Sampling Ref End Time | Flow Rate | Air Volume | Suspension water volume | $n$INP corresponding to the first frozen droplet |
|---|---|---|---|---|---|---|
| | DAT_UTC | DAT_UTC | LPM | L | mL | m$^{-3}$ |
| NYA_GVB_01 | 4/16/2018 17:00 | 4/20/2018 10:00 | 5.100 | 13617.000 | 2.84 | 1.00 |
| NYA_GVB_02 | 4/20/2018 14:40 | 4/24/2018 14:40 | 5.070 | 14601.600 | 3.08 | 1.01 |
| NYA_GVB_03 | 4/24/2018 18:20 | 4/28/2018 16:00 | 5.450 | 15314.500 | 3.10 | 0.97 |
| NYA_GVB_04 | 4/29/2018 13:30 | 5/2/2018 16:15 | 5.550 | 12445.875 | 2.42 | 0.93 |
| NYA_GVB_05 | 5/2/2018 16:20 | 5/6/2018 14:37 | 5.470 | 15471.895 | 3.26 | 1.01 |
| NYA_GVB_06 | 5/6/2018 14:45 | 5/10/2018 13:00 | 5.420 | 15325.050 | 3.26 | 1.02 |
| NYA_GVB_07 | 5/10/2018 13:10 | 5/14/2018 11:05 | 5.640 | 15890.700 | 3.29 | 0.99 |
| NYA_GVB_08 | 5/14/2018 11:15 | 5/18/2018 7:50 | 5.465 | 15179.037 | 3.10 | 0.98 |
| NYA_GVB_09 | 5/18/2018 8:00 | 5/22/2018 8:28 | 5.500 | 15917.000 | 3.32 | 1.00 |
| NYA_GVB_10 | 5/22/2018 8:30 | 5/26/2018 11:33 | 4.800 | 14263.200 | 2.98 | 1.00 |
| NYA_GVB_21 | 5/26/2018 11:45 | 6/3/2018 18:30 | 5.515 | 32883.188 | 6.86 | 1.00 |
| NYA_GVB_22 | 6/3/2018 18:35 | 6/7/2018 17:20 | 5.480 | 15576.900 | 3.25 | 1.00 |
| NYA_GVB_23 | 6/7/2018 17:24 | 6/11/2018 17:35 | 5.430 | 15668.265 | 3.27 | 1.00 |
| NYA_GVB_24 | 6/11/2018 17:40 | 6/15/2018 16:24 | 5.355 | 15218.910 | 3.18 | 1.00 |
| NYA_GVB_25 | 6/15/2018 16:28 | 6/19/2018 19:05 | 5.370 | 15887.145 | 3.32 | 1.00 |
| NYA_GVB_26 | 6/19/2018 19:09 | 6/23/2018 19:16 | 5.255 | 15152.792 | 3.16 | 1.00 |
| NYA_GVB_27 | 6/23/2018 19:20 | 6/27/2018 13:55 | 5.345 | 14525.037 | 3.03 | 1.00 |
| NYA_GVB_28 | 6/27/2018 14:00 | 7/1/2018 16:40 | 5.410 | 16013.600 | 3.34 | 1.00 |
| NYA_GVB_16 | 7/1/2018 16:50 | 7/5/2018 17:20 | 5.455 | 15792.225 | 3.30 | 1.00 |
| NYA_GVB_17 | 7/5/2018 17:25 | 7/9/2018 17:22 | 5.415 | 15587.078 | 3.25 | 1.00 |
| NYA_GVB_18 | 7/9/2018 17:27 | 7/13/2018 18:24 | 5.385 | 15662.273 | 3.27 | 1.00 |
| NYA_GVB_19 | 7/13/2018 18:33 | 7/17/2018 16:43 | 5.335 | 15071.375 | 3.15 | 1.00 |
| NYA_GVB_20 | 7/17/2018 16:52 | 7/21/2018 15:55 | 5.345 | 15241.268 | 3.18 | 1.00 |
| NYA_GVB_11 | 7/21/2018 16:02 | 7/25/2018 16:31 | 5.395 | 15602.340 | 3.26 | 1.00 |
| NYA_GVB_12 | 7/25/2018 16:38 | 7/29/2018 15:07 | 5.430 | 15391.334 | 3.21 | 1.00 |
| NYA_GVB_13 | 7/29/2018 15:14 | 8/2/2018 18:39 | 5.450 | 16254.626 | 3.39 | 1.00 |
| NYA_GVB_14 | 8/7/2018 15:55 | 8/11/2018 14:05 | 5.445 | 15382.126 | 3.21 | 1.00 |

| NYA_GVB_15 | 8/11/2018 14:12 | 8/15/2018 17:36 | 5.425 | 16177.350 | 3.38 | 1.00 |

Line 202ff: I couldn't find the size-range in Kanji et al., 2017. Pruppacher & Klett, 2010 section 9.2.3.2 suggest 0.1um as a lower size limit, which I would recommend. The choice needs to be motivated better. How does the lower size limit effect the results in sec. 3.3?

This is a valid question. Our choice (i.e., comparing $n$INP with particles larger than 500 nm) is arbitrary and motivated by different considerations. First, we followed the approach by DeMott et al. (2010), which report: "The choice of 0.5-μm diameter as the lower limit for summing number concentrations of "large" particles is a relatively arbitrary one of convenience, selected to limit the influence on derived relationships of high concentrations of non-IN particles in the range 0.1–0.5 μm, while retaining sufficient number concentrations of particles to reference to IN concentrations". Furthermore, we already adopted this choice in previous DFPC papers (Rinaldi et al., 2017; Rinaldi et al., 2019), so we aimed at maintaining the possibility of comparing with previous DFPC results.

We agree with the reviewer that there is no consensus in the literature on which aerosol particle size range would be more appropriate for the AF estimation. For instance, Kanji et al. (2017) report: "As the sites appear with a finite probability (Niedermeier et al. 2015), smaller particles (e.g., ,500 nm) are less likely to act as INPs". Nevertheless, they also add: "However, in Mertes et al. (2007) 200-nm diameter particles were inferred to make up the majority of all INPs, based on the mode size in atmospheric ice residual number distributions". For previous Arctic studies, Si et al. (2018) shows the presence of INPs also in the 200-300 and 400-500 nm size classes. Therefore, a lower particle size limit could be appropriate, and the reviewer's comment (How does the lower size limit impact the results in sec. 3.3?) is also plausible.

Using the AF data calculated using particle number concentration from 100 nm (AF$_{100}$ hereafter) instead of AF$_{500}$ did not change our findings and conclusion. We confirm our result regarding the higher ice nucleation efficiency of coarse atmospheric particles, enhancing the difference between AF$_{fine}$ and AF$_{coarse}$ (almost 2 orders of magnitude difference), remains valid. The spring to summer increase of the AF also remains valid. We have updated the manuscript by introducing the AF$_{100}$ values.

Line 215ff: Clarify what size-range was used to calculate AF. From line 203 it would seem that only APS data (0.5-10um) was used.

We now use AF$_{100}$, and the associated text has been revised accordingly to:

"The number concentration in the resulting overlapping range was taken from the SMPS data. Finally, commutative aerosol particle counts of SMPS and APS were considered as a total aerosol particle number concertation. In order to compare with $n$INP and to calculate the AF, the particle number concentrations at 10 minutes time resolution were averaged over each filter sampling period. AF was calculated using the size range 0.1 – 10 μm for DFPC$_{PM10}$ and WT-CRAFT data, 0.1 – 1 μm for DFPC$_{PM1}$ data and 1 – 10μm for DFPC data in the super-micrometre regime.".

Line 246: provide a reference for the marine boundary layer height in the Arctic.

We provided the requested reference: DAI Cheng-Ying, GAO Zhi-Qiu, WANG Qing, and CHENG Gang, Analysis of Atmospheric Boundary Layer Height Characteristics over the Arctic Ocean Using the Aircraft and GPS Soundings, ATMOSPHERIC AND OCEANIC SCIENCE LETTERS, 2011, VOL. 4, NO. 2, 124-130.

Line 270: It could be already mentioned here that nINP at -15°C, PM1 are used for this exercise.

We believe that adding this information without explaining why would be confusing. Our reasons are explained later on in the text.

Line 280: to be consistent throughout the paper, consider using nINP instead of Ct for the INP concentration.

Done.

Line 295: It could be explained here how the CHL correlation analysis and CWT were overlayed (Fig.8b).
We believe the text is clearer as it is.

Line 303: Can you provide an interpretation of the difference in slope? It could suggest that there is a specific type of high temperature INP, that was only detected in the DFPC measurement. Could storing the filters at different conditions cause such an effect?
The difference in sample storing methods cannot be a sole factor to explain the discrepancy. As we kept the samples for DFPC at the room air $T$ (and the WT-CRAFT samples at 4 °C except during transportation), the suppression of INPs is expected to be more obvious for DFPC than WT-CRAFT, which is not supported by the observed trend of $n$INP$_{DFPC}$ > $n$INP$_{WT-CRAFT}$.
We are aware that Beall et al. (2020) recently demonstrated that storage of precipitation samples can lead to losses of INPs (especially the heat-labile ones). Storage at room temperature and at 4°C resulted basically in INP losses, with the former conditions showing somewhat higher losses than the latter. Nevertheless, the results by Beall et al. (2020) are not necessarily valid for particle samples collected on filters. Furthermore, there is not yet a consensus in the literature (and within the scientific community) about the effect of sample storage on INP measurements, so answering this question quantitatively is very hard at present. For instance, Wex et al. (2019) stated that "it is not yet known with certainty how the temperature during storage will affect INP concentrations". Bigg (1990) reported no significant deterioration in INP concentration after 5 years, for filters sealed in a dry-sealed container. Conen et al. (2015) commented that storage of filters at -20°C may not have been necessary, because re-analysis of filters analyzed previously showed no effect of storage. Wex et al. (2015) and Polen et al. (2016) showed that storage at temperatures above 0°C or even under freezing conditions has been found to reduce the ice activity of biogenic INPs. Stopelli et al. (2014) studied INP concentrations in a snow sample stored at 4°C and observed a decrease in the concentration of INPs active at -10°C over 30 days (factor of 2).
Regarding the effect of sample storage on the $\Delta n$INP/$\Delta$T slope, we evidence that DFPC samples were not refrigerated during storage, while WT-CRAFT ones were kept at 4°C. Therefore, based on the limited literature on this topic, degradation of INPs during storage would have caused higher losses in DFPC samples, resulting in the opposite effect on the $\Delta n$INP/$\Delta$T slope. For this reason, we believe that losses during storage cannot be considered a valid explanation for the different slopes, at least based on the current knowledge of the topic. We now discuss this point in the revised Sect. 4.1.

Bigg, E.K., 1990. Measurement of concentrations of natural ice nuclei. Atmos. Res., 25, 397-408.
Conen et al., 2015. Atmospheric ice nuclei at the high-altitude observatory Jungfraujoch, Switzerland. Tellus B, 67, 1-10.
Conen et al., 2015. Atmospheric ice nuclei at the high-altitude observatory Jungfraujoch, Switzerland. Tellus B, 67, 1-10.
Polen, M., Lawlis, E., and Sullivan, R. C.: The unstable ice nucleation properties of Snomax (R) bacterial particles, J. Geophys. Res.-Atmos., 121, 11666–11678, https://doi.org/10.1002/2016jd025251, 2016.
Stopelli, E., Conen, F., Zimmermann, L., Alewell, C., and Morris, C. E.: Freezing nucleation apparatus puts new slant on study of biological ice nucleators in precipitation, Atmos. Meas. Tech., 7, 129–134, https://doi.org/10.5194/amt-7-129-2014, 2014.

Line 304ff: Consider referring to Tab.2 instead of listing the concentrations.
We would like to report this key information in the text as well.

Line 307: give number of samples instead of "<50%".

The requested information was added in the revised manuscript: "(1 sample, at T of -9°C, and 13, at *T* of -13.5°C, over 28 total samples)".

Line 311: give T-range for the nINP, see comment on Tab.1
The T range was specified as follows:
"The range of *n*INP from Table 1 is roughly between $10^{-2}$ and $10^3$ m$^{-3}$ in the *T* range between -9 and -25°C, in which we detected ice nucleation activity in our samples. This *n*INP range covers the majority of our measurements".

Line 312ff: Can the higher nINP from condensation mode measurements compared to immersion mode be confirmed from the studies listed in Tab.1?
Comparing *n*INP from various studies to discuss condensation vs. immersion modes may be misleading as ambient conditions, aerosol particle abundance and experimental parameters vary from one study to another. We cannot provide a quantitative response to this comment.

Line 320: give number of samples that span the 3-200m-3 nINP range.
Added as follows: "First, while Wex et al. (2019) report a very narrow concentration range (27-33 m$^{-3}$) at -22°C, having only three samples, our 61 DFPC plus WT-CRAFT data points span a much wider range (ca. 3-200 m$^{-3}$)".

Line 323ff: The difference in nINP depending on the sensitivity of methods indicates that the INP population is highly variable and the variation is underestimated at the detection limits of methods.
The reviewer is right. We just wanted to point out that our WT-CRAFT's lower detection limit ($\approx$ 1 INP m$^{-3}$ in the range of 0.93 – 1.02 m$^{-3}$ as mention above) is not as good as what is reported in Wex et al., and this difference should be noted to the reader. We rephrased this sentence to clarify our point as:
"The difference in *n*INP towards a lower bound is due to different sensitivities and detection limits of the two methods: WT-CRAFT (ca. 1 m$^{-3}$) and the immersion freezing measurement technique used by Wex et al. (2019, ca. $10^{-1}$ m$^{-3}$)".

Line 325: Clarify for what the nINP range is substantial. Below what concentration would it be negligible? Line 328: give nINP at -22°C to compare to the range at -22°C on line 325. Line 333ff: add nINP at -22°C measured at the cited locations.
The Paragraph associated with these comments was removed upon request of the other reviewer.

Line 344: As I understand a normal distribution is assumed to obtain the difference at p<0.005. I doubt that the dataset (<20 measurements per season) is large enough to obtain a valid distribution to perform statistics.
We have now presented our statistical approach to testing differences between datasets more in detail in the new Sect. 2.4, as presented above. We have used both the t-test (assuming normal distributions of the data) and the non-parametric Wilkoxon-Mann-Whitney test (not requiring normally distributed data). We have considered statistically significant only differences that resulted significant for p<0.05, according to both tests. We point out that this (same result from both parametric and non-parametric tests) resulted true for the large majority of the tested cases, which suggests that the normal distribution assumption was not so far from reality in many cases. For homogeneity, we now report through the text only the indication of the minimum tested significance level (p<0.05) even in cases that resulted significant for higher confidence levels.

Line 346: From Tab.2 it can be seen that he median nINP,PM10 is the same in summer and spring. The nINP,PM1 is lower in summer thus contradicting the conclusion that coarse INP from exposed surface cause the difference. The higher nINP,PM1 in spring could indicate INP from the arctic haze.

Yes. This is exactly our interpretation, and we do not find any contradiction between Table 2 and our conclusions. The fact that $n$INP$_{PM10}$ is not statistically different between spring and summer, in the DFPC dataset, at $Ts$ of -18 and -15°C, is discussed in detail in the manuscript. But, what is important is the fact that the coarse INP fraction is larger in summer at the same $Ts$ than in springtime. Our data-driven conclusion is that the spring and summer-time aerosol particle populations are different: the spring one is mostly comprised of sub-micrometer INPs, likely transported from lower latitudes (Arctic haze), while the summer one is comprised of both fine and coarse particles (with coarse particles often exceeding a 50% contribution). There is a clear consensus in the literature (e.g., Udisti et al., 2016, Browse et al., 2012) on the fact that the Arctic haze period does not extend into the summer season. Considering the short atmospheric lifetime of aerosol particles, it is reasonable to assume that the spring and summer time aerosol populations are different and due to different sources (distant PM$_1$-dominated sources in spring; local sources providing a high contribution of coarse particles in summer).

Browse, J., Carslaw, K. S., Arnold, S. R., Pringle, K., and Boucher, O.: The scavenging processes controlling the seasonal cycle in Arctic sulphate and black carbon aerosol, Atmos. Chem. Phys., 12, 6775–6798, https://doi.org/10.5194/acp-12-6775-2012, 2012.

Line 347: on line 344 the contribution of coarse INP is described as significant to a high level of certainty. How were the "substantial uncertainties" considered to estimate the significance of an increase in coarse INP contribution?

The consideration regarding the "substantial uncertainties" refers to the fact that particle number measurements are generally affected by lower uncertainties than INP measurements. The increase in the contribution of coarse INPs in summer is clear and confirmed by all the performed statistic tests (discussed above). It is also true that the absolute concentration of coarse INPs is generally higher in summer than in spring, with a statistically significant difference (p<0.05, n1=15, n2=17) for T of -18 and -15°C. This can be assessed from the plots below.

[Figure]

[Figure]

Line 349: On the PM1 filters not only particles larger than 0.5um are sampled, as was assumed for the fractions shown in Fig. S2. I suggest using the full size-range below 1um for the fine particle concentration to calculate the change in the particle population.

We took the reviewer's word for it. We now use the aerosol particle size range of 0.1-10 µm for our AF estimation.

Line 350: The coarse INP fraction doesn't dominate. Looking at Tab.2, the fine fraction makes up 80% of nINP in spring and 50% in summer.

The "coarse fraction dominated" refers to the summer season, where coarse particles are averagely from 45% (-22°C) to 65% (-15°C). Anyhow we admit that "dominated" may not be the most appropriate term. We have clarified it as follows:

"An INP population similar to summer values at GVB, with a significant coarse fraction contribution, was reported by Mason et al. (2016), from the Alert Arctic station. The authors conducted INP measurements from 29 March to 23 July 2014 and observed an increasing contribution of coarse INPs as a function of the activation T".

Line 352: As mentioned on line 350, Mason et al. also reported a spring to summer dataset, making the results not "unique".

Mason et al. did not distinguish between spring and summertime INPs. From the paper, it is not possible to asses if they observed a seasonal increase of the coarse fraction contribution at Alert. In that sense, our publication is "unique" as it presents this finding for the first time for the Arctic atmosphere.

Line 360: Repeat here how coarse and fine AF are calculated, i.e. particle size-range used and nINP coarse as difference of PM10-PM1 nINP. Also mention it in the caption of Fig.3. What particle size range was used for the WT-CRAFT (sampling through a TSP inlet) AF?

The requested information have been added (Sect. 2.3.1).

For WT-CRAFT, we used the same size range used for PM10 DFPC samples. Our size distribution measurements were made through the TSP inlet, under which the WT-CRAFT filter sampler was deployed. Our APS data shows negligible contribution of > 10 µm particles in terms of number concentration in 2018. The aerosol particles measured by APS for > 10 µm accounted for on average only 0.07% of supermicron particles in the range of 1-20 µm during our sampling period in 2018.

Line 360, 414, 544, 573, 579, 585, 587: Ice nucleation efficiency/ability/activity/capability was not measured in this study, but concentrations of INP. Ice nucleation efficiency requires knowing the concentration of the ice active species. To be more consistent replace them throughout the paper with nINP or AF where appropriate.

The AF can be understood as a simple metric that indicates the ice-nucleating efficiency of particles within a specific aerosol sample (Schrod et al., 2020). We have added this "definition" at the end of the Introduction

Section and used it through the document. We agree with the reviewer that we did not measure the AF, but only $n$INP and aerosol number concentration; therefore, we have amended the text from any misleading expressions referring to "measurements of AF".

Line 362: the difference would be much larger if fine particles would not be limited to particles larger than 0.5um.

We confirm it. By calculating the AF using particles from 100 nm, the AF of coarse particles results in almost 2 orders of magnitude higher than that of fine ($PM_1$) particles.

Line 363ff: instead of listing AF, point to Fig.3. Disentangle if the AF is governed by changes in particle concentration or nINP.

We thank the reviewer for the suggestion. In this Section, we have provided more details on the variation of AF as a function of the particle size. The higher AF of coarse particles results from the significantly lower particle number concentration in the coarse mode (relative to the fine one), combined with the comparable $n$INP in fine and coarse modes. Sub-micrometer particle number concentration during our study was of the order of 10 to 200 $cm^{-3}$, while in the coarse mode it was less than 1 $cm^{-3}$.

"Examining the size-segregated DFPC data (Fig. 3a and b), substantially higher ice nucleation efficiencies were found in coarse compared to sub-micrometer particles. The enhanced AF of coarse particles is due to the significantly lower number of particles in the 1-10 μm compared to the 0.1-1 μm size range (about two orders of magnitude), coupled with the comparable $n$INP observed in both size ranges (see Sect. 3.2). As a result, the AF of coarse particles was more than 2 orders of magnitude greater than that of fine particles. In other words, the AF for coarse particles was estimated to be in the order of $10^{-6}$ to $10^{-3}$ at $Ts$ between -18 and -22°C, while the AF of sub-micrometer particles was in the order of $10^{-8}$ to $10^{-5}$ at the same $Ts$".

In the following Section (3.4), we added more details on the seasonal variation of AF.

"Both the DFPC and WT-CRAFT datasets showed a general increase of the aerosol particle AF from spring to summer as shown in Fig. 5. This increase in the AF is mainly due to a significant reduction of the particle number concentration in the 0.1-10 μm range (p<0.05; $n_1>10^3$; $n_2>10^3$; Fig. S4), combined with similar or slightly higher $n$INP (depending on the $T$). DFPC showed a statistically significant AF increase (p<0.05; $n_1$=16, $n_2$=17) going from the spring campaign to the summer period for all the probed activation $T$s. The seasonal increase in the AF was more evident at higher $Ts$: the summer to spring mean ratio was 6.2 at $T$ of -15°C and 2.5 at $T$ of -22°C. Fairly consistent results can be observed in the WT-CRAFT dataset. Comparing the samples collected before June 3 with those collected after that date, an AF enhancement (from 1.1 to 3.7 fold) can be estimated for all the activation $T$s. This difference was statistically significant (p<0.05, $n_1$=11, $n_2$=15) for all the activation $T$s between -17 and -22.5°C. Unlike the DFPC data, the spring to summer AF increase from WT-CRAFT data peaked at $T$ = -20°C (3.7; Fig. S5)".

Line 373: mention what particle size-ranges Si et al. used.

The paragraph was updated as follows:

"In particular, Si et al. (2018) reported average AF at -25°C of ~$10^{-4}$, $2\times10^{-3}$ and $6\times10^{-2}$ for the 0.56-1.0, 3.2-5.6 and 5.6-10 μm size ranges, respectively".

Sec.3.4: This section needs to be structured better. It is currently unclear what the important results are.

The Section was reformulated for major clarity. Please refer to the revised text.

Line 387, 391ff: factors of 1.5 or 1.6 are very small differences to draw conclusions. Additionally, the number of samples could be too small to determine the underlying nINP distributions on which the comparisons in this section are based on.

That is our point: we do not see any important (and significant) increase. Increase or decrease factors of 1.5 and 1.6 indicate an almost flat $n$INP trend. In brief, our results are more in Line with those by Schrod et al. (2020), presenting a flat trend, than with the previous publications showing a sharp increase of $n$INP in summer.

The number of samples has been already commented on above.

Line 395: State the number of June samples and explain how it was determined that they represent a significant peak.

During June, 7 WT-CRAFT samples have been collected. At all $Ts$ <-18°C, more than 50% (from 57 to 71%) of the June $n$INP data points are higher than the whole campaign median. For $Ts$ between -19.5 and -21.5°C, 57% of the data points are also above the 75$^{th}$ percentile. This percentage varies between 14% and 43% for the remaining $Ts$. Furthermore, at all $Ts<-18°C$, the average and mean values of the June samples are higher than the campaign median.

The requested information was added as follows: "Of the 7 samples collected in June, more than 50% (57-71%) were higher than the whole campaign median at this $T$ range. In addition, the average $n$INP during June was up to ~3 ($T$ = -20°C) times higher than the average for the rest of the observation period".

Line 409ff: The effect of particle concentration and nINP need to be disentangled. The absence of a change in nINP with season points to the particle concentration causing the change in the AF.

This issue is now addressed as explained above. Anyhow, we point out that whatever the cause (increasing $n$INP or decreasing particle number concentration) the significant increase in AF demonstrates that the summertime aerosol population is generally more ice active than the spring one.

Line 430: the 50-120nm size range mentioned here provides additional evidence against the chosen lower size limit of 500nm for this study.

Addressed above.

Line 462, Fig.6: The trajectories with land contact are not visible in Fig.6. Removing the not used grey trajectories could help.

They are barely visible because land contact are sporadic. Removing grey lines does not improve the plots. We would like to retain this figure as it is.

Line 468f: specify that the -15°C, PM1 dataset was used. It is unclear why the coarse and fine data is an advantage for DFPC. There is no comparison between coarse and fine nINP shown here, except Figs. S7 and S8. Do these maps confirm Mansour et al. and McCluskey et al.?

Done. Indeed, the maps confirm the results of Mansour et al. and McCluskey et al. Similarly to the cited papers, correlation is evident only for PM1 samples. Using PM10 or TSP data results in no correlation; from that derives the advantage of DFPC data.

Line 472f: Has the analysis been tried for -18°C and -22°C? It is mentioned on line 494ff that experiments showed INP active at -22°C are generated.

Yes. The analysis was performed also on $n$INP datasets at T=-18 and -22°C. The resulting correlation maps were less clear, even though some of the features observed at $T$ of -15°C also appear in these $Ts$. We interpret it as the effect of background land INPs, presuming the influence of dust, which is more active at low temperatures. We notice that in a previous deployment of the technique, we obtained significant correlation maps for Mace Head station using $n$INP at $T$ of -22°C (Mansour et al., 2020b). Nevertheless, in that case the sampling occurred in carefully selected clean marine air masses, which limited the contribution of land sources virtually to zero. Unfortunately, such a technical solution was not applicable at GVB.

Line 483: Provide an interpretation of significant, negative correlated areas. Are they caused by elevated CHL without the expected nINP response?

The basic assumption for the time-lagged correlation approach (Rinaldi et al., 2013; Mansour et al., 2020a; b) is that marine biological aerosols (in this case, organic-enriched sea-spray particles acting as INPs) should follow the evolution of marine biological activity (traced by CHL). For this reason, we target positively correlating sea regions: in our interpretation, these regions have a higher probability of being related to the observed aerosol properties. An inverse correlation with CHL cannot be explained by a physical mechanism: if we assume the INP are biogenic, their concentration can only increase with increasing algal activity (positive correlation) and decrease with decreasing algal activity. Therefore, we excluded negative correlating regions from the analysis as all the non-correlating regions. They are certainly not associated with INP emission, even though we cannot assess their level of productivity.

Line 487: see comment on Fig.7.

Line 534ff: There could be an increase in biological, high T INP from growing biota on CFPC filter during storage at room T.

This hypothesis is very hard to address at present. Maybe additional analysis of heat treated samples could provide some indications in this sense. Unfortunately, the DFPC technique is not suitable for this kind of tests. Actually, we do not know either if the INP properties of our samples are related to the presence of entire cells or cellular exudates or fragments. The $PM_1$ size range is unlikely to host entire cells, submicron sea-spray organic matter is mostly composed of exudates (e.g., Facchini et al., 2008; Orellana et al., 2011). This is also confirmed by Wilson et al. (2015), which showed that filtering sea-surface microlayer samples by 0.2 µm pore size does not modify the INP properties. This points to INP properties not related to entire organisms, but more likely to molecules, molecular aggregates (the so-called nanogels) or small cell fragments. This suggests that the biological material collected on the filters (and particularly on the PM1 samples) should not be able to "proliferate" during storage.

Facchini M.C., et al., Primary submicron marine aerosol dominated by insoluble organic colloids and aggregates, GEOPHYSICAL RESEARCH LETTERS, 35, L17814, doi:10.1029/2008GL034210, 2008.
Orellana M., et al., Marine microgels as a source of cloud condensation nuclei in the high Arctic, PNAS, 108, 33, 13612–13617, 2011.

Line 538: It could be added that similar air volumes were collected on the filter for CFPC and WTCRAFT.
This is a very useful suggestion. Thank you and it has been added.

Line 549: It could be expected that soluble compounds suppress ice nucleation. Why would the opposite be observed for these samples?

We note that the effect of soluble compounds on the ice nucleation process is a complex problem and partially unsolved question. As a matter of fact published experimental measurements don't give a unique answer. Reischel and Vali performed a study to determine the influence of 22 different salts dissolved in water on four different nucleants. The responses were complicated, with enhancements and suppressions of ice nucleation much larger than what would be expected from changes in water activity only. More recently, Boose et al. (2016) found that the presence of a soluble salt ion leads to an improved ice nucleation ability of dust particles. INP concentrations were measured in the deposition and condensation mode at temperatures between 233 and 253K. Paramonov et al. (2018) examined surface dust collected from three different locations around the world with respect to its ice nucleation activity in deposition and condensation freezing modes. In order to remove the soluble material, the dust samples were washed. Some of the soil

dusts results in an increase or decrease of ice nucleation activity after the washing procedure. Kumar et al. (2018), in laboratory experiments, found an increase of ice nucleation efficiency for K-feldspar microcline only in dilute NH4+ solutions, and a decrease in more concentrated solutions. This result was confirmed by Whale et al. (2018), who found that droplets of suspended particles of feldspars and quartz in ammonium solutions, nucleated ice up to around 3°C warmer than pure water case. Belosi et al. (2019) investigated, by means of the DFPC, the ice nucleating effectiveness of Arizona Test Dust (ATD), bare and coated with NaCl. Results showed a decrease in the AF and $n_s$ between bare and ATD coated with NaCl both at Sw=1.02 and Sw=0.96.

The cited papers confirms that it is difficult to predict the effect on the measured INP number of removing the soluble fraction from ambient aerosol particles (as in immersion freezing WT-CRAFT procedure) or analyzing particles mostly undisturbed (as in the DFPC procedure). This makes our hypothesis reasonable.

The text was updated treating briefly the above discussion:

"The literature offers diverse results and data interpretations, evidencing both increase and suppression of the ice nucleation ability by soluble aerosol components (Reischel and Vali, 1975; Boose et al., 2016; Paramonov et al., 2018; Kumar et al., 2018; Whale et al., 2018)".

Belosi F., Santachiara G., Laboratory investigation of aerosol coating and capillarity effects on particle ice nucleation in deposition and condensation modes, Atmos. Res., 230, 2019.

Reischel M.T., G.Vali, 1975. Freezing nucleation in aqueous electrolytes Tellus, 27, 414–427.

Boose et al., 2016. Ice nucleating particles in the Saharan Air Layer Atmos. Chem. Phys., 16, 9067–9087, 2016.

Kumar et la., 2018. Ice nucleation activity of silicates and aluminosilicates in pure water and aqueous solutions – Part 1: The K-feldspar microcline. Atmos. Chem. Phys., 18, 7057–7079.

Paramonov et al., 2018. A laboratory investigation of the ice nucleation efficiency of three types of mineral and soil dust. Atmos. Chem. Phys., 18, 16515–16536.

Whale et al., 2018. The enhancement and suppression of immersion mode heterogeneous ice-nucleation by solutes. Chem. Sci. 9, 4142-4151.

Line 552: A detailed comparison is not needed. To claim sensitivity of arctic nINP on the ice nucleation mode, exemplary DFPC filter need to be analysed with the WT-CRAFT method and vice versa. Otherwise it is speculative.

Yes. This suggestion is valid. The reason why we could not share the filter for both techniques was provided in a previous answer. In addition, in the revised manuscript, we limited to present the sensitivity of Arctic nINP on the ice nucleation mode as a possibility. Please, refer to the revised Sect. 4.1 for more details.

Line 566: It seems plausible that due to small number of samples, the INP population was not well characterized in aforementioned studies, leading to misinterpretation.

We also think the lack of long-term Arctic nINP data is problematic, and we clearly suggest the necessity of future long-term INP monitoring with an online instrument in the same paragraph. Nevertheless, we do not feel comfortable mentioning the number of samples used for previous works as a misleading factor and prefer to avoid discussing it.

Line 578: state which results
Done, in the following way:
"Our AF estimates support the hypothesis that…".

Line 591: where is shown that land sources have this potential? PM10 nINP do not change much even when more land and open sea are along trajectories and PM1 nINP decrease.

We have reformulated this part, also adding the appropriate reference, to support our statement: "Our analysis points out that both marine and terrestrial sources may contribute to the INP population in the study area. Land sources may be potentially important given the higher ice activity of mineral dust and soil particles in comparison to marine particles (McCluskey et al., 2018). On the contrary, marine sources may be significant because of the extension of ice-free sea waters during the Arctic summer".

Line 591, 592: at what T have mineral dusts a higher activity and at what T are marine INP less active?
This information is already addressed in the Introduction section.

Line 598: specify the results, e.g. "location of hot spots for marine INPs"
Added: "In particular, the approach adopted highlights the sea waters to the southwest of Svalbard, those immediately to the east of Greenland and to the northeast of Iceland as potential INP hotspots during our summer campaign".

Line 610f: The presented analysis does not support this conclusion. A more plausible reason could be differences in the DFPC and WT-CRAFT samples. As mentioned above, to arrive at this conclusion the same samples must be analysed with both methods. Replace "undeniable" by "potential".
We have reworded the text as: "We considered many factors that could potentially explain the discrepancy observed (Sect. 4). While differences in the sampling approach and overall measurement uncertainties have certainly contributed to the offset, a different response of aerosol particles to the ice nucleation mode could be also considered as a potential contributing factor. All future investigations into Arctic INP compositions and the ice nucleation process employing both the condensation and immersion freezing approach will provide further understanding of this issue".

Line 616: The dataset is not unique, other studies have collected Arctic INP covering different seasons e.g. Manson et al., Schrod et al.
Reworded as: "This study also examined the seasonality of INPs in the Arctic with respect to $n$INP and AF….".

Line 624: how did the back-trajectories show this separation? They were done for summer only.
All the BTs are presented in Figure 6.

Line 629: repetition of line 625
Modified to avoid repetition.

Tab.1: Consider making 3 columns showing the nINP at -15°C, -18°C, -22°C, relevant for this study, instead of the current last two columns. A minus sign is missing in Bigg, 1996 T range.
Not all the papers present $n$INP data at these precise $Ts$, so we preferred the presented option to provide a wider overview.

Tab.2: Subtracting 1-3 standard deviations from the average values gives negative concentrations. nINP can not be negative, the real variation is clearly asymmetric. Consider reporting the range and average instead and refer to the table instead of listing these values in the text.
The fact that the standard deviation is higher than the mean value depends on the data distribution and is fairly common in atmospheric data. In the revised version, we reported the standard error, instead. Furthermore, we provided the data range, together with the median value, for completeness.

Tab.3, 4: Scatterplots for all 39 significant correlations reported in these tables could be shown in the supplement and investigated for outliers to exclude false positives.

Scatter plots for all the correlations (significant and not) have been added in the Supporting (Fig. S7-S10).

Fig.3: mention the particle size range used to calculate AF in a), b) c)
Done: "Particle size ranges used to calculate AF are 0.1-1 μm, 0.1-10 μm and 0.1-10 μm for DFPC PM1, DFPC PM10 and WT-CRAFT samples, respectively".

Fig.4: indicate what is shown in a), b), c) in the caption. Specify that uncertainties can be found in sec.2.2.1 (CFPC) and 2.2.2 (WT-CRAFT)
Done.

Fig.6: consider removing grey trajectories.

Fig.7: specify which region belongs to which time lag. Consider overlaying the trajectories as thin lines to indicate upwind locations and clarify the choice of regions.
The regions corresponding to the different time lags are now indicated in the revised caption.
BTs are considered by merging the correlation maps with the CWT plots (which are BT-based). No region can be evidenced in Figure 8b if it is not upwind to the sampling station. So there is no reason to make these plots more complex to present redundant information.

Fig. 8 b): consider showing the trajectories corresponding to the 14 INP samples as thin lines
Only areas covered by the BTs (i.e., upwind to the sampling location during the sampling periods) can be evidenced in this plot as the CWT is BT-derived. So there is no reason to overlap the BTs.

Tab. S1: Column headers "Total Flow (optimized for 50% of filter)" and "Suspension water volume (First frozen drop=0.001 INP L-1)" are unclear. Give a description in the legend and rename the columns, e.g. "air volume" and "suspension water volume". Check the calculation of the suspension volumes.
Done – see above.

Fig.S3: consider using the same y-axis scale for subplots showing the same temperature. Even if there are only 3 spring datapoints in Wex et al. at this T, add -22°C for DFPC and WT-CRAFT as the seasonal change is discussed in the main text. Give a description of the subplots a)-f) in the figure legend.
The seasonal trend at T=-22°C is already in Figure 3a.

Fig. S4: Describe y-axis in the figure legend. Is PM1 or PM10 shown for DFPC? Mark the increase in AF due to the decrease in particle concentration from spring to summer (= particle conc. summer/particle conc. spring) as horizontal line. Spring to summer AF increase close to this line indicate no change in nINP.
Respectfully, we believe that this way of presenting the plot would suggest wrong conclusions. The particle populations are not the same in spring and summer, so even though the AF variation was due only to particle number concentration decrease (being $n$INP constant), this would imply nonetheless a seasonal modification in the general ice nucleation properties of the particle population (i.e., particles are more efficient in forming ice in summer than in spring).

Technical corrections
Provide more links/references between sections and to tables and figures to navigate the paper.
Line 27: remove "trustful"
Done.
Line 79: "suggested them to be…"
Done.

Line 200: remove "now"

Done.

Line 210: replace "side-by-side" with "in paralle".

Done.

Line 413: replace "point sin" with "points in"

Done.

Tab. 4b: remove "I" before nINP in the caption.

Done.

Fig. S1: change (a) -18°C to (b) -18°C in the second figure.

Done.

We would like to express sincere gratitude for the referee's helpful comments. Below, we provide our point-by-point responses (in blue colour).

Review of "Ice-nucleating particle concentration measurements from Ny-Ålesund during the Arctic Spring-Summer in 2018" by Rinaldi et al.

General comment:
The authors incorporated several changes in the revised version taking into account most of the listed comments from my initial evaluation. The readability of the manuscript was improved, in part by the new structure and the new additions (text, figures/tables, and data, etc.). Given that several parts of the revised version are completely new, additional concerns appear to me. Although I appreciate the efforts made by the authors, I cannot recommend the publication of this manuscript until the following points are carefully addressed.

We appreciate these general remarks. Following the referee's advice, we revised our manuscript structure and contents to improve the readability and conciseness of this paper. Below, we provide our point-by-point responses.

Major comments:
1. Although English is not my mother tongue, I think that it still needs to be improved. Some examples related to the Introduction only (this apply to the entire text) are provided below.

In the revised version, the text has been language-checked by a native speaker with expertise in the editing of scientific texts.

2. In the first round of comments I asked the following major point:
What is different or what is novel in the present study compared to Wex et al. (2019) and Hartmann et al. (2019)? The answer was that "In this study we present parallel observations of immersion and condensation INPs, which was never achieved in the Arctic. One of our major findings is indeed related to the different behaviour of aerosol particles sampled at GVB under different ice nucleation modes".

We believe this is still a valid uniqueness. We compiled past studies of $n$INP in the Arctic in Table 1, and we see no studies reported $n$INP from multiple ice nucleation measurement techniques. We invite the reviewer to read below for further considerations on the novelty of our manuscript.

However, the authors stated in the manuscript and in the reviewer's response (citing a paper from one of the coauthors) that both heterogeneous ice nucleation mechanisms "might be the same process". Therefore, they are contradicting themselves.

Here, we only intended to introduce what has been studied for condensation vs. immersion, by referring to the two cited studies. We did not mean to conclude that condensation = immersion in the Introduction section. Further assessments from the laboratory and field settings are needed to understand the similarity of ice nucleation modes and processes. Whether condensation and immersion are the same processes or not highly depends on the aerosol properties. We discuss this point in the revised Sect. 4.1.

For major clarity, we rephrased the sentence in the introduction section to: "However, the recent inter-comparison study with two different organic fiber samples shows a difference between condensation freezing and immersion freezing measurements (i.e., ice nucleation efficiency of the former is higher than the latter (Hiranuma et al., 2019). Further laboratory and field assessments are therefore necessary to understand the similarity of ice nucleation modes and processes".

If they authors would like to state that they are reporting the INP concentrations from two different mechanisms, a more in depth analysis is needed where the ice nucleation efficiencies of different aerosol "standards" (e.g., SNOMAX, illite, ATD, etc.) from both instruments are reported. Such experiments need to be run in parallel and using the same aerosol type and particle size. Currently, it is unclear if the observed differences are related to the used methods.

We agree with the reviewer on the importance of understanding more in detail the ice nucleation modes and processes. Nevertheless, this is not the focus of the present paper. In our paper, we present the results of two different INP quantification techniques which provide complementary information. These techniques were developed to represent different ice nucleation modes. Independently on the fact that immersion freezing and condensation freezing may be or may be not the same process in the atmosphere, it is undeniable that aerosol particles are exposed to different conditions during the measurements in the DFPC and WT-CRAFT. For instance, soluble components remain over the sampled particles in DFPC analyses, while they are diluted in the bulk of the extraction water in the WT-CRAFT. Furthermore, in condensation mode measurements, water vapours condense on the surface of sampled aerosol particles, possibly triggering the pore condensation freezing (David et al., 2019; Wagner et al., 2016). In the case of immersion freezing measurements, the pore condensation freezing is not assessable because all particles are scrubbed in the bulk suspension water. For these reasons, it is reasonable to assume that ambient aerosol particles may respond differently to the different analytical techniques. However, we stress that this is not a central point of our work (see below for further discussion). Considering that our INP detection techniques are within an acceptable agreement, according to previously published INP intercomparisons (DeMott et al., 2017; Hiranuma et al., 2019), and that the observed concentration gap does not affect the conclusions presented in this study (given the consistency of the DFPC and WT-CRAFT time trends) we believe that our results do not need the support of an in-depth intercomparison study with aerosol "standards" to be published.

However, we considered the reviewer suggestion and conducted additional immersion freezing experiments of Arizona Test Dust (ATD) with WT-CRAFT, comparing the results to previously published ATD result by another immersion freezing technique, Colorado State University Ice Spectrometer (CSU-IS; Perkins et al., 2020). The ATD used for WT-CRAFT is the same sample used in Möhler et al. (2006) and Niemand et al. (2012) (Powder Technology Inc.; 0-3 µm). As ascribed in Perkins et al (2020), the A2 ATD (Powder Technology Inc.; ~22% <2.75 µm, ~58% <11 µm, and ~90% <44 µm) was used for the CSU-IS analysis. For the WT-CRAFT measurements, we prepared an original stock suspension of 0.1 wt% ATD in ultrapure water and two diluted suspensions (x100 and x1000). The diluted spectra and original spectrum were merged with a method introduced in the main manuscript Sect. 2.2.2. The figure shown below presents the estimated INP concentration per unit mass (nm, g$^{-1}$) with WT-CRAFT and CSU-IS in the temperature range of -5 °C ≥ T ≥ -25 °C and -2 °C ≥ T ≥ -24 °C, respectively. As seen in the figure below, we see a reasonable agreement between the two techniques within uncertainties for the assessed temperatures. Thus, we believe that the applicability of WT-CRAFT for immersion freezing experiment is validated within its detection limits and capabilities (Sect. 2.2.2).

[Figure]

**Figure.** Indirect comparison of immersion freezing efficiencies ($n_m$) of Arizona Test Dust estimated by WT-CRAFT and CSU-IS.

In addition, condensation freezing mode with the same ATD dust was investigated by means of the DFPC (Belosi F. et al., 2018). The obtained ice-active surface site density $n_s$ at T of -22°C and $S_w$ of 1.02 (ca. $10^{10}$ m$^{-2}$) was comparable with the results by Niemand et al. (2012) (compare Figure 1b of Belosi et al. with Figure 6 of Niemand et al.). Furthermore, an intercomparison between DFPC and FRIDGE (FRankfurt Ice Nucleation Deposition freezinG Experiment) was carried out with different dust types, including Illite, and the obtained $n_s$ values were in agreement between the two techniques (Belosi et al., 2016; see Figure below).

[Figure]

**Figure.** Comparison of $n_s$ of standard aerosol types obtained by DFPC and FRIDGE (from Belosi et al., 2016)

Although the above results cannot replace a systematic intercomparison study, they suggest that for simple aerosol standards, DFPC and WT-CRAFT respond consistently to other well characterized INP quantification techniques (and therefore, we can assume, consistently to each other). This suggests that the complexity of ambient aerosol particles may be a potential reason to explain the observed discrepancy between the two techniques at GVB.

Nevertheless, we fully understand the concerns of the reviewers about relying on ice nucleation modes for explaining the observed difference between $n$INP$_{DFPC}$ and $n$INP$_{WT-CRAFT}$. As we have made clear in the revised manuscript, we cannot be conclusive about this, and we hope to be able to further investigate this issue in future follow-up studies. To allay further misgivings, we softened our tones and revised the text to present the particle sensitivity to different ice nucleation modes only as a potential reason. In the revised

manuscript, we also discussed other potential factors for the observed discrepancy (please refer to the revised Sect. 4.1). Overall, the addressed potential factors include (1) differences in measurement uncertainties, (2) sampling apparatus, (3) sample storage protocols, (4) substrate types, (5) sampling durations and (6) ice nucleation paths (condensation vs immersion freezing).

Unfortunately, we could not reduce the potential discrepancy factors during our 2018 sampling campaign, for the following reasons:

(1) we cannot use an identical substrate for the two INP analysis techniques. We have assessed the applicability of the cellulose membrane (optimal for DFPC) in WT-CRAFT, and we found notable background artifacts from the blank substrate below -22°C. Thus, the same filter sample cannot be shared for both INP measurement techniques.

(2) Our aerosol samplers employed different sampling periods for optimized flow conditions and estimated detection limits. The aerosol sampler for DFPC employed high flows, which were needed for the desired cut-size (see Sect. 2.1). If we conducted sampling with a longer period than what we employed, DFPC would likely suffer from filter overloading issues. Likewise, reducing the sampling time for WT-CRAFT would have created detection limit issues.

Finally, we would like to clarify here that explaining the gap is not a major point of the study. Our work extends the INP observations at GVB, contributing to filling the present lack of observations in the Arctic. Furthermore, it provides information on the AF, which was never calculated before at GVB and which was only rarely addressed at Arctic sites. One of the major findings of our work is that the seasonality of *n*INP can be significantly different from what was observed in previous studies. This finding is supported by both the INP datasets, notwithstanding the concentration gap and all the sampling differences. Our results evidence potentially a great interannual variability and the necessity for further data coverage to better understand INP dynamics over the Arctic. Furthermore, we present and discuss for the first time the seasonality of the AF, intended as a proxy of the overall ice nucleation ability of the particle population at the study site. Finally, our study reports information on the ice behaviour of fine vs coarse aerosol particles, which was addressed only in a few more Arctic studies so far.

Belosi F., Schrod J., Nicosia A., Santachiara G., Prodi F., Weber D., Bingemer H., Off-Line measurements of ice nucleating particles, European Aerosol Conference, Tours 4-9 september, 2016.

Belosi F., Piazza M., Nicosia A., Santachiara G., Influence of supersaturation on the concentration of ice nucleating particles, Tellus B: Chemical and Physical Meteorology, 70:1, 2018.

David, R. O., Marcolli, C., Fahrni, J., Qiu, Y., Perez-Sirkin, Y. A., Molinero, V., Mahrt, F., Brühwiler, D., Lohmann, U., and Kanji, Z. A.: Pore condensation and freezing is responsible for ice formation below water saturation for porous particles, Proceedings of the National Academy of Sciences, 116, 8184–8189, https://doi.org/10.1073/pnas.1813647116, 2019.

Möhler, O., Field, P. R., Connolly, P., Benz, S., Saathoff, H., Schnaiter, M., Wagner, R., Cotton, R., Krämer, M., Mangold, A., and Heymsfield, A. J.: Efficiency of the deposition mode ice nucleation on mineral dust particles, Atmos. Chem. Phys., 6, 3007–3021, https://doi.org/10.5194/acp-6-3007-2006, 2006.

Niemand, M., Moehler, O., Vogel, B., Vogel, H., Hoose, C., Connolly, P., Klein, H., Bingemer, H., DeMott, P., Skrotzki, J., and Leisner, T.: Parameterization of immersion freezing on mineral dust particles: An application in a regional scale model, J. Atmos. Sci., 69, 3077–3092, 2012.

Perkins, R. J., Gillette, S. M., Hill, T. C. J., and Demott, P. J.: The labile nature of ice nucleation by Arizona Test Dust, ACS Earth Sp Chem. 4, 133–141, 2020.

Wagner, R., Kiselev, A., Möhler, O., Saathoff, H., and Steinke, I.: Pre-activation of ice-nucleating particles by the pore condensation and freezing mechanism, Atmos. Chem. Phys., 16, 2025–2042, https://doi.org/10.5194/acp-16-2025-2016, 2016.

Note that the size of the aerosol particles analyzed by both instruments are not the same as the pore size of the filters used by the DFPC (0.45 um) and WT-CRAFT (0.2 um) are not identical. Although it is clear that supermicron particles are likely the most efficient INPs, small particles (between 0.2 um and 0.45 um) can be soluble and change the composition of the droplets and their freezing points. Note that the used filters in both systems are neither the same. Although the authors argued in their answer that the pore size is not important, I disagree with them and I think that the combination of different filter type and different pore size can result in the capture of different aerosol particles.

The difference in pore sizes might not be a substantial factor to explain the gap between DFPC and WT-CRAFT measurements. While we cannot rule out the possibility that DFPC misses ice nucleation active aerosol particles in the size range between 0.2 and 0.45 μm, this difference might not substantially contribute to the gap as $n\text{INP}_\text{DFPC}$ is generally higher than $n\text{INP}_\text{WT-CRAFT}$. This point is now discussed in the revised Sect. 4.1.

Also, given that the sampling time of the samples analyzed in the WT-CRAFT was much longer (4 days) than the samples analyzed in the DFPC (4 h) it is very likely that the aerosol particles analyzed in both instruments are completely different. It would have been desirable that the same filter was analyzed by both methods.

The motivations for which it was not possible to analyse the same samples by the two techniques are enlisted above. The difference in sampling durations (~4 hours for the DFPC and ~4 days for WT-CRAFT) is a valid concern. Although we cannot exclude the possibility that short episodes of high INPs-containing air masses increased $n\text{INP}_\text{DFPC}$, it is unlikely that this factor can explain a systematic difference, like that object of this discussion. This would presume a strong $n\text{INP}$ daily trend with the maximum coinciding with the DFPC sampling time. We excluded this by analysing the daily evolution of the particle number concentration, which does not present any evident diurnal trend both in spring and summer (not shown).

3. (1) How can the authors confirm the absence of the liquid phase in the DFPC? (2) Is it really possible to have ice crystal growing on individual aerosol particles? From my personal experience it is very difficult to avoid that several particles enter in contact when collected in a filter. (3) Therefore, It would be nice if the authors can provide evidence of how the aerosol particles were distributed in their nitrocellulose filters and a picture on how an ice crystals forms on a single particle.

(1) The air enters the DFPC chamber through a perforated plate, spreads into an ice bed, and becomes saturated with respect to ice, but undersaturated with respect to water. Afterward, the air flows into the second chamber, grazing the filter located on a metal plate cooled by a Peltier device. Only in this small surface air becomes supersaturated with respect to water. The following Figure presents a scheme of the DFPC.

[Figure]

**Figure.** 1: air inlet; 2: minced ice; 3: slit and air temperature thermocouple; 4: filter; 5: filter temperature thermocouple; 6: Peltier cooling device; 7: thermocouple; 8: air outlet; 9: plexiglass cover; 10: observation slit.

The manuscript was modified as follows:

"Particle-free air entered the DFPC chamber through a perforated plate, spreading to an ice bed to become saturated with respect to ice but undersaturated with respect to water. The air then proceded to the filter, cooled by a Peltier device in contact with the supporting metal plate. Only at this point, did the air become supersaturated with respect to water. By controlling the $T$s of the filter and surrounding air, the samples could be exposed to different $T$s while keeping the water saturation ratio ($S_w$) above 1. The supersaturation ratio was calculated theoretically from vapour pressures of ice and water at the $T$s considered (Buck, 1981). More details of the DFPC working principle can be found in the supplement of deMott et al. (2018)".

DeMott P.J., Möhler O., Cziczo D.J., Hiranuma N., Petters M.D., Petters Sarah S., Belosi F., Bingemer H.G., Brooks S.D., Budke C., Burkert-Kohn M., Collier K.N., Danielczok A., Eppers O., Felgitsch L., Garimella S., Grothe H., Herenz P., Hill T.C.J., Höhler K.,  Kanji Z. A., Kiselev A., Koop T., Kristensen T. B., Krüger K., Kulkarni G., Levin E. J. T., Murray B.J., Nicosia A., O'Sullivan D., Peckaus A., Polen M. J., Price H. C., Reicher N., Rothenberg D. A., Rudich Y., Santachiara G., Schiebel T., Schrod J., Seifried T. M., Stratmann F., Sullivan R. C., Suski K. J., Szakáll M., Taylor H. P., Ullrich R., Vergara-Temprado J., Wagner R., Whale T. F., Weber D., Welti A.,  Wilson T. W., Wolf M. J., and Zenke J., " The Fifth International Workshop on Ice Nucleation phase 2 (FIN-02): Laboratory intercomparison of ice nucleation measurements", Atmos. Meas. Tech., https://doi.org/10.5194/amt-2018-191, Vol. 11, 6231-6257, 2018

(2) Avoiding the overlapping of particles on the filter is the main reason for the low sampling times of DFPC samples. By considering the sampled air volume (about 5.5 $m^3$) and the aerosol particle concentration in the range 0.1-10 μm (about 80 $cm^{-3}$), the probability of particles overlapping on the filter surface (47 mm diameter) is very low. More in detail, considering also the average aerosol particle size distribution at the sampling site, we estimated a total surface occupied by particles on the filter of the order of 0.02 $cm^2$ on a total filtering surface of ~10 $cm^2$. This means a low particle density on the filter.

(3) We attach a picture of ice crystals grown on one of the analyzed filters to show the sparseness of the ice crystals on the filter. Analysis conditions are the following: T(filter)=-22°C; T(air): -19.6°C; Sw: 1.02; PM1 fraction.

[Figure]

4. "nINP measured in condensation mode (DFPC) resulted generally higher than those measured in immersion mode (WT-CRAFT) and the deviation became even more apparent towards higher T". The authors need to clearly discuss this. How realistic is this observation and what is the reasoning for more aerosol particles to act as INPs via the condensation freezing? Also, why at higher temperatures condensation freezing becomes a more efficient pathway to catalyze ice formation compared to immersion freezing?

In the revised version, we considered a different response of aerosol particles to the different INP measurement techniques as a possibility, among others, mainly related to differences in the sampling approaches and measurement uncertainties. We admit that our observations are not conclusive and that further studies would be necessary to address this point. Please see our answer to comment #2 above.

5. The overall uncertainties of individual ice nucleation measurements cannot explain entirely the observed discrepancy". How the authors reached this conclusion. What type of analysis was it performed? How the different uncertainties were calculated and combined?

We compared DFPC and WT-CRAFT results during periods of parallel sampling to reduced the sources of variability in the results. To do this, we averaged multiple DFPC samples to match with the corresponding WT-CRAFT sample collected over the same period. For all the samples, $nINP_{DFPC}$ was higher than $nINP_{WT-CRAFT}$ and the difference was:

$\Delta\_nINP = nINP_{DFPC} - nINP_{WT-CRAFT}$.

For each sample, we calculated:

$\Delta\_nINP' = (nINP_{DFPC} - \sigma INP_{DFPC}) - (nINP_{WT-CRAFT} + \sigma INP_{WT-CRAFT})$,

where $\sigma INP_{DFPC}$ and $\sigma INP_{WT-CRAFT}$ are the absolute $nINP$ uncertainties for DFPC and WT-CRAFT respectively, calculated according to the overall relative uncertainties presented in Sect. 2.2.1 and 2.2.2. The assumption is that both DFPC and WT-CRAFT uncertainties contributed to their maximum to generate the observed $nINP$ gap. In other words, we considered the maximum possible underestimation error for DFPC and the maximum possible overestimation error for WT-CRAFT, to check if this was enough to close the $nINP$ gap. Comparing $\Delta\_nINP$ with $\Delta\_nINP'$, we estimated to what extent the measurement uncertainty can explain the observed concentration gaps.

The results of this comparison are reported in the revised text as follows:

"The uncertainties of individual ice nucleation measurements cannot entirely explain the discrepancy observed. Even considering the largest error contribution, uncertainties can on average explain up to 50, 66 and 76% of the observed $nINP$ offset at -18, -22 and -15°C, respectively. These percentages were calculated by assuming that the measurement uncertainties combined with each other to determine the maximum possible reduction of $nINP$ difference (i.e., assuming the maximum possible underestimation of $nINP_{DFPC}$ and the maximum overestimation of $nINP_{WT-CRAFT}$) and considering only periods of parallel sampling (to minimize sources of discrepancy unrelated to measurement uncertainty)".

6. "As we do not observe any strong indications of these influence within given uncertainties, it is at least conclusive that the size-dependant collection efficiency of aerosol particles is not a solo-dominating factor causing the difference between $nINP_{WT-CRAFT}$ and $nINP_{DFPC}$." This is in indirect conclusion without any clear robust evidence supporting it.

We admit that our evidence cannot be fully conclusive. Please, see the above considerations.

7. "the suppression of INPs due to the sample storage difference does not explain the observed general trend of nINPDFPC > nINPWT-CRAFT." Why not? How was this evaluated and confirmed?

The difference in sample storing methods cannot be a sole factor to explain the discrepancy. As we kept the samples for DFPC at the room air $T$ (and the WT-CRAFT samples at 4 °C except during transportation), the

suppression of INPs is expected to be more obvious for DFPC than WT-CRAFT, which is not supported by the observed trend of $n\text{INP}_{\text{DFPC}} > n\text{INP}_{\text{WT-CRAFT.}}$

We are aware that Beall et al. (2020) recently demonstrated that storage of precipitation samples can lead to losses of INPs (especially the heat-labile ones). Storage at room temperature and at 4°C resulted basically in INP losses, with the former conditions showing somewhat higher losses than the latter. Nevertheless, the results by Beall et al. (2020) are not necessarily valid for particle samples collected on filters. Furthermore, there is not yet a consensus in the literature (and within the scientific community) about the effect of sample storage on INP measurements, so answering this question quantitatively is very hard at present. For instance, Wex et al. (2019) stated that "it is not yet known with certainty how the temperature during storage will affect INP concentrations". Bigg (1990) reported no significant deterioration in INP concentration after 5 years, for filters sealed in a dry-sealed container. Conen et al. (2015) commented that storage of filters at -20°C may not have been necessary, because re-analysis of filters analyzed previously showed no effect of storage. Wex et al. (2015) and Polen et al. (2016) showed that storage at temperatures above 0°C or even under freezing conditions has been found to reduce the ice activity of biogenic INPs. Stopelli et al. (2014) studied INP concentrations in a snow sample stored at 4°C and observed a decrease in the concentration of INPs active at -10°C over 30 days (factor of 2).

We also evaluated the effect of sample storage on the $\Delta n\text{INP}/\Delta T$ slope, we evidence that DFPC samples were not refrigerated during storage, while WT-CRAFT ones were kept at 4°C. Therefore, based on the limited literature on this topic, degradation of INPs during storage would have caused higher losses in DFPC samples, resulting in the opposite effect on the $\Delta n\text{INP}/\Delta T$ slope. For this reason, we believe that losses during storage cannot be considered a valid explanation for the observed gap, at least based on the current knowledge of the topic. We now discuss this point in the revised Sect. 4.1.

Bigg, E.K., 1990. Measurement of concentrations of natural ice nuclei. Atmos. Res., 25, 397-408.

Conen et al., 2015. Atmospheric ice nuclei at the high-altitude observatory Jungfraujoch, Switzerland. Tellus B, 67, 1-10.

Conen et al., 2015. Atmospheric ice nuclei at the high-altitude observatory Jungfraujoch, Switzerland. Tellus B, 67, 1-10.

Polen, M., Lawlis, E., and Sullivan, R. C.: The unstable ice nucleation properties of Snomax (R) bacterial particles, J. Geophys. Res.-Atmos., 121, 11666–11678, https://doi.org/10.1002/2016jd025251, 2016.

Stopelli, E., Conen, F., Zimmermann, L., Alewell, C., and Morris, C. E.: Freezing nucleation apparatus puts new slant on study of biological ice nucleators in precipitation, Atmos. Meas. Tech., 7, 129–134, https://doi.org/10.5194/amt-7-129-2014, 2014.

8. "results from higher ice activation at the conditions of DFPC analyses rather than from higher aerosol particle number concentration in DFPC samples". I sort of agree with the authors; however, how about the chemical composition? No drastic changes in the aerosol particle concentration does not mean that the chemical composition along 4 days is constant.

This comment is pertinent. Nevertheless, we believe that a systematic aerosol particle chemical composition difference occurring only during DFPC sampling time, so that it can determine the observed difference, is highly unlikely.

9. "A detailed intercomparison of techniques is not under the scope of this study". I disagree with this statement. Given that the authors are claiming that they report the INPs concentration via two different ice nucleation modes, they need to provide convincing evidence and not unsupported statements such as "suggest that the unexplained concentration gap might stem from other factors, and it is plausible to consider a different sensitivity of Arctic INPs to different ice nucleation modes".

Discussed above – see our response to comment #2.

10. "we conclude that a different sensitivity of Arctic INPs to different ice nucleation modes explains the observed discrepancy." and "it seems conclusive to address there is ice nucleation mode dependent INP propensity at GVB in 2018 at least." I disagree with both statements.

Both statements have been removed, and other factors contributing to the observed discrepancy between DFPC and WT-CRFAT measurements are discussed in Sect. 4.1.

Minor comments:

1. I am not sure how useful is to use "T" and "Ts" instead of "temperature" and "temperatures".

The temperature word appears numerous times in our manuscript, and we decided to use an abbreviation for it to increase the conciseness and readability of our manuscript. We had a previous suggestion of doing this for another ACP manuscript. However, we understand that the ACP standard may have been updated and are willing to comply with the current preference/rule by the editorial team.

2. "We hypothesized that the nINP variability at a single T can be explained by differences in freezing modes." What does it mean?

We admit that it was not intuitive, and the sentence was removed from the revised Introduction. The potential factors of the observed $n$INP variability are now discussed in Sect. 4.1.

3. It is mentioned that the Back-trajectories were "simulated for an altitude of 100 m above mean sea level (amsl) over the GVB". Given that the long-range transport of aerosol particles was evaluated in the present study, why higher altitudes were not taken into account as the long-range transport of air masses does not take place that close to the surface?

For clarity, 100 m was the height of the arrival point of air mass evaluated by the back-trajectory analysis. It was assumed to be close to the ground because we sampled at the ground level. This does not mean that the trajectory always travelled at that same height.

4. "The range of nINP from Table 1 is roughly comprised between $10^{-2}$ and $10^3$ m$^{-3}$". I would restrict this to the temperate range of the present study.

We considered the range between -9 to -25°C which covers the range in which we observed some ice nucleation activities in our samples. Restricting it only to the DFPC temperatures would not be representative of our full dataset. The temperature range is now clarified in the revised text:

"The range of $n$INP from Table 1 is roughly between $10^{-2}$ and $10^3$ m$^{-3}$ in the $T$ range between -9 and -25°C, in which we detected ice nucleation activity in our samples. This $n$INP range covers the majority of our measurements".

5. "suggesting that the dominant INP sources may be located at long distances (scale of the order of 100s-1000s km)" How the authors reached those numbers?

We removed those numbers: we inserted this specification upon request of the other reviewer. Long-range transport is a common expression in the field, indicating sources located far from the sampling location. Usually, this expression is used in contraposition to "local sources", when it is not possible to provide more details. We believe in this case this general indication is enough.

6. Last paragraph Section 4.2. I suggest to expand this a bit more including previous studies where the Arctic aerosol composition and sources are discussed including new particle formation.

This is a good suggestion. The paragraph was extended as follows: "The higher AF of summertime (local) aerosol particles may be related to the enhanced contribution of super-micrometer aerosol particles, which

we have shown to be markedly more ice active than sub-micrometer particles. Nevertheless, we cannot exclude or quantify, the contribution of other physico-chemical properties of aerosol particles, which may vary between spring and summer (e.g., chemical composition).

It is worth considering that changes in the estimated AF are influenced not only by variations of $n$INP but also by variations of the concentration of non-ice-active aerosol particles, including secondary aerosols formed through new particle formation (NPF) mechanisms. Secondary aerosol particles may not contribute to INPs (Kanji et al., 2017), but they can lower the estimated AF. Recently, Beck et al. (2021) evidenced that different mechanisms, precursors and formation rates characterize spring and summertime NPF events at GVB. Dall'Osto et al. (2019) evidenced that the production of fresh particles is frequent during the period from May to August at GVB, while April is characterized by the presence of aged, accumulation mode particles. These aspects may influence the seasonal variation of the estimated AF. Dall'Osto et al. (2017; 2018; 2019) linked NPF frequency in the Arctic atmosphere to the fast-decreasing sea ice extent, probably via increased phytoplankton productivity. This leads to the hypothesis of increasing NFP impact in the future. By the same token, the predicted shrinking of snow and sea-ice coverage in the Arctic is likely to increase the ambient $n$INP from sea spray and terrestrial sources, such as mineral and soil dust particles (Tobo et al., 2019). Predicting future $n$INP and aerosol particle AF over the Arctic in such a rapidly changing scenario is challenging. It, however, provides the motivation for further investigation of INP processes in the Arctic region".

7. The authors showed the clear difference in the ice nucleating abilities of submicron and supermicron particles; however, little was mentioned about their composition. I suggest to add a little discussion about this based on the available literature for the Arctic trying to link the particles composition with their size.

In principle, this is a good suggestion and we already considered adding such considerations while working on the manuscript. Previous studies (e.g., Giardi et al. 2018, Zangrando et al. 2012, Young et al., 2016, to remain close to the study location) have shown that sub-micrometer aerosol particles in the Arctic are dominated by ammonium sulfate and nitrate, together with organic matter. These components can be both biogenic or anthropogenic, with a dominance of the former with respect to the latter which varies according to the period of the year (Arctic Haze). Coarse fraction aerosol particles are instead contributed by sea salt and mineral/soil dust particles, even though organic and biological particles can be also important in this size range. These differences certainly contribute to the different ice nucleation efficiencies of fine and coarse aerosol particles discussed in this work. Nevertheless, the bulk aerosol chemical composition can hardly be related to INP properties in a quantitative way, considering that only about 1 particle over $10^5$ can act as an INP. The bulk chemical composition depends necessarily on the rest of the particle population. Therefore, we preferred to avoid presenting considerations on the link between chemical composition and INP properties in the manuscript, which could be seen as speculative.

We clarified the necessity of identifying the chemical identity of the Arctic INPs in the revised Sect. 3.6.1: "It should be noted, however, that our tracer analysis only infers the aerosol properties, with the result that further analysis of INP identities and properties (e.g., ice crystal residual analysis) would be necessary to reveal the source of INPs".

Giardi et al., Rend. Fis. Acc. Lincei, DOI 10.1007/s12210-016-0529-3
Young et al., Atmos. Chem. Phys., 16, 4063–4079, 2016
Zangrando et al., Atmos. Chem. Phys., 12, 10453–10463, 2012

Technical Comments:

Line 41: Add a reference after "quantify".
Added ( Schmale et al., 2021)

Schmale, J., Zieger, P. & Ekman, A.M.L. Aerosols in current and future Arctic climate. *Nat. Clim. Chang.* **11,** 95–105 (2021). https://doi.org/10.1038/s41558-020-00969-5

Line 57: Change "condensation nucleus" by "INP".

Done.

Line 57: Change "a nucleus" by "INP".

Done.

Line 58: Add "at temperatures above 0C" after "water droplet".

Done.

Line 58: Change "in" by "via".

Done.

Line 58: Add "when T is decreased" after "immersion freezing".

Done.

Line 58: What do the authors mean with "extramural"?

Changed in: "In contact freezing, an INP promotes freezing when it comes into contact with a supercooled droplet from the outside".

Line 65: Remove "according to Fig. 13".

Done.

Line 66: "an exception of K-feldspar". Do the authors mean that K-feldspar it not a mineral?

We mean that it is an exception as it "can facilitate ice nucleation at much higher $T$s compared to other mineral compositions (Atkinson et al., 2013; Harrison et al., 2019)".

Line 67: "biogenic INPs". It was called biotic above.

We say that INPs can be of "biotic origin", which means they are "biogenic". We do not see any inconsistency in the terminology here.

Line 67: "to support". To favor?

Changed into favour.

Line 75: Delete "next".

Done.

Line 76: "The Ocean was considered to be a prevalent source of INPs". What do the authors mean?

Reworded: "The Ocean was identified as a major source of INPs,…"

Lines 76-77: "based on the high negative correlation between nINP and the time since the sampled air masses have been over the open ocean". Unclear.

We have rephreased the sentence as follows:

"The ocean was identified as a major source of INPs since $n$INP fell as a function of the length of time that had elapsed since the air masses had left the open sea".

Line 79: "probable submicron fragments". Unclear.

Reworded in "fragments of…".

Lines 82-83: "oceanic air tripled after about one day of passage over land". Unclear.

Reworded: "During the summer, the authors observed that $n$INP ($T$ of -15°C) in air masses from the ocean increased three fold after about one day of passage over land".

Line 94: Remove "six samples" and "seven samples" as I found it useless.

Done.

Line 96: Should "Santl-Temkiv et al., 2019; Wex et al., 2019" be "Tobo et al. (2019)"?

This sentence is not related to Tobo et al. (2019) specifically. We refer here to the enhancement of $n$INP reported by the all the cited papers (Wex et al., 2019; Santl-Temkiv et al., 2019 and Tobo et al. 2019). In this sense, the citation is not wrong as:

- Wex et al. (2019) state that: "In summer, the higher bioaerosol concentrations compared to spring indicate contributions from local and regional terrestrial and marine ice-free areas"; "these INPs can originate from both terrestrial and marine sources in the Arctic. These sources are strong in summer and weak or absent in winter, depending on the conditions on the ground".
- Santl-Temkiv et al. (2019) state that: "In summer, the higher bioaerosol concentrations compared to spring indicate contributions from local and regional terrestrial and marine ice-free areas"; "Based on this observation, we suggest that the high concentrations of INPs in air during summer may be partially related to strengthened local terrestrial sources, in particular soil dust"
- Tobo et al. (2019) state that: "This suggests that significant local sources of INPs other than marine organic materials might exist in and/or around the Svalbard region in the summer"

In brief, all the three papers attribute the summer time enhancement of $n$INP to local sources. We have added the reference to Tobo et al. (2019).

Line 99: "to fill the present gap of INP observations in the Arctic environment". The lack of measurements?

Reworded: "Our study aims to add to the still scant INP observations in the Arctic environment…".

Lines 100-101: "by two INP quantification techniques". This is very awkward.

Reformulated: "In particular, we extend the INP observations at GVB, previously only 13 samples (Wex et al., 2019), presenting the results of 61 samples investigated with two offline INP measurement techniques".

Lines 102-104: Delete "Recent modeling simulation and remote sensing studies suggest immersion freezing is the most relevant heterogeneous ice nucleation mechanism in mixed-phase clouds, which are prevalent in the Arctic (Hande and Hoose, 2017; Westbrook and Illingworth, 2011)."

Done.

Line 175: Delete "Next, we briefly explain our experimental procedure".

Done.

Lines 214-215: "The number concentration in the resulting overlapping range was taken from the SMPS data as SMPS provides more size bins". Unclear.

"The number concentration in the resulting overlapping range was taken from the SMPS data".

Line 228: "operating flow rate of 2.3 m3 h-1" for how long?

24 h, added.

Lines 325-235: Delete.

Deleted.

Line 510: "bio INP". It was not defined before.

"Bio" was removed.

---

## Author Response (AR3)

Editor Decision: Publish subject to minor revisions (review by editor) (08 Aug 2021) by Allan Bertram
Comments to the Author:
Dear authors,
Thank you for carefully considering the referee's comments and significantly improving their manuscript based on the referee's comments! I think all the modifications and responses are adequate, except for the response to the first main comment raised by Referee 2. Below I restate the referee's comment and suggest changes to further address the referee's comment.

Main comment:
"… It needs to be highlighted in the abstract and conclusion that results are preliminary because of the small sample size. Findings in the abstract and conclusion should be limited to the strong signals that are expected to be reproducable in future investigations. For less clear results it should be stressed that more observations are needed. The limitations of the applied analysis due to the size of the available dataset should be stated clearly in each section. …

In response to the main comment above, I think the authors have done a reasonable job of discussing the limitation of the applied analysis in Section 3 (Results) and Section 4 (Discussion). However, the limitations of the applied analysis are not acknowledged well in the Abstract or Conclusions, especially when discussing the sources of the INPs. Please edit the Abstract and Conclusion so that the limitations of the analysis for source identification are included in the Abstract and Conclusions and/or weaken the statements in the Abstract and Conclusions regarding the source of the INPs so that the Abstract and Conclusions are more consistent with Section 3 and 4. If you can make these changes, I think you will have adequately addressed all the referee's comments.

Sincerely,
Allan

Answer
We sincerely thank the Editor for appreciating our efforts to improve the manuscript and for this last suggestion. We have modified Abstract and Conclusions adding the requested considerations.
The abstract now reports the following sentence:
"Our spatiotemporal analyses of satellite retrieved Chlorophyll-a, as well as spatial source attribution, indicate that the maritime INPs at GVB may come from the seawaters surrounding the Svalbard archipelago and/or in proximity to Greenland and Iceland during the observation period. Nevertheless, further analyses, performed on larger datasets, would be necessary to reach firmer and more general conclusions".
The Conclusion Section now includes the following:
"Our summer-season analysis also suggests a relationship between the biological activity in specific seawater regions and $n$INP at the sampling point. Nevertheless, we evidence that this result was achieved with a limited number of observations and that further studies, based on larger datasets, would be desirable for a better understanding of marine sources of INPs over the Arctic".